# Causes and consequences of child growth faltering in low-resource settings

Andrew Mertens[1✉], Jade Benjamin-Chung[1,2,3], John M. Colford Jr[1], Jeremy Coyle[1], Mark J. van der Laan[1], Alan E. Hubbard[1], Sonali Rosete[1], Ivana Malenica[1], Nima Hejazi[1], Oleg Sofrygin[1], Wilson Cai[1], Haodong Li[1], Anna Nguyen[1], Nolan N. Pokpongkiat[1], Stephanie Djajadi[1], Anmol Seth[1], Esther Jung[1], Esther O. Chung[1], Wendy Jilek[1], Vishak Subramoney[4], Ryan Hafen[5], Jonas Häggström[6], Thea Norman[7], Kenneth H. Brown[8], Parul Christian[9], Benjamin F. Arnold[10,11✉] & The Ki Child Growth Consortium*

Growth faltering in children (low length for age or low weight for length) during the first 1,000 days of life (from conception to 2 years of age) influences short-term and long-term health and survival[1,2]. Interventions such as nutritional supplementation during pregnancy and the postnatal period could help prevent growth faltering, but programmatic action has been insufficient to eliminate the high burden of stunting and wasting in low- and middle-income countries. Identification of age windows and population subgroups on which to focus will benefit future preventive efforts. Here we use a population intervention effects analysis of 33 longitudinal cohorts (83,671 children, 662,763 measurements) and 30 separate exposures to show that improving maternal anthropometry and child condition at birth accounted for population increases in length-for-age $z$-scores of up to 0.40 and weight-for-length $z$-scores of up to 0.15 by 24 months of age. Boys had consistently higher risk of all forms of growth faltering than girls. Early postnatal growth faltering predisposed children to subsequent and persistent growth faltering. Children with multiple growth deficits exhibited higher mortality rates from birth to 2 years of age than children without growth deficits (hazard ratios 1.9 to 8.7). The importance of prenatal causes and severe consequences for children who experienced early growth faltering support a focus on pre-conception and pregnancy as a key opportunity for new preventive interventions.

Growth faltering in children in the form of stunting, a marker of chronic malnutrition, and wasting, a marker of acute malnutrition, is common among young children in low-resource settings, and may contribute to child mortality and adult morbidity[1,2]. Worldwide, 22% of children under 5 years of age exhibit stunting and 7% exhibit wasting, with most of the burden occurring in low- and middle-income counties[3] (LMICs). Current estimates attribute more than 250,000 deaths annually to stunting and more than 1 million deaths annually to wasting[2]. People who exhibit stunting or wasting in childhood also experience worse cognitive development[4–6] and worse economic outcomes as adults[7].

Despite widespread recognition of the importance of growth faltering to global public health, preventive interventions in LMICs have had limited success[8]. A range of nutritional interventions targeting various life stages during the fetal and childhood periods, including nutrition education, food and micronutrient supplementation during pregnancy, promotion of exclusive breastfeeding for 6 months and continued breastfeeding for 2 years, and food and micronutrient supplementation during complementary feeding, have shown beneficial effects on child growth[9–11]. However, postnatal breastfeeding interventions and nutritional interventions delivered to children who have begun complementary feeding have had only small effects on population-level stunting and wasting burdens, and implementation remains a substantial challenge[9,12,13]. Additionally, water, sanitation and hygiene interventions, which aim to reduce childhood infections that may increase the risk of wasting and stunting, have had no effect on child growth in several large randomized control trials[14–16].

Modest effects of interventions to prevent stunting and wasting may reflect an incomplete understanding of the optimal manner and timing of interventions[17]. In recent decades, this knowledge gap has spurred renewed interest in combining rich data sources with advances in statistical methodology[18] to more deeply understand the key causes of growth faltering[19]. Understanding the relationship between the causes and timing of growth faltering is also crucial because children who falter early could be at higher risk of more severe growth faltering subsequently. In the accompanying Articles, we present data showing that the highest rates of incident stunting and wasting occur by 3 months of age[20,21].

[1]Division of Epidemiology and Biostatistics, University of California, Berkeley, Berkeley, CA, USA. [2]Department of Epidemiology and Population Health, Stanford University, Stanford, CA, USA. [3]Chan Zuckerberg Biohub, San Francisco, CA, USA. [4]DVPL Tech, Dubai, United Arab Emirates. [5]Hafen Consulting, West Richland, WA, USA. [6]Cytel, Waltham, MA, USA. [7]Quantitative Sciences, Bill & Melinda Gates Foundation, Seattle, WA, USA. [8]Department of Nutrition, University of California, Davis, Davis, CA, USA. [9]Center for Human Nutrition, Department of International Health, Johns Hopkins Bloomberg School of Public Health, Baltimore, MD, USA. [10]Francis I. Proctor Foundation, University of California, San Francisco, San Francisco, CA, USA. [11]Department of Ophthalmology, University of California, San Francisco, San Francisco, CA, USA. *A list of authors and their affiliations appears at the end of the paper. ✉e-mail: amertens@berkeley.edu; ben.arnold@ucsf.edu

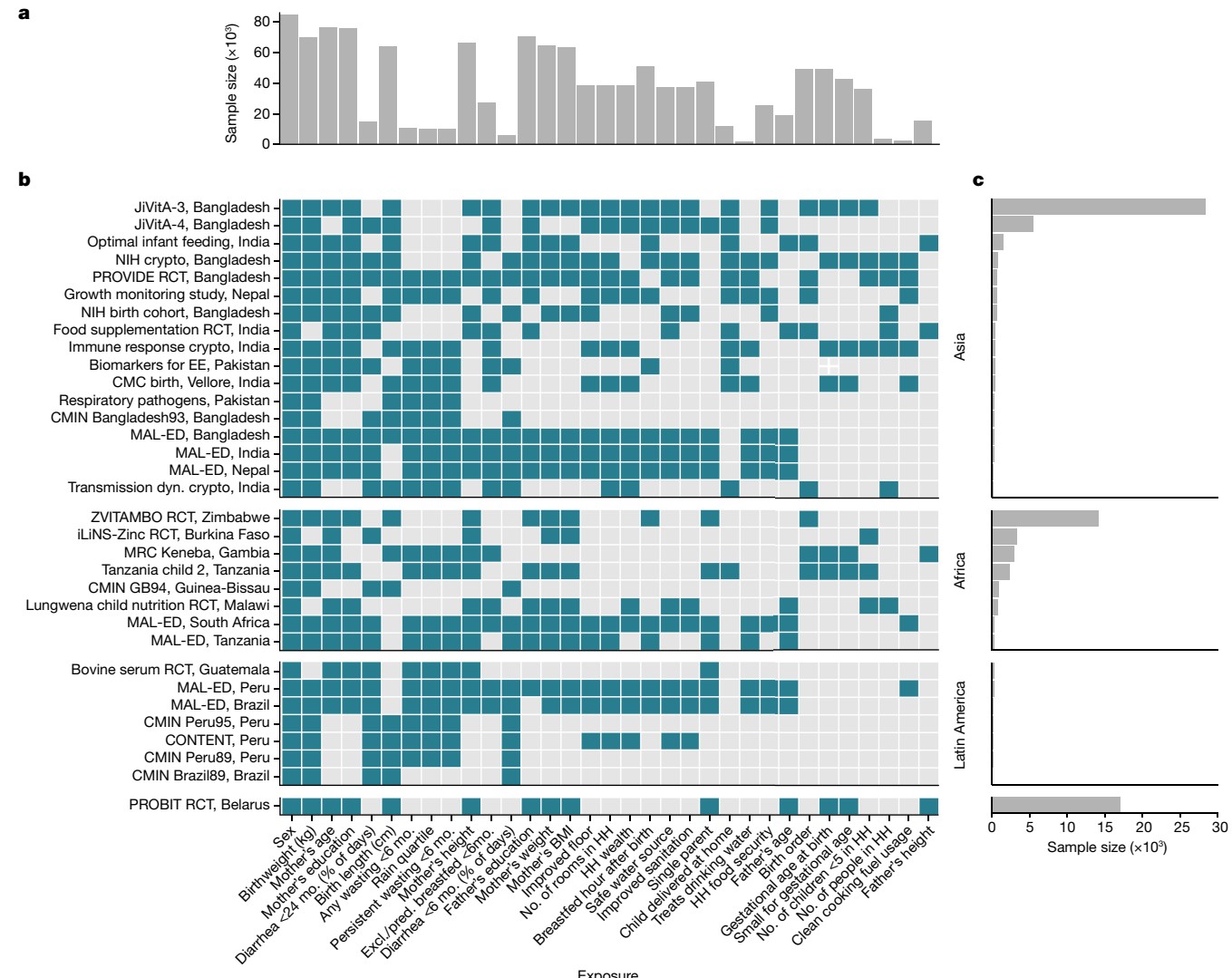

**Fig. 1 | Cohort sample sizes and measured exposures. a**, The total number of children with each measured exposure, sorted from left to right by the number of cohorts measuring the exposure. **b**, The presence of 30 exposure variables in the ki data by within each included cohort. Cohorts are sorted by geographic region and sample size. Details of the cohorts are provided in Extended Data Table 1. CMC, Christian Medical College; Crypto, Cryptosporidium; dyn., dynamics; EE, Environmental Enteropathy; Excl., exclusively; HH, household; NIH, National Institute of Health; mo., months; pred., predominantly; RCT, randomized controlled trial. **c**, The number of child anthropometry observations contributed by each cohort.

## Pooled longitudinal analyses

Here we report a pooled analysis of 33 longitudinal cohorts in 15 LMICs in south Asia, sub-Saharan Africa, Latin America and eastern Europe, in which data collection was initiated between 1987 and 2014. Our objective was to estimate relationships between child, parental and household characteristics and measures of child anthropometry, including length-for-age z-score (LAZ), weight-for-length z-score (WLZ), weight-for-age z-score (WAZ), stunting, wasting, underweight and length and weight velocities from birth to 24 months of age. The estimation of growth faltering outcomes is detailed in the accompanying Articles[20,21]. We also estimated associations between early growth faltering and more severe growth faltering or mortality by 24 months of age.

Cohorts were assembled as part of the Bill & Melinda Gates Foundation's Knowledge Integration (ki) initiative, which included studies of growth and development during the first 1,000 days of life, beginning at conception. We selected longitudinal cohorts from the database that met 5 inclusion criteria: (1) they were conducted in LMICs; (2) they enroled children between birth and 24 months of age and measured their length and weight repeatedly over time; (3) they did not restrict enrolment to

acutely ill children; (4) they enroled children with a median birth year after 1990; and (5) they collected anthropometric status measurements at least quarterly. These inclusion criteria ensured that we could rigorously evaluate the timing and onset of growth faltering among children who were broadly representative of populations in LMICs. Thirty-three cohorts from 15 countries met the inclusion criteria, and 83,671 children and 592,030 total measurements were included in the analysis (Fig. 1). Child mortality was rare and was not reported in many of the ki datasets, so we relaxed inclusion criteria for studies used in the mortality analysis to include studies that measured children at least twice a year. Four additional cohorts met these inclusion criterion, and 14,317 children and 70,733 additional measurements were included in mortality analyses (97,988 total children, 662,763 total observations; Extended Data Table 1). The cohorts were distributed throughout south Asia, Africa and Latin America, with a single European cohort from Belarus.

## Population intervention effects

In a series of analyses, we estimated population intervention effects (PIEs) on growth faltering, the estimated change in population mean

z-score if all individuals in the population had their exposure shifted from observed levels to the lowest-risk reference level[22]. The PIE is a policy-relevant parameter; it estimates the improvement in outcome that could be achievable through intervention for modifiable exposures, as it is a function of the degree of difference between the unexposed and the exposed in children's anthropometry z-scores, as well as the observed distribution of exposure within the population. We selected exposures that were measured in multiple cohorts, could be harmonized across cohorts for pooled analyses, and had been identified as important predictors of stunting or wasting in prior literature (Fig. 1 and Extended Data Tables 2 and 3). Exposure measurement varied by cohort, but all estimates were adjusted for all other measured exposures that we assumed were not on the causal pathway between the exposure of interest and the outcome. For example, the association between maternal height and stunting was not adjusted for child birth weight, because low maternal height could increase stunting risk through lower child birth weight[5]. Parameters were estimated using targeted maximum-likelihood estimation, a doubly robust, semi-parametric method that enables valid inference while adjusting for potential confounders using ensemble machine learning[18,23] (Methods). We estimated cohort-specific parameters, adjusting for measured covariates within each cohort, and then pooled estimates across cohorts using random-effects models[24] (Extended Data Fig. 1). As the reference exposure for PIEs, we used the lowest risk level across cohorts. We also estimated the effects of optimal dynamic interventions, where each child's individual low-risk level of exposure was estimated from potential confounders (Methods). The timing of exposures varied from parental and household characteristics present before birth, to fetal, at-birth or postnatal exposures. We estimated associations with growth faltering that occurred after exposure measurements to ensure temporal ordering of exposures and outcomes.

Population-level improvements in maternal height and child birth size would be expected to improve child LAZ and WLZ at 24 months of age substantially, owing to the high prevalence of suboptimal anthropometry in the populations and their strong association with attained growth at 24 months of age (Figs. 2 and 3). Beyond anthropometry, key predictors of higher z-scores included markers of better household socioeconomic status (for example, the number of rooms in the home, parental education, clean cooking fuel use and household wealth index). The pooled, cross-validated $R^2$ for models that included the top-10 determinants for each z-score plus child sex was 0.25 for LAZ (n = 20 cohorts, 25,647 children) and 0.07 for WLZ (n = 18 cohorts, 17,853 children). The population-level effect of season on WLZ was large, with higher WLZ in drier periods (Fig. 3), consistent with seasonal differences[21]. Exclusive or predominant breastfeeding before 6 months of age was associated with higher WLZ but not LAZ at 6 months of age and was not a major predictor of z-scores at 24 months of age[25] (Extended Data Figs. 2–4). Girls had consistently higher LAZ and WLZ than boys, potentially resulting from sex-specific differences in immunology, nutritional demands, care practices and intrauterine growth[26].

These findings underscore the importance of prenatal exposures for child growth outcomes, and it may remain difficult to reduce the incidence growth faltering at the population level without broad improvements in living standards[7,27]. Maternal anthropometric status can influence child z-scores by affecting fetal growth and birth weight[28,29]. Maternal height and body mass index (BMI) could directly affect postnatal growth through breastmilk quality or could reflect family poverty, genetics, undernutrition, food insecurity or family lifestyle and diet[30,31]. In a secondary analysis, we estimated the associations between parental anthropometry and child z-scores, controlling for birth characteristics, and found that the associations were only partially mediated by birth size, order, hospital delivery and gestational age at birth, with adjusted z-score differences attenuated by a median of 30% (Extended Data Fig. 5).

The strongest predictors of stunting and wasting estimated through population-attributable fractions closely matched those identified for child LAZ and WLZ at 24 months of age (Extended Data Figs. 6 and 7), suggesting that information embedded in continuous and binary measures of child growth provide similar inferences with respect to identifying causes relevant to public health. Potential improvements through population interventions were relatively modest. For example, if all children were born to mothers with higher BMI (20 or more) compared with the observed distribution of maternal BMI—one of the largest predictors of wasting—we estimate that the incidence of wasting by 24 months of age would be reduced by 8.2% (95% confidence interval: 4.4, 12.0; Extended Data Fig. 7). Patterns in associations across growth outcomes were broadly consistent except for preterm birth, which had a stronger association with stunting outcomes than wasting outcomes, and rainy season, which showed a strong association with wasting but not with stunting (Extended Data Fig. 2). The direction of associations did not vary across regions; however, we observed variation in the magnitude of associations across regions—notably, male sex showed a weaker association with low LAZ in south Asia (Extended Data Figs. 8 and 9).

## Age-varying effects on growth faltering

We estimated trajectories of mean LAZ and WLZ stratified by maternal height and BMI. We found that maternal height strongly influenced at-birth LAZ, and that LAZ progressed along similar trajectories up to 24 months of age regardless of maternal height (Fig. 4a), with similar but slightly less pronounced differences when stratified by maternal BMI (Fig. 4b). By contrast, children born to taller mothers had similar WLZ at birth and similar WLZ trajectories up to 3 to 4 months of age, when they diverged substantially (Fig. 4a). WLZ trajectory differences were even more pronounced when stratified by maternal BMI (Fig. 4b). These findings illustrate how maternal status strongly influences the point at which child growth trajectories begin, and how growth trajectories subsequently evolve in parallel, appearing to respond similarly to postnatal insults independently of their starting point.

We hypothesized that causes of growth faltering could differ according to the age of growth faltering onset—for example, we expected children who were born preterm would have a higher risk of incident growth faltering immediately after birth, whereas food insecurity might increase the risk in older children, after weaning. For exposures studied in the PIE analyses, we conducted analyses stratified by age of onset and in many cases found age-varying effects (Fig. 4c). For example, most measures of socioeconomic status were associated with incident wasting or stunting only after 6 months of age, and higher birth order reduced risk for growth faltering below 6 months of age, but increased the risk thereafter. First-born babies are born with lower WLZ and catch up rapidly postnatally (Extended Data Fig. 10). This is probably because first-born babies suffer uterine constraint caused by a less developed uterine–placental–vascular supply[32,33], resulting in birth weights being lower by 100–200 g in most of the studied cohorts; weight is generally more compromised than height[34]. The switch from a constrained uterine–placental nutrient supply line to oral nutrition permits the postnatal catch up. Stronger relationships between key socio-demographic characteristics and wasting and stunting as children age probably reflect cumulative factors that result from household conditions, particularly as complementary feeding is initiated and children begin to explore their environment and potentially face higher levels of food insecurity, especially in homes with multiple children[35]. When viewed across multiple definitions of growth faltering, most exposures had stronger associations with severe stunting, severe wasting or persistent wasting (more than 50% of measurements showing WLZ below −2)—rarer but more serious outcomes—than with incidence of any wasting or stunting (Fig. 4d). Additionally, the characteristics that showed strong association with

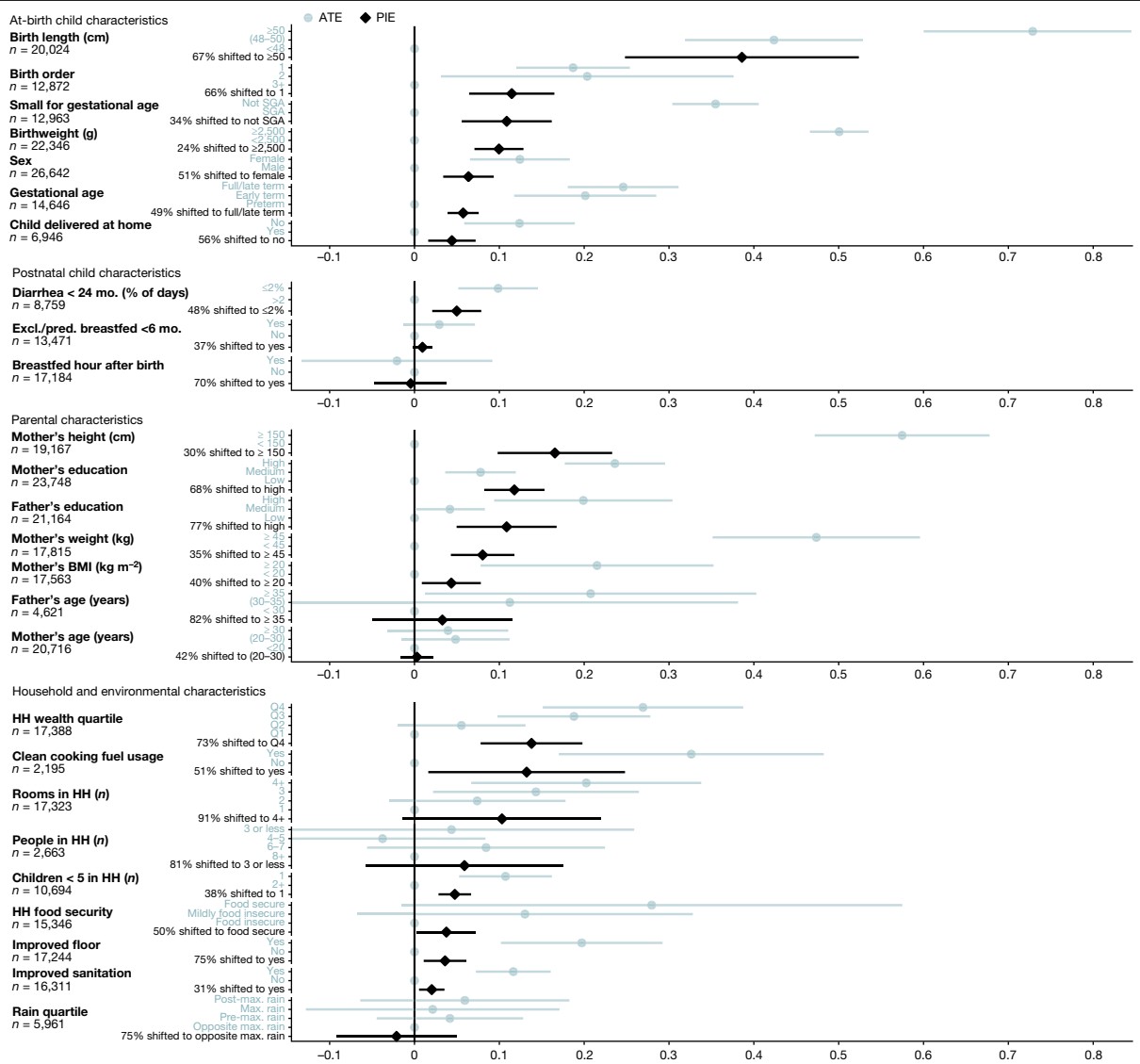

**Fig. 2 | Population intervention effects and mean differences for child, parental, and household exposures on LAZ at 24 months of age.** Adjusted mean differences in average treatment effects (ATEs) (blue) between the labelled higher-risk level of exposures and the reference level (grey dot on the vertical line), and population intervention effects (PIEs) (black), the estimated difference in LAZ after shifting exposure levels for all children to the reference level. The number of children that contributed to each analysis is listed for each exposure. Labels on the *y* axis indicate the level of exposure used to estimate the ATE (blue) or the percentage of the population shifted to the lowest-risk level to estimate the PIE (black). Cohort-specific estimates were adjusted for all measured confounders using ensemble machine learning and targeted maximum-likelihood estimation (TMLE) and then pooled using random effects (Methods). Estimates are shown only for exposures measured in at least four cohorts. Max. maximum; Q, quartile; SGA, small for gestational age.

lower wasting recovery by 90 days of age (birth size, small maternal stature, lower maternal education, later birth order and male sex) increased the risk of wasting prevalence and cumulative incidence (Extended Data Fig. 2).

## Consequences of early growth faltering

In the accompanying Articles, we document high incidence rates of wasting and stunting from birth to six months of age[20,21]. On the basis of previous studies, we hypothesized that early wasting could contribute to subsequent linear growth restriction, and early growth faltering could be consequential for persistent growth faltering and mortality during the first 24 months of life[36–38]. Among cohorts with monthly measurements, we examined age-stratified linear growth velocity by quartiles of WLZ at previous ages. We found a

consistent exposure–response relationship between higher mean WLZ and faster linear growth velocity in the following 3 months (Fig. 5a). Persistent wasting from birth to 6 months of age (defined as less than 50% of measurements wasted) was the wasting exposure that showed the strongest association with incident stunting in older children (Fig. 5b).

We next examined the relationship between measures of growth faltering during the first 6 months and serious growth-related outcomes: persistent wasting from 6–24 months and concurrent wasting and stunting at 18 months of age, both of which put children at high risk of mortality[1,36]. We measured concurrent wasting and stunting at 18 months because stunting prevalence peaked at this age, and because the largest number of measurements across cohorts was for children at 18 months of age[20]. All measures of early growth faltering were significantly associated with later, more serious growth faltering, with

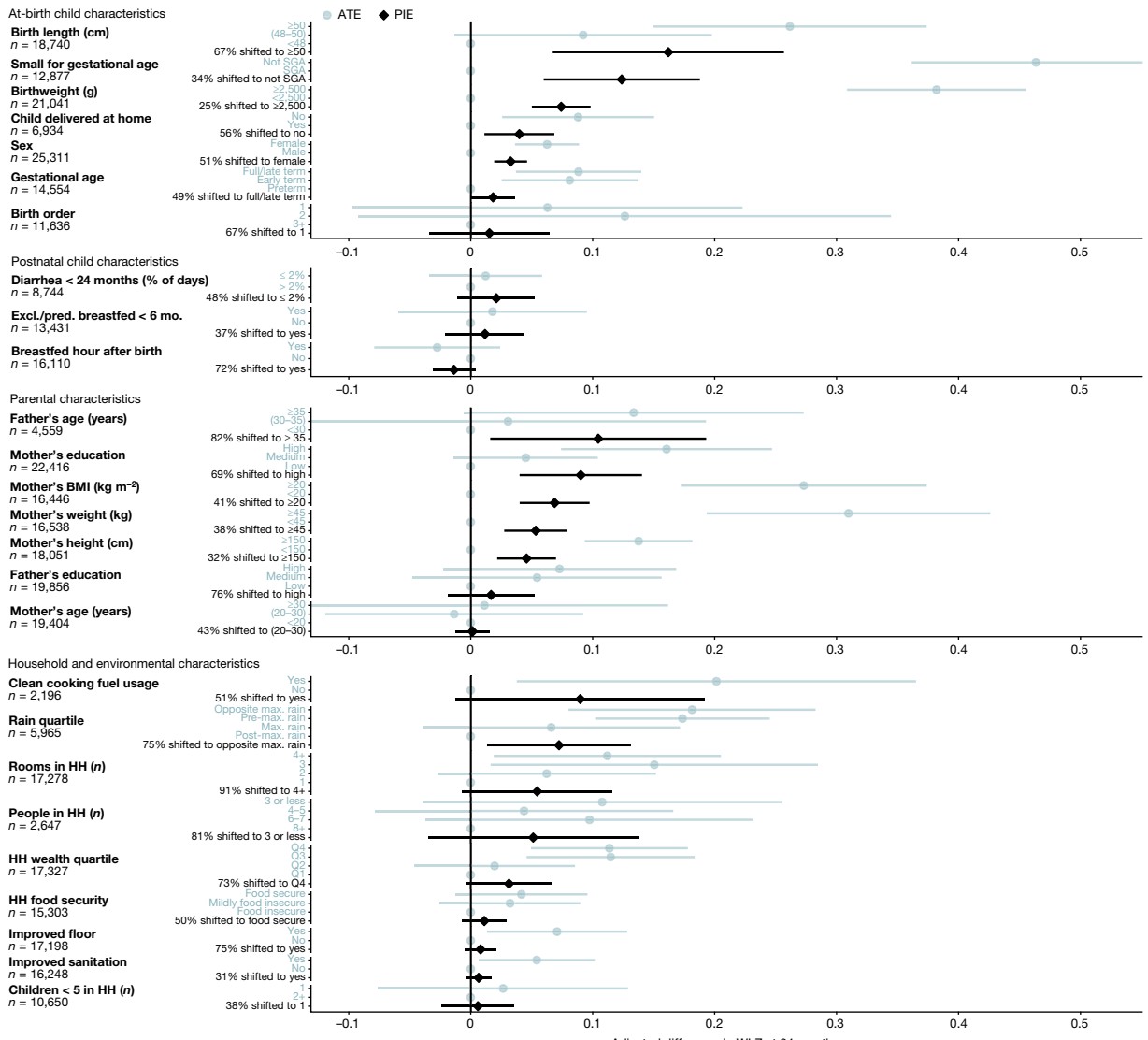

**Fig. 3 | PIEs and mean differences for child, parental and household exposures on WLZ at 24 months of age.** Adjusted mean differences in ATEs (blue) between the labelled higher-risk level of exposures and the reference level (grey dot on vertical line), and PIEs (black), the estimated difference in WLZ after shifting exposure levels for all children to the reference level. The number of children that contributed to each analysis is listed for each exposure.

Labels on the y axis indicate the level of exposure used to estimate the ATE (blue) or the percentage of the population shifted to the lowest-risk level to estimate the PIE (black). Cohort-specific estimates were adjusted for all measured confounders using ensemble machine learning and targeted maximum-likelihood estimation (TMLE) and then pooled using random effects (Methods). Estimates are shown only for exposures measured in at least four cohorts.

measures of ponderal growth faltering being among the strongest predictors (Fig. 5c).

Finally, we estimated hazard ratios of all-cause mortality by 2 years of age associated with measures of growth faltering in 8 cohorts that reported ages of death, which included 1,689 child deaths by 24 months of age (2.4% of children in the 8 cohorts). The included cohorts were highly monitored, and in most cohorts mortality rates were lower than in the general population (Extended Data Table 4). Additionally, the data included only deaths that occurred after anthropometry measurements, so many neonatal deaths may have been excluded, and lacked data on cause-specific mortality, so some deaths may have occurred from causes unrelated to growth faltering. Despite these caveats, growth faltering increased the hazard of death before 24 months for all measures except stunting alone, with the strongest associations observed for severe wasting and stunting (hazard ratio = 8.7, 95% confidence interval: 4.7 to 16.4) and severe underweight alone (hazard ratio = 4.2, 95% confidence interval: 2.0 to 8.6) (Fig. 5d).

## Discussion

This synthesis of cohorts during the first 1,000 days of life from LMICs has provided new insights into the principal causes and near-term consequences of growth faltering. Our use of a semi-parametric method to adjust for potential confounding provided a harmonized approach to estimate PIEs that spanned child-, parent- and household-level exposures with unprecedented breadth (30 exposures) and scale (662,763 anthropometric measurements from 33 cohorts). Our focus on the effects of shifting population-level exposures on continuous measures of growth faltering reflects a growing appreciation that growth faltering is a continuous process[39]. The results show that children in LMICs stand to benefit from interventions to support optimal growth during the first 1,000 days of life. Combining information from high-resolution, longitudinal cohorts enabled us to study critically important outcomes—such as persistent wasting and mortality—that it would not be not possible to study in smaller studies or in cross-sectional data, as well as to examine risk factors by age.

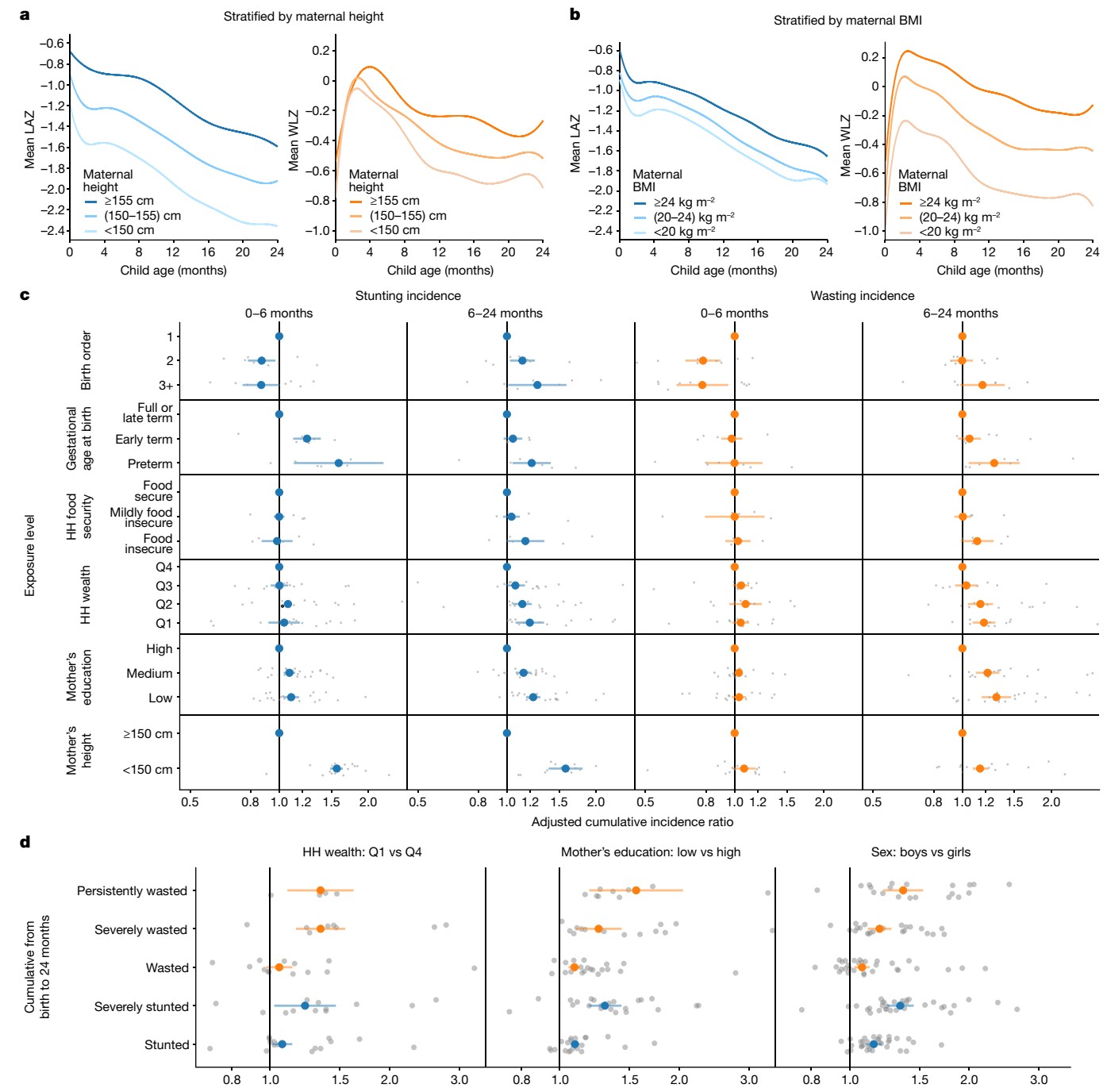

**Fig. 4 | Effect of key exposures on the trajectories, timing and severity of child growth faltering. a**, Child LAZ and WLZ trajectories stratified by maternal height ($n = 413,921$ measurements, 65,061 children, 20 studies). **b**, Child LAZ and WLZ stratified by maternal BMI ($n = 373,382$ measurements, 61,933 children, 17 studies). Growth trajectories stratified by all other examined risk factors are available in Supplementary Note 5. **c**, Associations between key exposures and cumulative wasting incidence, stratified by age of the child during wasting incidence. Grey dots indicate cohort-specific estimates. **d**, Associations between key exposures and growth faltering of different severities. Cumulative incidence ratios compare the highest and lowest-risk categories of each exposure, as indicated above each graph. Grey dots indicate cohort-specific estimates.

Maternal, prenatal and at-birth characteristics were the strongest predictors of growth faltering across regions in LMICs. Our results underscore prenatal exposures as key determinants of child growth faltering[40]. The limited effect of exclusive or predominant breastfeeding during the first 6 months of life (+0.01 LAZ) aligns with a meta-analysis of breastfeeding promotion[25], but our finding of a limited effect of reducing diarrhea during the first 24 months (+0.05 LAZ) contrasts with some observational studies[41,42]. Many predictors such as child sex,

birth order and season are not modifiable but could guide interventions that mitigate their effects, such as seasonally targeted supplementation or enhanced monitoring among boys. Strong associations between maternal anthropometry and early growth faltering highlight the role of intergenerational transfer of growth deficits between mothers and their children[30]. Shifting several key population exposures (maternal height or BMI, education and birth length) to their observed low-risk level would improve LAZ by up to 0.40$z$ and WLZ by up to 0.15$z$ in target

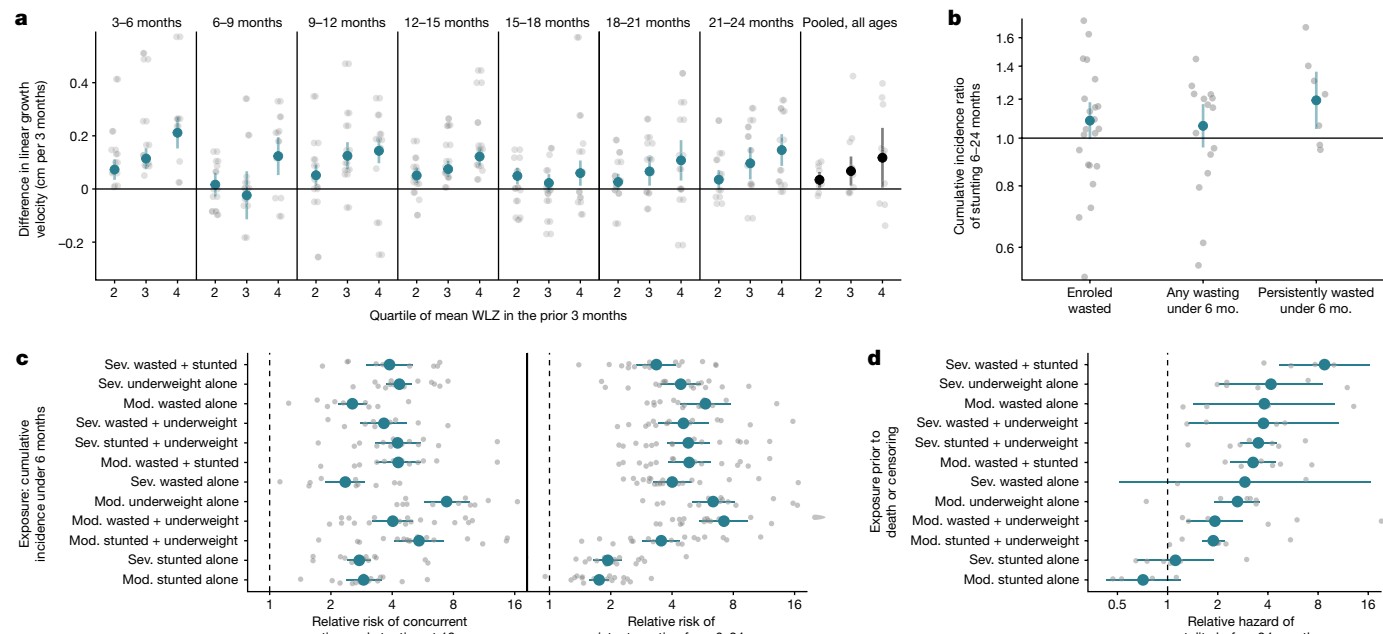

**Fig. 5 | Growth faltering in early life increases risk of more severe growth faltering and mortality. a**, Adjusted differences in linear growth velocity across three-month age bands by quartile of WLZ in the preceding three months. The reference group (horizontal line) comprises children in the first quartile of WLZ in each age stratum. Far right, pooled estimates unstratified by child age. Velocity was calculated from the closest measurements within 14 days of the start and end of the age period. **b**, Relative risk of stunting onset between 6 and 24 months of age among children who experienced measures of early wasting before 6 months of age compared with children who did not. Grey dots indicate cohort-specific estimates. **c**, Association between cumulative incidence of mutually exclusive definitions of growth faltering before 6 months of age and persistent wasting from 6 to 24 months of age (33 cohorts, 6,046 cases and 68,645 children) or concurrent wasting and stunting at 18 months of age (31 cohorts, 1,447 cases, and 22,565 children). The reference group (vertical dashed line) comprises children with no measure of growth failure. Growth faltering definitions are sorted by estimates in **d**. **d**, Hazard ratios between mutually exclusive definitions of growth faltering and mortality before 24 months of age (8 cohorts, 1,689 deaths with known age of death, and 63,812 children). The reference group (vertical dashed line) comprises children with no measure of growth failure. Grey dots indicate cohort-specific estimates. Mod, moderately; sev, severely.

populations and could be expected to prevent 8% to 32% of incident stunting and wasting (Figs. 2 and 3 and Extended Data Figs. 6 and 7). Maternal anthropometric status was highly influential on child birth size, but the parallel drop in postnatal z-scores among children born to different maternal phenotypes was much larger than differences at birth, indicating that growth trajectories were not fully 'programmed' at birth (Fig. 4a,b). This is in accordance with the transition from a placental to oral nutrient supply at birth.

There are caveats to these analyses. The PIEs were based on exposure distributions in the 33 cohorts, which were not necessarily representative of the general population in each setting. The use of external exposure distributions from population-based surveys would be difficult because many key exposures that we considered, such as at-birth characteristics or longitudinal diarrhea prevalence, are not measured in such surveys. In some cases, detailed exposure measurements such as longitudinal breastfeeding or diarrhea history were coarsened to simpler measures to harmonize definitions across cohorts, potentially attenuating their association with growth faltering. Other key exposures such as dietary diversity, nutrient consumption, micronutrient status, maternal and child morbidity indicators, pathogen-specific infections and sub-clinical inflammation and intestinal dysfunction were measured in only a few cohorts, and were therefore not included[43,44]. The absence of these exposures in the analysis, some of which have been found to be important within individual contributed cohorts[44,45], means that our results emphasize exposures that were more commonly collected, but probably exclude some additional causes of growth faltering. A final caveat is that we studied consequences up to 24 months of age—the primary age range of contributed ki cohort studies—and thus did not consider effects on longer-term outcomes. Several studies have suggested that

puberty could be another potential window for intervention to enhance catch-up growth[46]. Improving girls' stature at any point up to the end of puberty could help to reduce intergenerational transfer of growth faltering by increasing maternal height[47], which could in turn improve outcomes among their children (Figs. 2, 3 and 4a,b).

The countries that have shown the greatest reductions in stunting have undergone improvements in maternal education, nutrition and maternal and newborn healthcare and reductions in the number of pregnancies[48], reinforcing the importance of interventions from conception to 1 year of age, when fetal and infant growth velocity is high and energy expenditure for growth and development is about 50% above adult values[49] (adjusted for fat-free mass). A stronger focus on prenatal interventions should not distract from renewed efforts on postnatal prevention. The prenatal and postnatal growth faltering that we observed reinforce the need for sustained support of mothers and children throughout the first 1,000 days of life. Efficacy trials that deliver prenatal nutrition supplements to pregnant women[50–53], therapies to reduce infection and inflammation in pregnant women[54–58] and nutritional supplements to children aged 6–24 months[11,12] have reduced child growth faltering but have fallen short of completely preventing it. Our results suggest that the next generation of preventive interventions should focus on the early period of a child's first 1,000 days—throughout the period from pre-conception to 24 months of age—because maternal status and at-birth characteristics are key determinants of growth faltering during the first 24 months of life. Halting the cycle of growth faltering early should reduce the risk of severe consequences, including mortality, during this formative window of child development. Long-term investments and patience may be required, as it will take decades to eliminate the intergenerational factors that limit maternal height.

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

**The Ki Child Growth Consortium**

**Souheila Abbeddou**[12], **Linda S. Adair**[13], **Tahmeed Ahmed**[14], **Asad Ali**[15], **Hasmot Ali**[16], **Per Ashorn**[17], **Rajiv Bahl**[18], **Mauricio L. Barreto**[19], **Elodie Becquey**[20], **France Bégin**[21], **Pascal Obong Bessong**[22], **Maharaj Kishan Bhan**[23], **Nita Bhandari**[24], **Santosh K. Bhargava**[25], **Zulfiqar A. Bhutta**[26], **Robert E. Black**[27], **Ladaporn Bodhidatta**[28], **Delia Carba**[29], **William Checkley**[27], **Parul Christian**[27,30], **Jean E. Crabtree**[31], **Kathryn G. Dewey**[32], **Christopher P. Duggan**[33], **Caroline H. D. Fall**[34], **Abu Syed Golam Faruque**[14], **Wafaie W. Fawzi**[35], **José Quirino da Silva Filho**[36], **Robert H. Gilman**[27], **Richard L. Guerrant**[37], **Rashidul Haque**[14], **S. M. Tafsir Hasan**[14], **Sonja Y. Hess**[9], **Eric R. Houpt**[37], **Jean H. Humphrey**[36], **Najeeha Talat Iqbal**[38], **Elizabeth Yakes Jimenez**[39,40], **Jacob John**[41], **Sushil Matthew John**[41], **Gagandeep Kang**[42], **Margaret Kosek**[37], **Michael S. Kramer**[43], **Alain Labrique**[44], **Nanette R. Lee**[45], **Aldo Ângelo Moreira Lima**[36], **Tjale Cloupas Mahopo**[46], **Kenneth Maleta**[47], **Dharma S. Manandhar**[48], **Karim P. Manji**[49], **Reynaldo Martorell**[50], **Sarmila Mazumder**[24], **Estomih Mduma**[51], **Venkata Raghava Mohan**[52], **Sophie E. Moore**[53,54], **Robert Ntozini**[55], **Mzwakhe Emanuel Nyathi**[56], **Maribel Paredes Olortegui**[57], **Césaire T. Ouédraogo**[9], **William A. Petri**[37], **Prasanna Samuel Premkumar**[41], **Andrew M. Prentice**[54], **Najeeb Rahman**[15], **Manuel Ramirez-Zea**[58], **Harshpal Singh Sachdev**[59], **Kamran Sadiq**[15], **Rajiv Sarkar**[54], **Monira Sarmin**[14], **Naomi M. Saville**[60], **Saijuddin Shaikh**[16], **Bhim P. Shrestha**[61], **Sanjaya Kumar Shrestha**[62,63], **Alberto Melo Soares**[36], **Bakary Sonko**[54], **Aryeh D. Stein**[64], **Erling Svensen**[65], **Sana Syed**[15,66], **Fayaz Umrani**[15], **Honorine D. Ward**[67], **Keith P. West Jr**[27], **Lee Shu Fune Wu**[27], **Seungmi Yang**[68] & **Pablo Penataro Yori**[37]

[12]Food Safety and Nutrition Unit, Department of Public Health and Primary Care, Ghent University, Ghent, Belgium. [13]University of North Carolina at Chapel Hill, Chapel Hill, NC, USA. [14]International Centre for Diarrhoeal Disease Research, Dhaka, Bangladesh. [15]Aga Khan University, Karachi, Pakistan. [16]JiVitA Project, Johns Hopkins Bangladesh, Bangladesh, Rangpur, Bangladesh. [17]Center for Child, Adolescent and Maternal Health Research, Faculty of Medicine and Health Technology, Tampere University and Tampere University Hospital, Tampere, Finland. [18]World Health Organization, Geneva, Switzerland. [19]Center of Data and Knowledge Integration for Health, Fundação Oswaldo Cruz, Salvador, Brazil. [20]International Food Policy Research Institute, Washington, DC, USA. [21]UNICEF, New York, NY, USA. [22]HIV/AIDS and Global Health Research Programme, University of Venda, Thohoyandou, South Africa. [23]Indian Institute of Technology, New Delhi, India. [24]Centre for Health Research and Development, Society for Applied Studies, New Delhi, India. [25]Sunder Lal Jain Hospital, Delhi, India. [26]Institute for Global Health and Development and Center of Excellence in Women and Child Health, The Aga Khan University, Karachi, Pakistan. [27]Bloomberg School of Public Health, Johns Hopkins University, Baltimore, MD, USA. [28]Armed Forces Research Institute of Medical Sciences, Bangkok, Thailand. [29]USC Office of Population Studies Foundation, University of San Carlos, Cebu, Philippines. [30]Bill & Melinda Gates Foundation, Seattle, WA, USA. [31]Leeds Institute for Medical Research, St James's University Hospital, University of Leeds, Leeds, UK. [32]Institute for Global Nutrition, Department of Nutrition, University of California, Los Angeles, CA, USA. [33]Center for Nutrition, Boston Children's Hospital, Boston, MA, USA. [34]MRC Lifecourse Epidemiology Centre, University of Southampton, Southampton, UK. [35]Department of Global Health and Population, Harvard TH Chan School of Public Health, Cambridge, MA, USA. [36]Federal University of Ceará, Fortaleza, Brazil. [37]University of Virginia, Charlottesville, VA, USA. [38]Department of Pediatrics and Child Health, Aga Khan University, Karachi, Pakistan. [39]Departments of Pediatrics, University of New Mexico Health Sciences Center, Albuquerque, NM, USA. [40]Department of Internal Medicine, University of New Mexico Health Sciences Center, Albuquerque, NM, USA. [41]Christian Medical College, Vellore, India. [42]Translational Health Science and Technology Institute, Faridabad, India. [43]McGill University and McGill University Health Centre, Quebec, Quebec, Canada. [44]Center of Human Nutrition, Department of International Health, Johns Hopkins Bloomberg School of Public Health, Baltimore, MD, USA. [45]Office of Population Studies Foundation, University of San Carlos, Cebu, Philippines. [46]Department of Nutrition, School of Health Sciences, University of Venda, Thohoyandou, South Africa. [47]Department of Public Health, School of Public Health and Family Medicine, College of Medicine, University of Malawi, Zomba, Malawi. [48]Mother and Infant Research Activities, Kathmandu, Nepal. [49]Department of Pediatrics and Child Health, Muhimbili University School of Health and Allied Sciences, Dar es Salaam, Tanzania. [50]Rollins School of Public Health, Emory University, Atlanta, GA, USA. [51]Haydom Lutheran Hospital, Haydom, Tanzania. [52]Community Medicine, Christian Medical College, Vellore, India. [53]Department of Women and Children's Health, Kings College London, London, UK. [54]MRC Unit The Gambia at London School of Hygiene and Tropical Medicine, Banjul, The Gambia. [55]Zvitambo Institute for Maternal and Child Health Research, Harare, Zimbabwe. [56]Department of Animal Sciences, School of Agriculture, University of Venda, Thohoyandou, South Africa. [57]AB PRISMA, Lima, Peru. [58]Research Center for the Prevention of Chronic Diseases, Institute of Nutrition of Central America and Panama, Guatemala City, Guatemala. [59]Sitaram Bhartia Institute of Science and Research, New Delhi, India. [60]Institute for Global Health, University College London, London, UK. [61]Health Research and Development Forum, Kathmandu, Nepal. [62]Walter Reed/AFRIMS Research Unit, Kathmandu, Nepal. [63]Centre for International Health, University of Bergen, Bergen, Norway. [64]Hubert Department of Global Health, Rollins School of Public Health, Emory University, Atlanta, GA, USA. [65]Haukeland University Hospital, Bergen, Norway. [66]Department of Pediatrics, Division of Gastroenterology, Hepatology and Nutrition, University of Virginia School of Medicine, Charlottesville, VA, USA. [67]Tufts Medical Center, Tufts University School of Medicine, Medford, MA, USA. [68]McGill University, Quebec, Canada.

## Methods

### Study designs and inclusion criteria

We included all longitudinal observational studies and randomized trials available through the ki project on 1 April 2018 that met 5 inclusion criteria: (1) they were conducted in LMICs; (2) they enroled children between birth and 24 months of age and measured their length and weight repeatedly over time; (3) they did not restrict enrolment to acutely ill children; (4) they enroled children with a median birth year after 1990; and (5) they collected anthropometric status measurements at least quarterly. We included all children under 24 months of age, assuming months were 30.4167 days, and we considered a child's first measure recorded by 7 days after birth as their anthropometry at birth. Four additional studies with high-quality mortality information that measured children at least every 6 months were included in the mortality analyses (The Burkina Faso Zinc trial, The Vitamin-A trial in India, and the iLiNS-DOSE and iLiNS-DYAD-M trials in Malawi).

### Statistical analysis

Analyses were conducted in R version 4.0.5.

### Outcome definitions

We calculated LAZ, WAZ and WLZ using World Health Organization (WHO) 2006 growth standards[59]. We used the medians of triplicate measurements of heights and weights of children from pre-2006 cohorts to re-calculate $z$-scores to the 2006 standard. We dropped 1,190 (0.2%) unrealistic measurements of LAZ (>+6 or <−6$z$), 1,330 (0.2%) measurements of WAZ (>5 or <−6$z$), and 1,670 (0.3%) measurements of WLZ (>+5 or <−5$z$), consistent with WHO recommendations[60]. Further details on cohort inclusion and assessment of anthropometry measurement quality are provided in the accompanying Article[20]. We also calculated the difference in linear and ponderal growth velocities over three-month periods. We calculated the change in LAZ, WAZ, length in cm and weight in kg within three-month age intervals, including measurements within a two-week window around each age in months to account for variation in the age at each length measurement.

We defined stunting as LAZ <−2, severe stunting as LAZ <−3, underweight as WAZ <−2, severe underweight as WAZ <−3, wasting as WLZ <−2, severe wasting as WLZ <−3, and concurrent stunting and wasting as LAZ <−2 and WLZ <−2. Children with ≥50% of WLZ measurements <−2 and at least 4 measurements over a defined age range were classified as persistently wasted (for example, birth to 24 months, median interval between measurements: 80 days, interquartile range: 62–93). Children were assumed to never recover from stunting episodes, but children were classified as recovered from wasting episodes (and at risk for a new episode of wasting) if their measured WLZ was at or above −2 for at least 60 days (details in the accompanying Article[21]). Stunting reversal was defined as children stunted under 3 months whose final 2 measurements before 24 months were non-stunted. Child mortality was all-cause and was restricted to children who died after birth and before age 24 months. For child morbidity outcomes (Fig. 4c), concurrent wasting and stunting prevalences at 18 months of age were estimated using the anthropometry measurement taken closest to 18 months of age, and within 17–19 months of age, while persistent wasting was estimated from child measurements between 6 and 24 months of age. We chose 18 months to calculate concurrent wasting and stunting because it maximized the number of child observations at later ages when concurrent wasting and stunting was most prevalent, and used ages of 6–24 months to define persistent wasting to maximize the number of anthropometry measurements taken after the early growth faltering exposure measurements[21].

### Estimating relationships between child, parental and household exposures and measures of growth faltering

**Exposure definitions.** We selected the exposures of interest based on variables present in multiple cohorts that met our inclusion criteria, were found to be important predictors of stunting and wasting in prior literature and could be harmonized across cohorts for pooled analyses. Extended Data Tables 2 and 3 list all exposures included in the analysis, as well as exposure categories used across cohorts, and the total number of children in each category. For parental education and asset-based household wealth, we categorized to levels relative to the distribution within each cohort. Continuous biological characteristics (gestational age, birth weight, birth height, parental weight, parental height and parental age) were classified based on a common distribution, pooling data across cohorts. Our rationale was that the meaning of socioeconomic variables is culturally context-dependent, whereas biological variables should have a more universal meaning.

**Risk set definition.** For exposures that occur or exist before birth, we considered the child at risk of incident outcomes at birth. Therefore, we classified children who were born stunted (or wasted) as incident episodes of stunting (or wasting) when estimating the relationship between household characteristics, paternal characteristics, and child characteristics such as gestational age, sex, birth order and birth location.

For postnatal exposures (for example, breastfeeding practices, water, sanitation and hygiene characteristics and birth weight), we excluded episodes of stunting or wasting that occurred at birth. Children who were born wasted could enter the risk set for postnatal exposures if they recovered from wasting during the study period[21]. This restriction ensured that for postnatal exposures, the analysis only included postnatal, incident episodes. Children born or enroled wasted were included in the risk set for the outcome of recovery from wasting within 90 days for all exposures (prenatal and postnatal).

**Estimating differences in outcomes across categories of exposures.** We estimated measures of association between exposures and growth faltering outcomes by comparing outcomes across categories of exposures in four ways:

Mean difference of the comparison levels of the exposure on LAZ, WLZ at birth, 6 months, and 24 months. The $z$-scores used were the measures taken closest to the age of interest and within 1 month of the age of interest, except for $z$-scores at birth which only included a child's first measure recorded by 7 days after birth. We also calculated mean differences in LAZ, WAZ, weight and length velocities.

Prevalence ratios between comparison levels of the exposure, compared to the reference level at birth, 6 months, and 24 months. Prevalence was estimated using anthropometry measurements closest to the age of interest and within one month of the age of interest, except for prevalence at birth which only included measures taken on the day of birth.

Cumulative incidence ratios (CIRs) between comparison levels of the exposure, compared to the reference level, for the incident onset of outcomes between birth and 24 months, 6 and 24 months, and birth and 6 months.

Mean $z$-scores by continuous age, stratified by levels of exposures from birth to 24 months were fit within individual cohorts using cubic splines with the bandwidth chosen to minimize the median Akaike information criterion across cohorts[61]. We estimated splines separately for each exposure category. We pooled spline curves across cohorts into a single prediction, offset by mean $z$-scores at one year, using random-effects models[62].

**Estimating population-attributable parameters.** We estimated three measures of the population-level effect of exposures on growth faltering outcomes:

(1) Population intervention effect (PIE), a generalization of population-attributable risk, was defined as the change in population mean $z$-score if the entire population's exposure was set to an ideal reference level. For each exposure, we chose reference levels

based on prior literature or as the category with the highest mean LAZ or WLZ across cohorts.

(2) Population-attributable fraction (PAF) was defined as the proportional reduction in cumulative incidence if the entire population's exposure was set to an ideal low-risk reference level. We estimated the PAF for the prevalence of stunting and wasting at birth, 6, and 24 months and cumulative incidence of stunting and wasting from birth to 24 months, 6 to 24 months, and from birth to 6 months. For each exposure, we chose the reference level as the category with the lowest risk of stunting or wasting.

(3) Optimal individualized intervention impact. We used a variable importance measure methodology to estimate the impact of an optimal individualized intervention on an exposure[63]. The optimal intervention on an exposure was determined through estimating individualized treatment regimes, which give an individual-specific rule for the lowest-risk level of exposure based on individuals' measured covariates. The covariates used to estimate the low-risk level are the same as those used for the adjustment documented in section 6 below. The impact of the optimal individualized intervention is derived from the variable importance measure, which is the predicted change in the population mean outcome from the observed outcome if every child's exposure was shifted to the optimal level. This differs from the PIE and PAF parameters in that we did not specify the reference level; moreover, the reference level could vary across participants.

PIE and PAF parameters assume a causal relationship between exposure and outcome. For some exposures, we considered attributable effects to have a pragmatic interpretation – they represent a summary estimate of relative importance that combines the exposure's strength of association and its prevalence in the population[64]. Comparisons between optimal intervention estimates and PIE estimates are shown in Extended Data Fig. 11.

## Estimation approach

**Estimation of cohort-specific effects.** For each exposure, we used the directed acyclic graph framework to identify potential confounders from the broader set of exposures used in the analysis[65]. We did not adjust for characteristics that were assumed to be intermediate on the causal path between any exposure and the outcome, because while controlling for mediators may help adjust for unmeasured confounders in some conditions, it can also lead to collider bias[66,67]. Detailed lists of adjustment covariates used for each analysis are available in Supplementary Note 1. Confounders were not measured in every cohort, so there could be residual confounding in cohort-specific estimates.

Analyses used a complete-case approach that only included children with non-missing exposure and outcome measurements. For additional covariates in adjusted analyses, we used the following approach to impute missing covariate values[68]. Within each cohort, if there was <50% missingness in a covariate, we imputed missing measurements as the median (continuous variables) or mode (categorical variables) among all children, and analyses included an indicator variable for missingness in the adjustment set. Covariates with >50% missingness were excluded from the potential adjustment set. When calculating the median for imputation, we used children as independent units rather than measurements so that children with more frequent measurements were not over-represented.

Unadjusted prevalence ratios and CIRs between the reference level of each exposure and comparison levels were estimated using logistic regressions[69]. Unadjusted mean differences for continuous outcomes were estimated using linear regressions.

To flexibly adjust for potential confounders and reduce the risk of model misspecification, we estimated adjusted prevalence ratios, CIRs, and mean differences using TMLE, a two-stage estimation strategy that incorporates state-of-the-art machine learning algorithms (super learner) while still providing valid statistical inference[23,70]. The effects of covariate adjustment on estimates compared to unadjusted estimates is shown in Extended Data Fig. 12, and E-values, summary measures of the strength of unmeasured confounding needed to explain away observed significant associations[71], are plotted in Extended Data Fig. 13. The super learner is an ensemble machine learning method that uses cross-validation to select a weighted combination of predictions from a library of algorithms[72]. We included in the library simple means, generalized linear models, LASSO penalized regressions[73], generalized additive models[74], and gradient boosting machines[75]. The super learner was fit to maximize the tenfold cross-validated area under the receiver operator curve (AUC) for binomial outcomes, and minimize the tenfold cross-validated mean-squared error (MSE) for continuous outcomes. That is, the super learner was fit using nine-tenths of the data, while the AUC/MSE was calculated on the remaining one-tenth of the data. Each fold of the data was held out in turn and the cross-validated performance measure was calculated as the average of the performance measures across the ten folds. This approach is practically appealing and robust in finite samples, since this cross-validation procedure uses unseen sample data to measure the estimator's performance. Also, the super learner is asymptotically optimal in the sense that it is guaranteed to outperform the best possible algorithm included in the library as sample size grows. The initial estimator obtained via super learner is subsequently updated to yield an efficient double-robust semi-parametric substitution estimator of the parameter of interest[23]. To estimate the $R^2$ of models including multiple exposures, we fit super learner models, without the targeted learning step, and within each cohort measuring the exposures. We then pooled cohort-specific $R^2$ estimates using fixed-effects models.

We estimated influence curve-based, clustered standard errors to account for repeated measures in the analyses of recovery from wasting or progression to severe wasting. We assumed that the children were the independent units of analysis unless the original study had a clustered design, in which case the unit of independence in the original study were used as the unit of clustering. We used clusters as the unit of independence for the iLiNS-Zinc, Jivita-3, Jivita-4, Probit, and SAS Complementary Feeding trials. We estimated 95% confidence intervals for incidence using the normal approximation.

Mortality analyses estimated hazard ratios using Cox proportional hazards models with a child's age in days as the timescale, adjusting for potential confounders, with the growth faltering exposure status updated at each follow-up that preceded death or censoring by 24 months of age. Growth faltering exposures included moderate (between $-2z$ and $-3z$) wasting, stunting, and underweight, severe (below $-3z$) wasting, stunting, and underweight, and combinations of concurrent wasting, stunting, and underweight. Growth faltering categories were mutually exclusive within moderate or severe classifications, so children were classified as only wasted, only stunted, or only underweight, or some combination of these categories. We estimated the hazard ratio associated with different anthropometric measures of child growth failure in separate analyses, considering each as an exposure in turn with the reference group defined as children without the deficit. For children who did not die, we defined their censoring date as the administrative end of follow-up in their cohort, or age 24 months (730 days), whichever occurred first. Because mortality was a rare outcome, estimates are adjusted only for child sex and trial treatment arm. To avoid reverse causality, we did not include child growth measures occurring within 7 days of death. Extended Data Table 4 lists the cohorts used in the mortality analysis, the number of deaths in each cohort, and a comparison to country-level infant mortality rates.

**Data sparsity.** We did not estimate relative risks between a higher level of exposure and the reference group if there were 5 or fewer cases in either stratum. In such cases, we still estimated relative risks between other exposure strata and the reference strata if those

strata were not sparse. For rare outcomes, we only included one covariate for every 10 observations in the sparsest combination of the exposure and outcome, choosing covariates based on ranked deviance ratios.

## Pooling parameters

We pooled adjusted estimates from individual cohorts using random-effects models, fit using restricted maximum-likelihood estimation. The pooling methods are detailed in the accompanying Article[20]. All parameters were pooled directly using the cohort-specific estimates of the same parameter, except for population-attributable fractions. Pooled PAFs were calculated from random-effects pooled population intervention effects (PIEs), and pooled outcome prevalence in the population using the following formulas[76]:

$$PAF = \frac{PIE}{Outcome\ prevalence} \times 100 \qquad (1)$$

$$PAF\ 95\%\ confidence\ interval = \frac{PIE\ 95\%\ confidence\ interval}{Outcome\ prevalence} \times 100 \qquad (2)$$

For PAFs of exposures on the cumulative incidence of wasting and stunting, the pooled cumulative incidence was substituted for the outcome prevalence in the above equations. We used this method instead of direct pooling of PAFs because unlike PAFs, PIEs are unbounded with symmetric confidence intervals.

For Fig. 4a,b, mean trajectories estimated using cubic splines in individual studies and then curves were pooled using random effects[62]. Curves estimated from all anthropometry measurements of children taken from birth to 24 months of age within studies that measured the measure of maternal anthropometry.

## Sensitivity analyses

We examined covariate missingness by study and assessed the effect of covariate missingness by comparing results with median/mode missingness imputation to a complete-case analysis (Supplementary Note 2). We compared estimates pooled using random-effects models, which are more conservative in the presence of heterogeneity across studies, with estimates pooled using fixed effects (Supplementary Note 3), and we compared adjusted estimates with estimates unadjusted for potential confounders (Supplementary Note 4). We also plotted splines of child growth trajectories, stratified by exposure levels, for all exposures in Supplementary Note 5. We re-estimated the attributable differences of exposures on WLZ and LAZ at 24 months, dropping the PROBIT trial, the only European study (Supplementary Note 6). Point estimates and confidence intervals from all age, exposure and growth outcome combinations (as presented in Extended Data Fig. 2) are plotted in Supplementary Note 7.

## Inclusion and ethics

This study analysed data that was collected in 15 LMICs that were assembled by the Bill & Melinda Gates Foundation Ki initiative. The datasets are owned by the original investigators that collected the data. Members of the Ki Child Growth Consortium were nominated by each study's leadership team to be representative of the country and study teams that originally collected the data. Consortium members reviewed their cohort's data within the i database to ensure external and internal consistency of cohort-level estimates. Consortium members provided significant input on the statistical analysis plan, interpretation of results and manuscript writing. Per the request of consortium members, the manuscript includes cohort-level and regional results to maximize the utility of the study findings for local investigators and public health agencies. Analysis code has been published with the manuscript to promote transparency and extensions of our research by local and global investigators.

## Reporting summary

Further information on research design is available in the Nature Portfolio Reporting Summary linked to this article.

## Data availability

The data that support the findings of this analysis are a combination of data from multiple principal investigators and institutions. The data are available, upon reasonable request, to the requestor by contacting the individual principal investigators. The individuals and the contact information to help the requestor obtain access to the data are listed at https://www.synapse.org/#!Synapse:syn51570682/wiki/. The analysis dataset is at https://www.synapse.org/#!Synapse:syn51570682/datasets/. This dataset is access controlled and not available publicly for privacy reasons.

## Code availability

Code used in the study has been deposited at Zenodo: https://zenodo.org/record/7937811[77].

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

**Acknowledgements** This research was financially supported by a global development grant (OPP1165144) from the Bill & Melinda Gates Foundation to the University of California, Berkeley, CA, USA. J.B.-C. acknowledges funding from the National Institute of Allergy and Infectious Diseases under Award K01AI141616. J.B.-C. is a Chan Zuckerberg Biohub investigator. The authors thank the following collaborators on the included cohorts and trials for their contributions to study planning, data collection, and analysis: M. Sharif, S. Kerio, Urosa, Alveen, S. Hussain, V. Paudel, A. Costello, N. Rouamba, J.-B. Ouédraogo, L. Prince, S. A. Vosti, B. Torun, L. M. Locks, C. M. McDonald, R. Kupka, R. J. Bosch, R. Kisenge, S. Aboud, M. Wang, Azaduzzaman, A. A. Shamim, R. Haque, R. Klemm, S. Mehra, M. Mitra, K. Schulze, S. Taneja, B. Nayyar, V. Suri, P. Khokhar, B. Nayyar, P. Khokhar, J. E. Rohde, T. Kumar, J. Martines, M. K. Bhan and all other members of the study staff and field teams; all study participants and their families for their important contributions; the LCNI5 and iLiNS research teams, participants and people of Lungwena, Namwera, Mangochi and Malindi; and our research assistants for their positive attitude, support and help in all stages of the studies.

**Author contributions** Conceptualization: A.M., J.B.-C., J.M.C., K.H.B., P.C. and B.F.A. Funding acquisition: J.M.C., A.E.H., M.J.v.d.L. and B.F.A. Data curation: A.M., J.B.-C., J.C., O.S., W. Cai, A.N., N.N.P., W.J., E.J., E.O.C., S.R., N.H., I. Malenica, H.L., R. Hafen, V.S., J.H. and T.N. Formal analyses: A.M., J.B.-C., J.C., O.S., W. Cai, A.N., N.N.P., W.J., E.J., E.O.C., S.R. S.D., N.H., I. Malenica, H.L., V.S. and B.F.A. Methodology: A.M., J.B.-C., J.M.C., J.C., O.S., N.H., I. Malenica, A.E.H., M.J.v.d.L., K.H.B., P.C. and B.F.A. Visualization: A.M., J.B.-C., A.N., N.N.P., S.R., A.S., E.J., J.C., R. Hafen, S.D. K.H.B., P.C. and B.F.A. Writing, original draft preparation: A.M., J.B.-C. and B.F.A. Writing, review and editing: A.M., J.B.-C., J.M.C., K.H.B., P.C., B.F.A. and all other authors.

**Competing interests** T. Norman is an employee of the Bill & Melinda Gates Foundation. K.H.B. and P. Christian are former employees of the Bill & Melinda Gates Foundation. J.C., V.S., R. Hafen and J.H. are research contractors funded by the Bill & Melinda Gates Foundation.

**Additional information**
**Correspondence and requests for materials** should be addressed to Andrew Mertens or Benjamin F. Arnold.

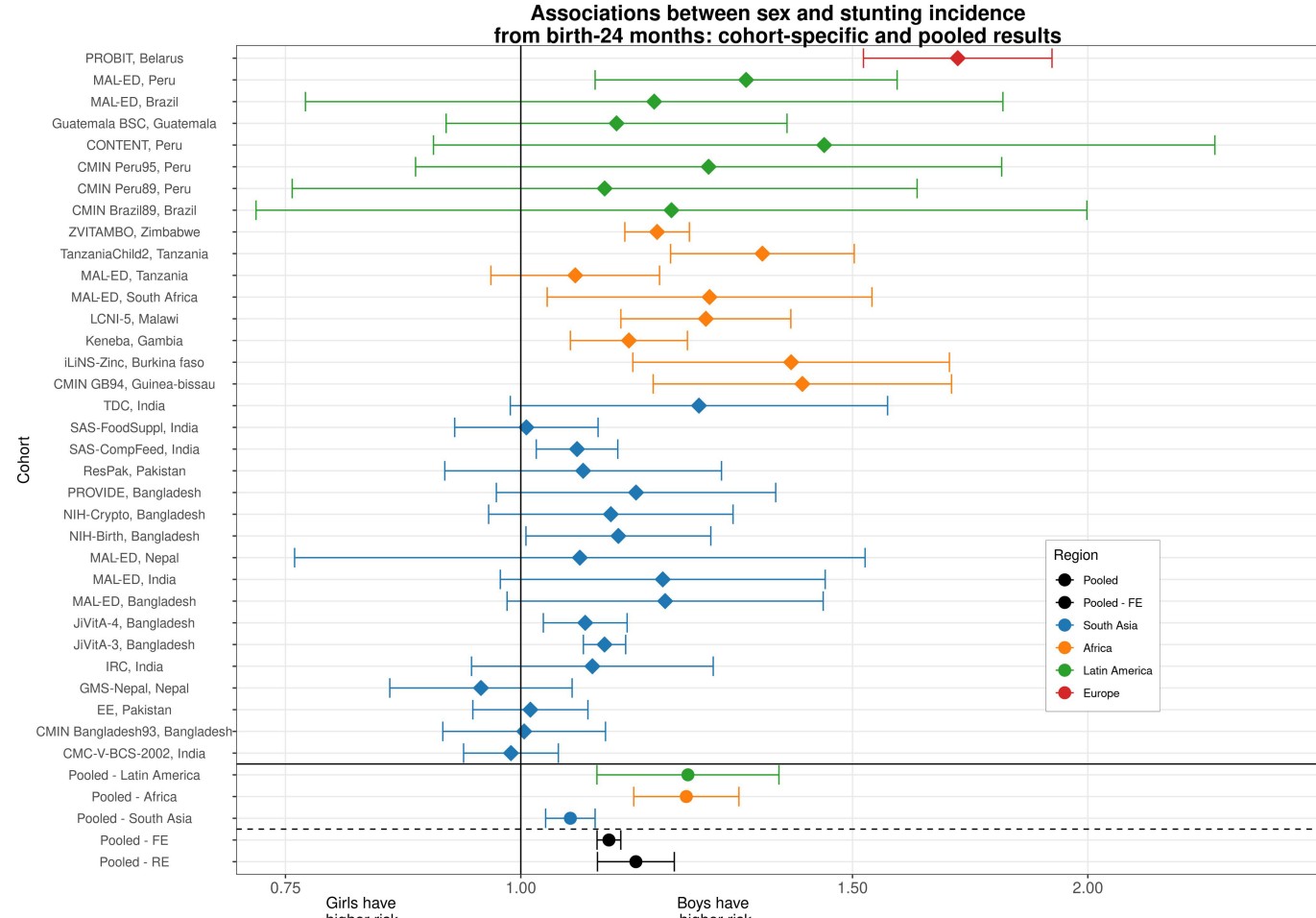

**Extended Data Fig. 1 | Example forest plot of cohort-specific and pooled parameter estimates.** Cohort-specific estimates of the cumulative incidence ratio of stunting are plotted on each row, comparing the risk of any stunting from birth to 24 months among boys compared to a reference level of girls. Below the solid horizontal line are region-specific pooled measures of association, pooled using random-effects models. Below the dashed line are overall pooled measures of association, comparing pooling using random or fixed effects models. The primary results reported throughout the manuscript are overall (not region stratified) estimates pooled using random effects models.

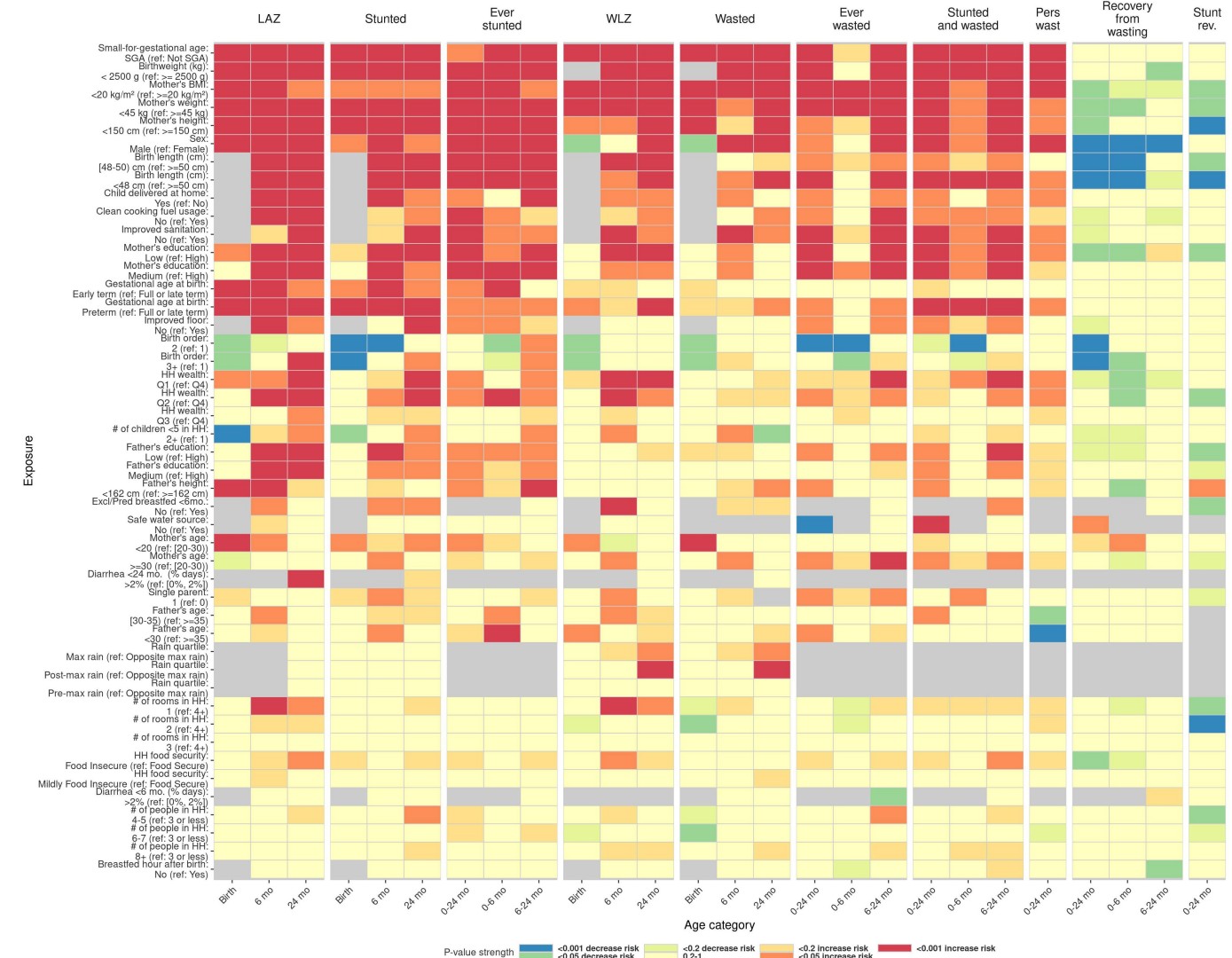

**Extended Data Fig. 2 | Heatmap of significance and direction across exposure-outcome combinations.** The heatmap shows the significance and direction of estimates through the cell colors, separated across primary outcomes by child age. Red and orange cells are exposures where the outcome is estimated have an increased probability of occurring compared to the reference level (harmful exposures except for recovery outcomes), while blue and green cells are exposures associated with a decreased probability of the outcome (protective exposures except for recovery outcomes). The outcomes are labeled at the top of the columns, with each set of three columns the set of three ages analyzed for that outcome. Each row is a level of an exposure variable, with reference levels excluded. Rows are sorted top to bottom by increasing average p-value. Grey cells denote comparisons that were not estimated or could not be estimated because of data sparsity in the exposure-outcome combination. All point estimates and confidence intervals for exposure-outcome pairs with P-values plotted in this figure are viewable online in Supplimentary Note 7.

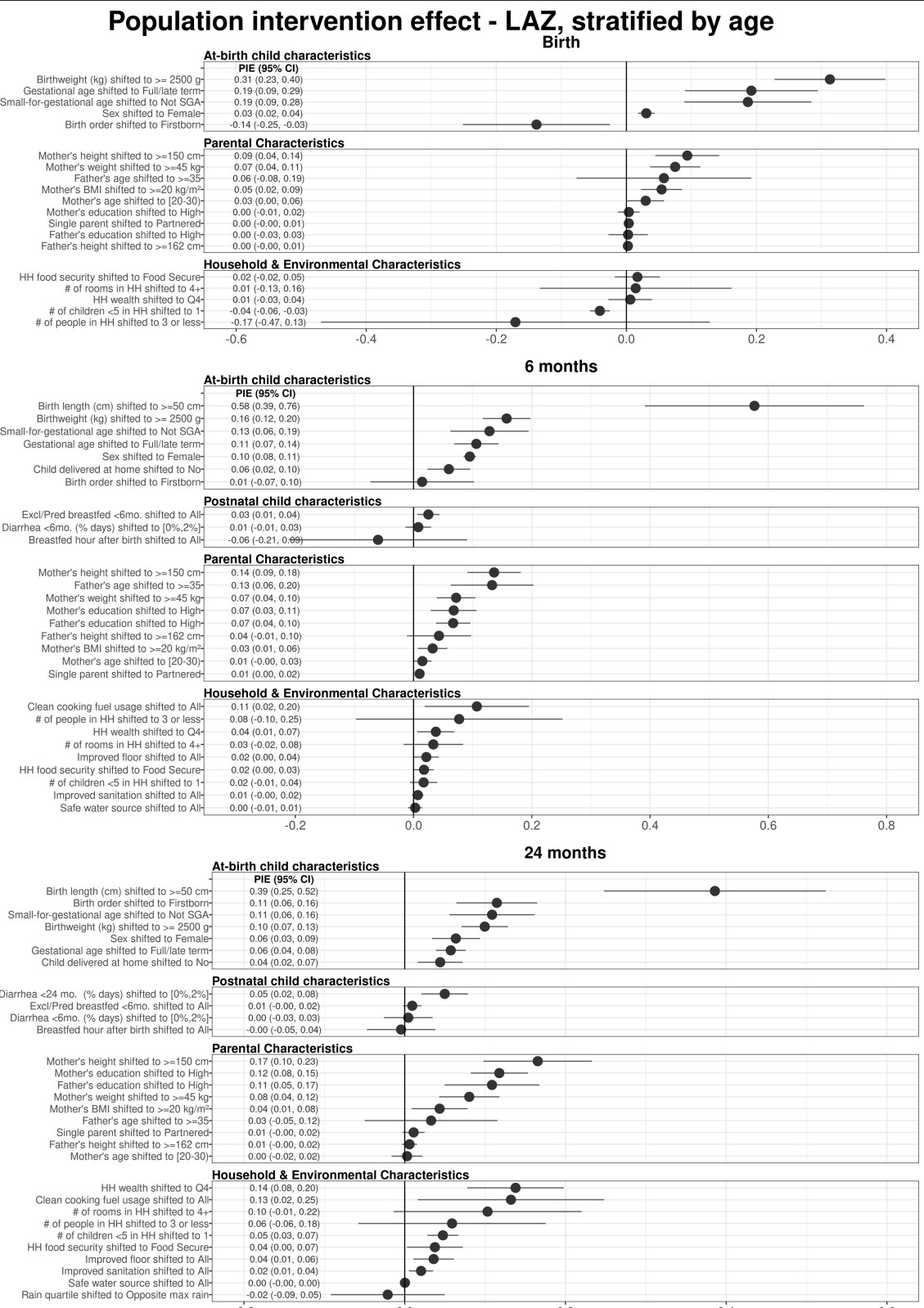

**Extended Data Fig. 3 | Age-stratified population intervention effects in length-for-age Z-scores.** Exposures, rank ordered by population intervention effect on child LAZ, stratified by the age of the child at the time of anthropometry measurement. The population intervention effect is the expected difference in mean Z-score if all children had the reference level of the exposure rather than the observed exposure distribution. Reference levels are printed in the exposure label. Cohort-specific estimates were adjusted for all measured confounders using ensemble machine learning and TMLE, and then pooled using random effects (Methods). Estimates are shown only for exposures measured in at least 4 cohorts.

# Population intervention effect - WLZ, stratified by age

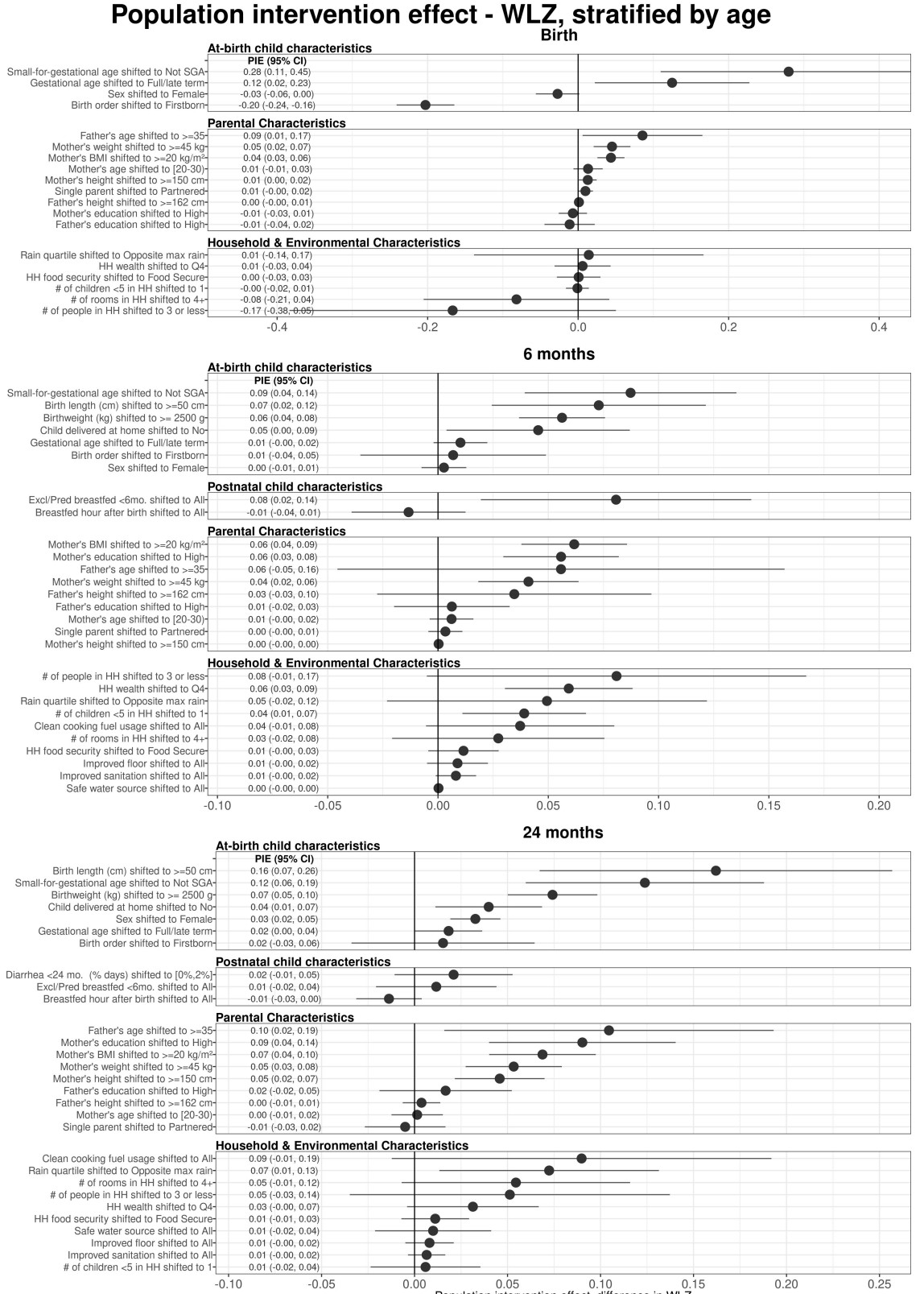

**Extended Data Fig. 4 | Age-stratified population intervention effects in weight-for-length Z-scores.** Exposures, rank ordered by population intervention effects on child WLZ, stratified by the age of the child at the time of anthropometry measurement. The population intervention effect is the expected difference in population mean Z-score if all children had the reference level of the exposure rather than the observed distribution. For all plots, reference levels are printed next to the name of the exposure. Cohort-specific estimates were adjusted for all measured confounders using ensemble machine learning and TMLE, and then pooled using random effects (Methods). Estimates are shown only for exposures measured in at least 4 cohorts.

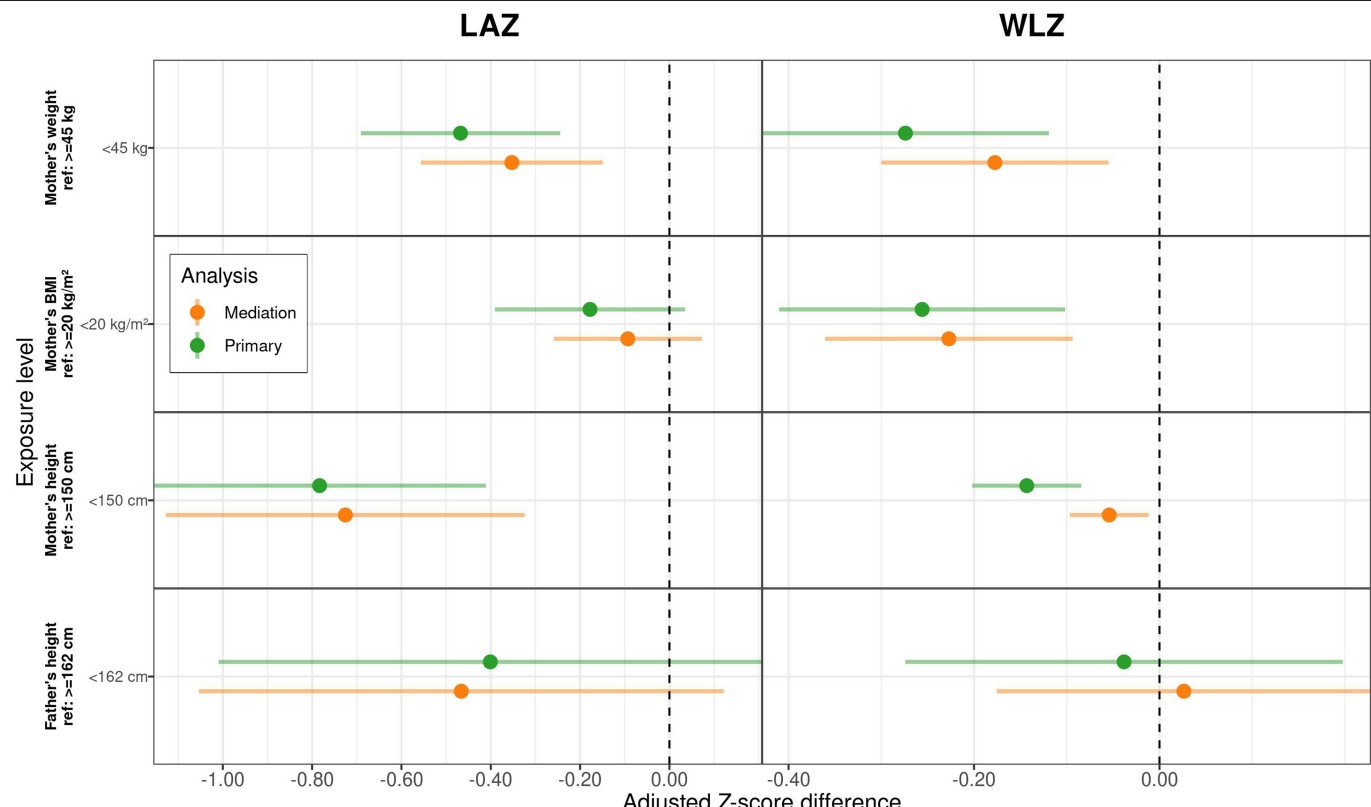

**Extended Data Fig. 5 | Mediation of parental anthropometry effects by birth size on child Z-scores at 24 months.** Mediating effect of adjusting for birth anthropometry and at-birth characteristics on the estimated Z-score differences between levels of parental anthropometry. Primary estimates were adjusted for all other measured exposures not on the causal pathway, while the mediation analysis estimates were additionally adjusted for birthweight, birth length, gestational age at birth, birth order, small-for-gestational age status, and home vs. hospital delivery. Only estimates from cohorts measuring at least 3 of the 6 at-birth characteristics were used to estimate the pooled Z-score differences (n = 6 cohorts, 17,124 observations). Mediation estimates were slightly attenuated toward the null, and only in the case of maternal height and child WLZ were they statistically different from the primary analysis. These results imply that the causal pathway between parental anthropometry and growth faltering operates through its effect on birth size, but most of the effect is through other pathways.

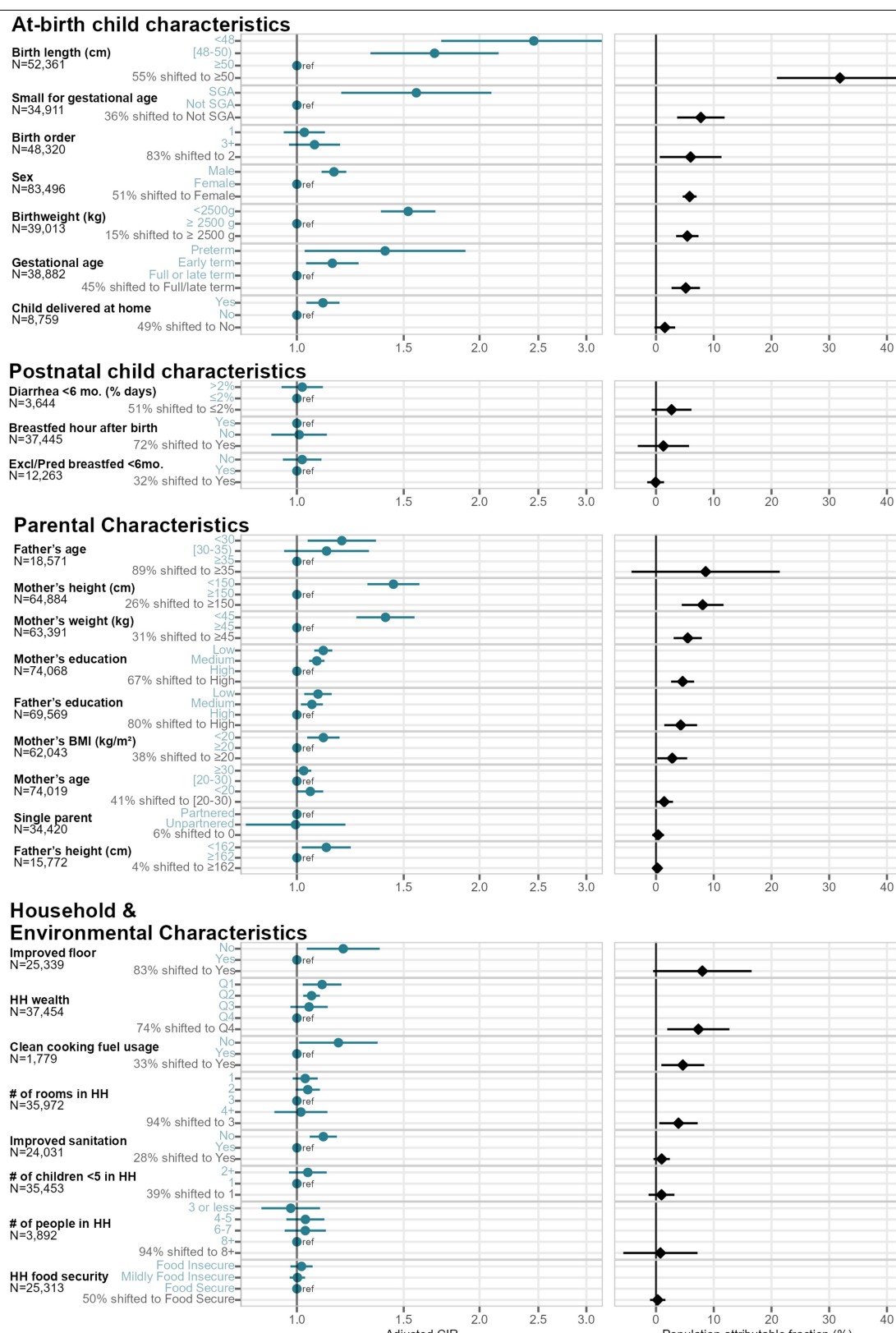

**Extended Data Fig. 6** | See next page for caption.

**Extended Data Fig. 6 | Rank-ordered associations between child, parental, and household characteristics and adjusted relative risks or population attributable fractions of stunting by age 24 months.** Blue points in the left panel show adjusted cumulative incidence ratios (CIRs) between higher-risk exposure levels and reference levels, and black points in the right panel show population attributable fractions (PAFs), the estimated proportion of the risk in the whole population that would be removed if the exposure were set to its indicated reference level. The number of children that contributed to each analysis is listed by exposure. The colored Y-axis label is either the level of exposure contrasted against the reference level to estimate the CIR, or the percent of the population shifted to the lowest-risk level to estimate the PAF. For at-birth exposures, at-birth stunting and wasting were excluded to focus on incidence of new (postnatal) cases, and for postnatal exposures (breastfeeding practice and diarrheal disease), the cumulative incidence of stunting from 6–24 months was used. Cohort-specific estimates were adjusted for all measured confounders using ensemble machine learning and TMLE, and then pooled using random effects (Methods). Estimates are shown only for exposures measured in at least 4 studies.

# At-birth child characteristics

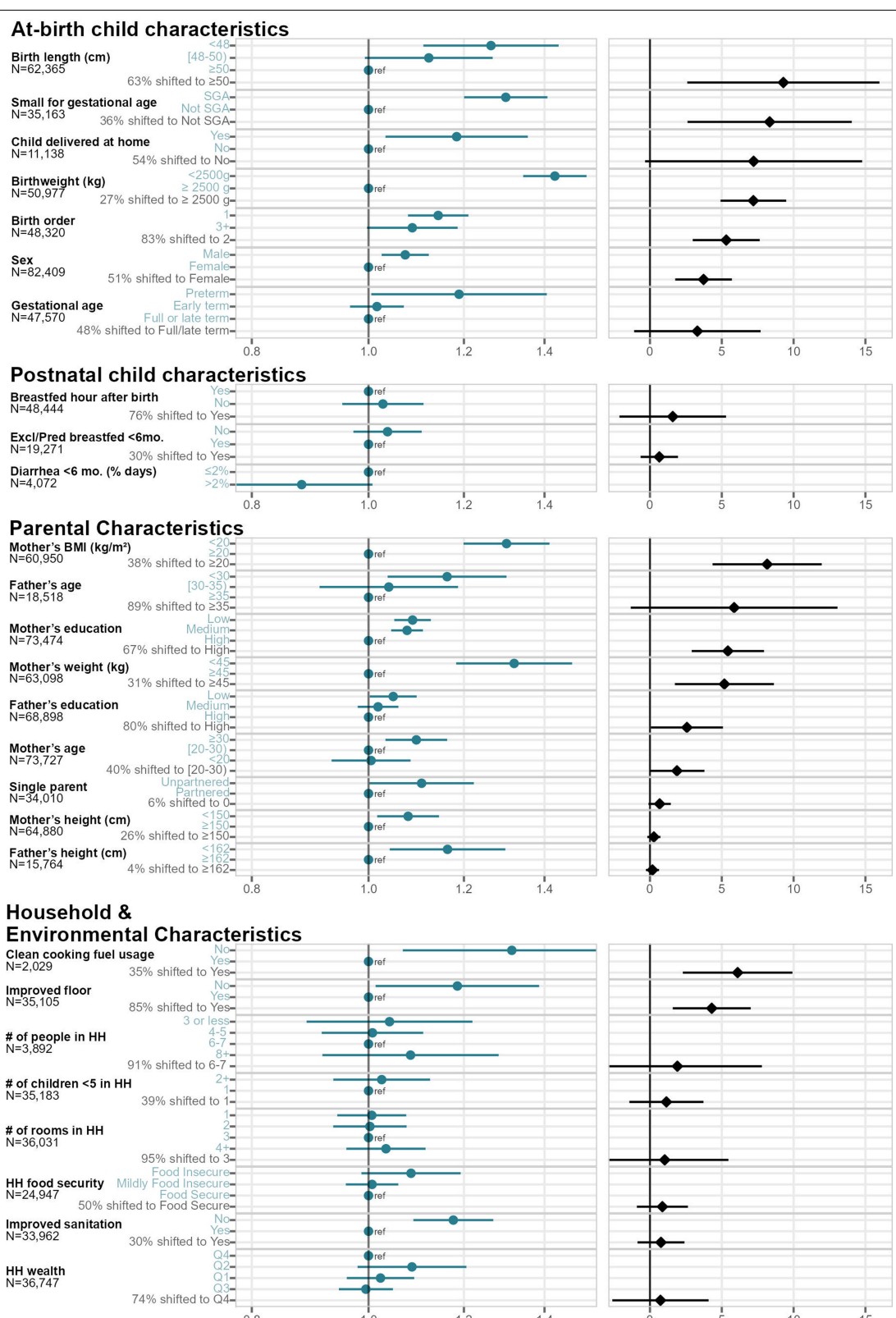

**Birth length (cm)**
N=62,365
<48
[48-50]
≥50 ref
63% shifted to ≥50

**Small for gestational age**
N=35,163
SGA
Not SGA ref
36% shifted to Not SGA

**Child delivered at home**
N=11,138
Yes
No ref
54% shifted to No

**Birthweight (kg)**
N=50,977
<2500g
≥ 2500 g ref
27% shifted to ≥ 2500 g

**Birth order**
N=48,320
1
3+
83% shifted to 2

**Sex**
N=82,409
Male
Female ref
51% shifted to Female

**Gestational age**
N=47,570
Preterm
Early term
Full or late term ref
48% shifted to Full/late term

# Postnatal child characteristics

**Breastfed hour after birth**
N=48,444
Yes ref
No
76% shifted to Yes

**Excl/Pred breastfed <6mo.**
N=19,271
No
Yes ref
30% shifted to Yes

**Diarrhea <6 mo. (% days)**
N=4,072
≤2% ref
>2%

# Parental Characteristics

**Mother's BMI (kg/m²)**
N=60,950
<20
≥20 ref
38% shifted to ≥20

**Father's age**
N=18,518
<30
[30-35]
≥35 ref
89% shifted to ≥35

**Mother's education**
N=73,474
Low
Medium
High ref
67% shifted to High

**Mother's weight (kg)**
N=63,098
<45
≥45 ref
31% shifted to ≥45

**Father's education**
N=68,898
Low
Medium
High ref
80% shifted to High

**Mother's age**
N=73,727
≥30
[20-30] ref
<20
40% shifted to [20-30]

**Single parent**
N=34,010
Unpartnered
Partnered ref
6% shifted to 0

**Mother's height (cm)**
N=64,880
<150
≥150 ref
26% shifted to ≥150

**Father's height (cm)**
N=15,764
<162
≥162 ref
4% shifted to ≥162

# Household & Environmental Characteristics

**Clean cooking fuel usage**
N=2,029
No
Yes ref
35% shifted to Yes

**Improved floor**
N=35,105
No
Yes ref
85% shifted to Yes

**# of people in HH**
N=3,892
3 or less
4-5
6-7 ref
8+
91% shifted to 6-7

**# of children <5 in HH**
N=35,183
2+
1 ref
39% shifted to 1

**# of rooms in HH**
N=36,031
1
2
3 ref
4+
95% shifted to 3

**HH food security**
N=24,947
Food Insecure
Mildly Food Insecure
Food Secure ref
50% shifted to Food Secure

**Improved sanitation**
N=33,962
No
Yes ref
30% shifted to Yes

**HH wealth**
N=36,747
Q4 ref
Q2
Q1
Q3
74% shifted to Q4

Adjusted CIR

Population attributable fraction (%)

**Extended Data Fig. 7** | See next page for caption.

**Extended Data Fig. 7 | Rank-ordered associations between child, parental, and household characteristics and adjusted relative risks or population attributable fractions of wasting by age 24 months.** Blue points in the left panel show adjusted cumulative incidence ratios (CIRs) between higher-risk exposure levels and reference levels, and black points in the right panel show population attributable fractions (PAFs), the estimated proportion of the risk in the whole population that would be removed if the exposure were set to its indicated reference level. The number of children that contributed to each analysis is listed by exposure. The colored Y-axis label is either the level of exposure contrasted against the reference level to estimate the CIR, or the percent of the population shifted to the lowest-risk level to estimate the PAF. For at-birth exposures, at-birth stunting and wasting were excluded, and for postnatal exposures (breastfeeding practice and diarrheal disease), the cumulative incidence of wasting from 6-24 months was used. Cohort-specific estimates were adjusted for all measured confounders using ensemble machine learning and TMLE, and then pooled using random effects (Methods). Estimates are shown only for exposures measured in at least 4 studies. The PAF for diarrhea under 6 months was not calculable or plotted due to the unexpected CIR <1 for estimated higher diarrheal disease burden.

# Population intervention effect - LAZ, stratified by region

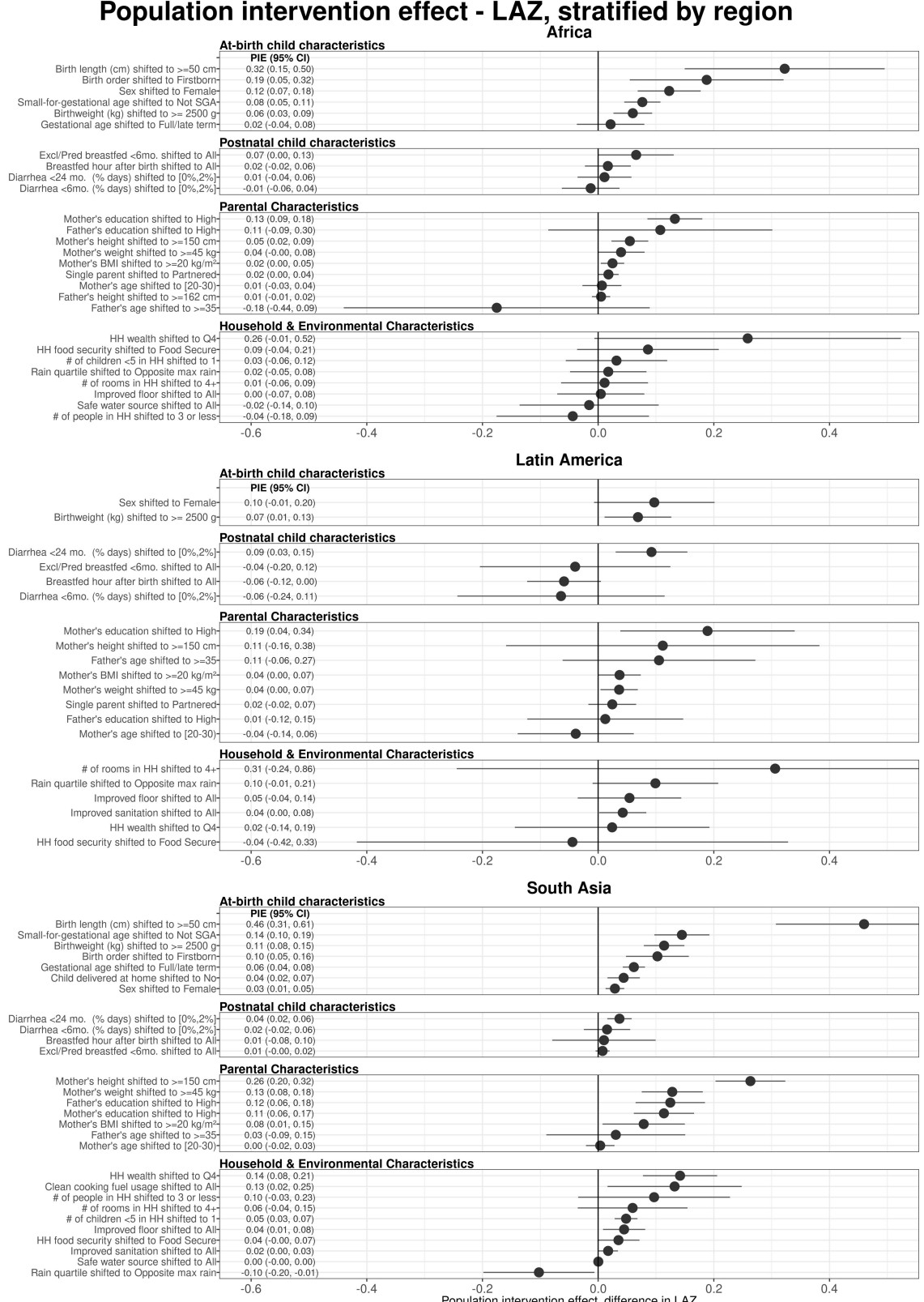

**Extended Data Fig. 8 | Regionally-stratified population intervention effects for length-for-age Z-scores at age 24 months.** Exposures, rank ordered by population intervention effect on child length-for-age z-score (LAZ) at age 24 months, stratified by region. The population intervention effect is the expected difference in population mean Z-score if all children had the reference level of the exposure rather than the observed distribution. For all plots, reference levels are printed next to the name of the exposure. Cohort-specific estimates were adjusted for all measured confounders using ensemble machine learning and TMLE, and then pooled using random effects (Methods). Estimates are shown only for exposures measured in at least 4 cohorts.

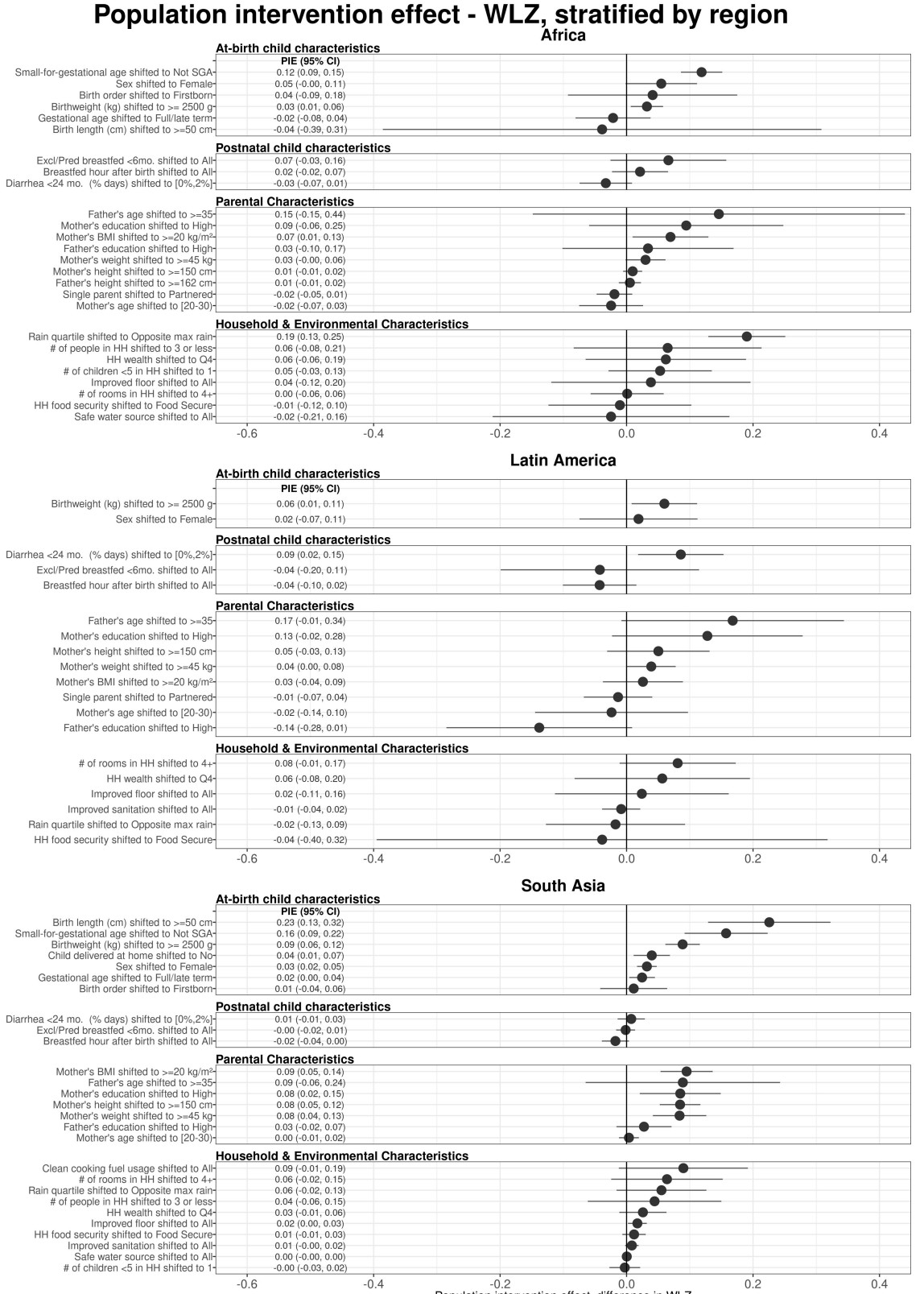

**Extended Data Fig. 9 | Regionally-stratified population intervention effects for weight-for-length Z-scores at age 24 months.** Exposures, rank ordered by population attributable difference on child weight-for-length z-score (WLZ) at age 24 months, stratified by region. The population intervention effect is the expected difference in population mean Z-score if all children had the reference level of the exposure rather than the observed distribution. For all plots, reference levels are printed next to the name of the exposure. Cohort-specific estimates were adjusted for all measured confounders using ensemble machine learning and TMLE, and then pooled using random effects (Methods). Estimates are shown only for exposures measured in at least 4 cohorts.

**a**

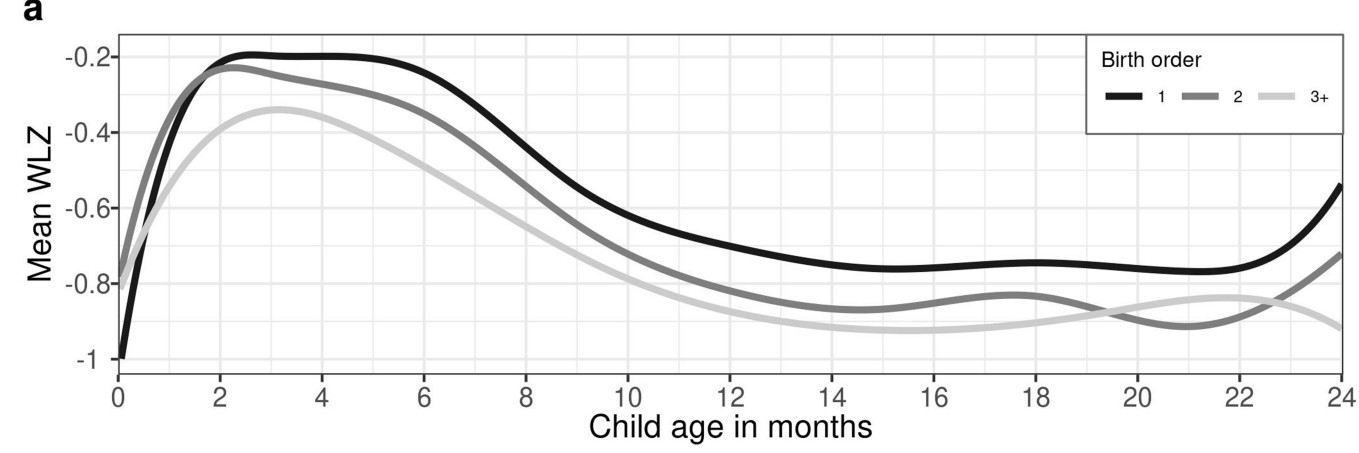

**b**

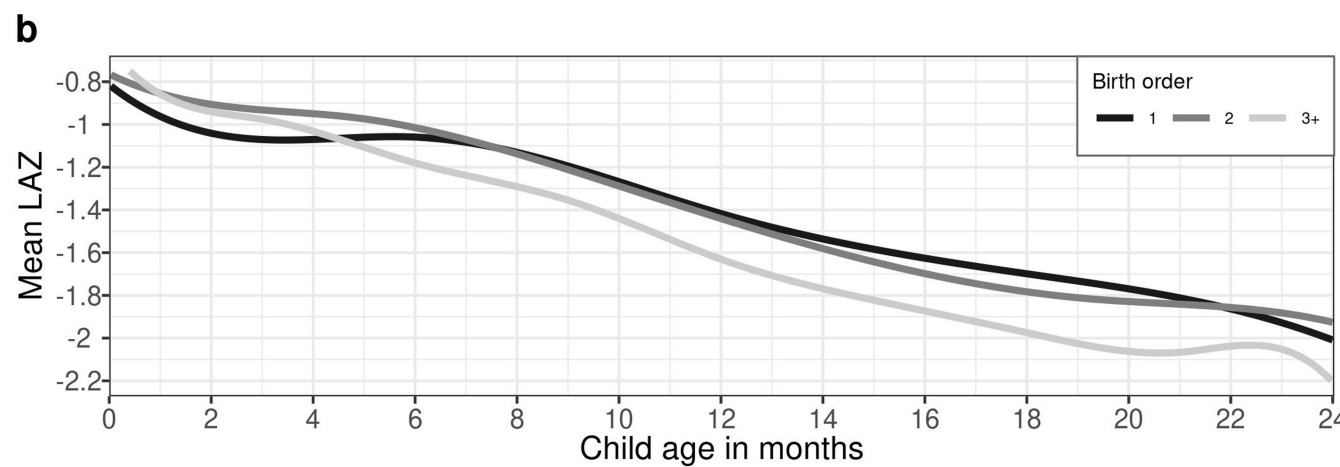

**Extended Data Fig. 10 | Child growth trajectories stratified by birth order.**
(a) Child weight-for-length Z-score (WLZ) trajectories, stratified by categories of child birth order. (b) Child length-for-age Z-score (LAZ) trajectories, stratified by categories of child birth order. Details on the estimation of growth trajectories are in the Methods. Child growth trajectories stratified by categories of all risk factors are available in Supplimentary Note 5.

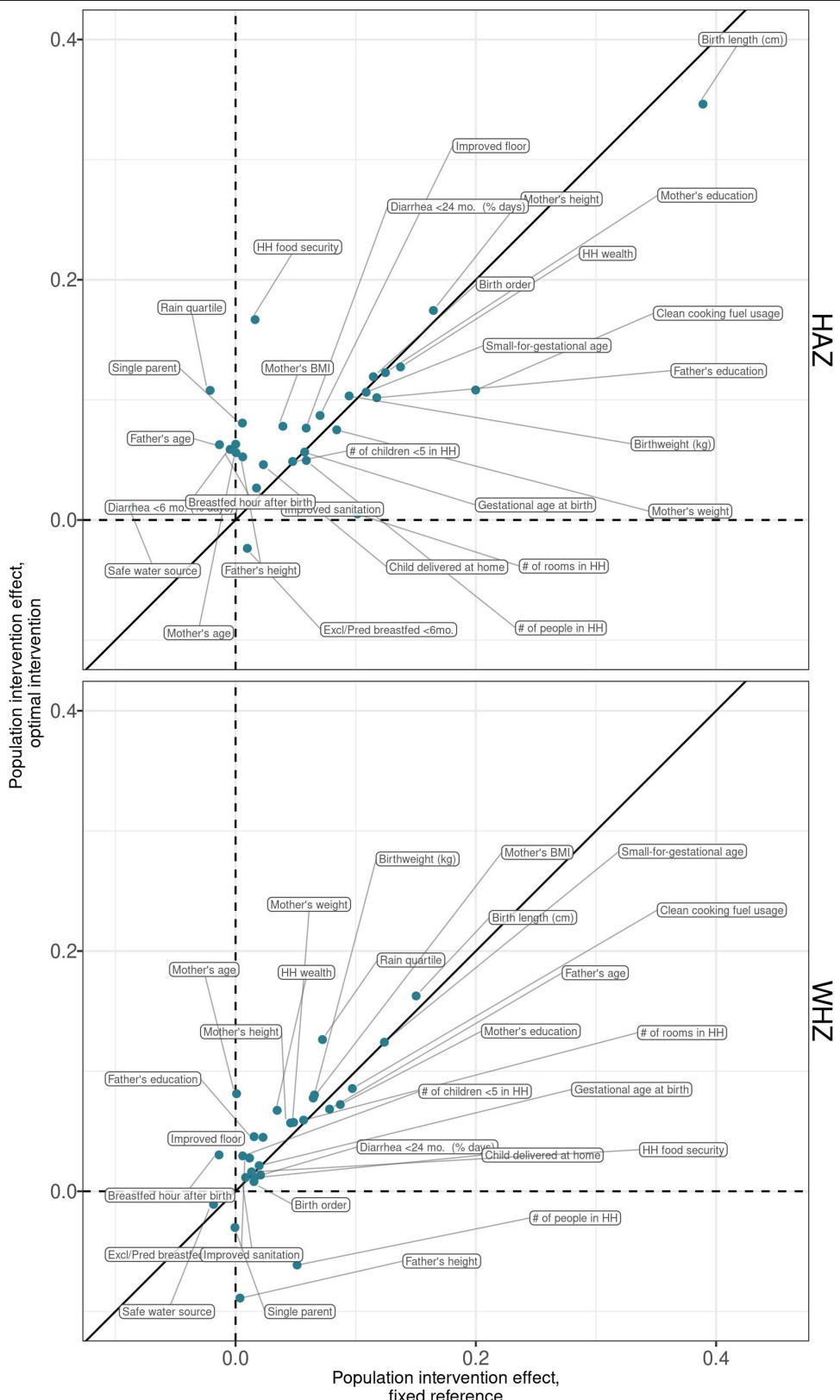

**Extended Data Fig. 11 | Comparing fixed-reference and optimal intervention estimates of the population intervention effect.** Pooled population intervention effects on child LAZ and WHZ at 24 months, with the X-axis showing attributable differences using a fixed, and the Y-axis showing the optimal intervention attributable difference, where the level the exposure is shifted to can vary by child. Points are labeled with the specific risk factor. Estimates farther from the diagonal line have larger differences between the static and optimal intervention estimates. The optimal intervention attributable differences, which are not estimated with an a-priori specified low-risk reference level, were generally close to the static attributable differences, indicating that the chosen reference levels were the lowest-risk strata in most or all children.

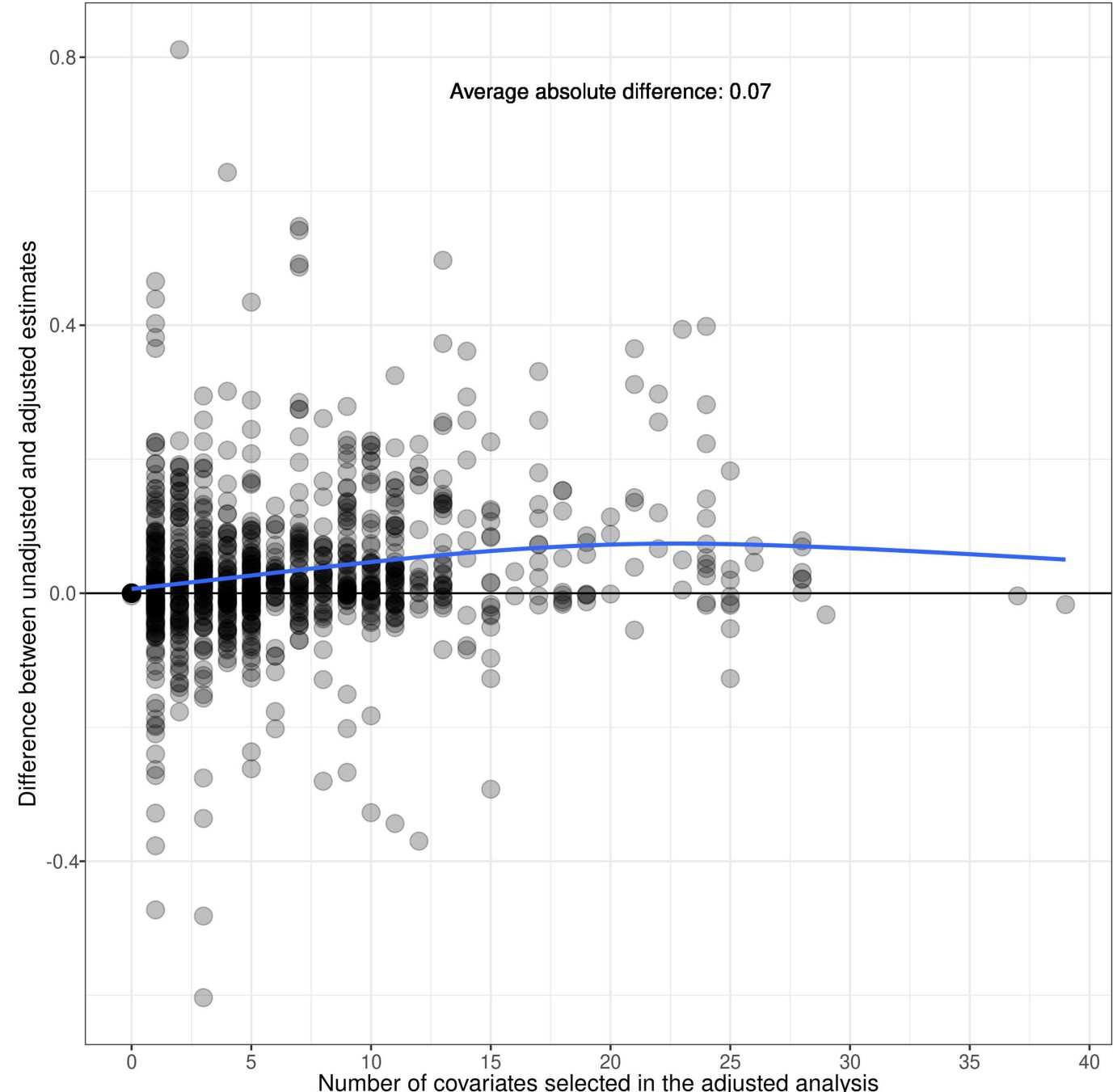

**Extended Data Fig. 12 | Difference between adjusted and unadjusted Z-score effects by number of selected adjustment variables.** Points mark the difference in estimates unadjusted and adjusted estimates of the difference in average Z-scores between exposed and unexposed children across 33 cohorts, 30 exposures and length-for-age and weight-for-length Z-score outcomes included in the analysis. Different cohorts measured different sets of exposures, and a different number of adjustment covariates were chosen for each cohort-specific estimate based on outcome sparsity, so cohort-specific estimates adjust for different covariates and numbers of covariates. The plot shows no systematic bias between unadjusted and adjusted estimates based on number of covariates chosen. The blue line shows the average difference between adjusted estimates from unadjusted estimates, fitted using a cubic spline.

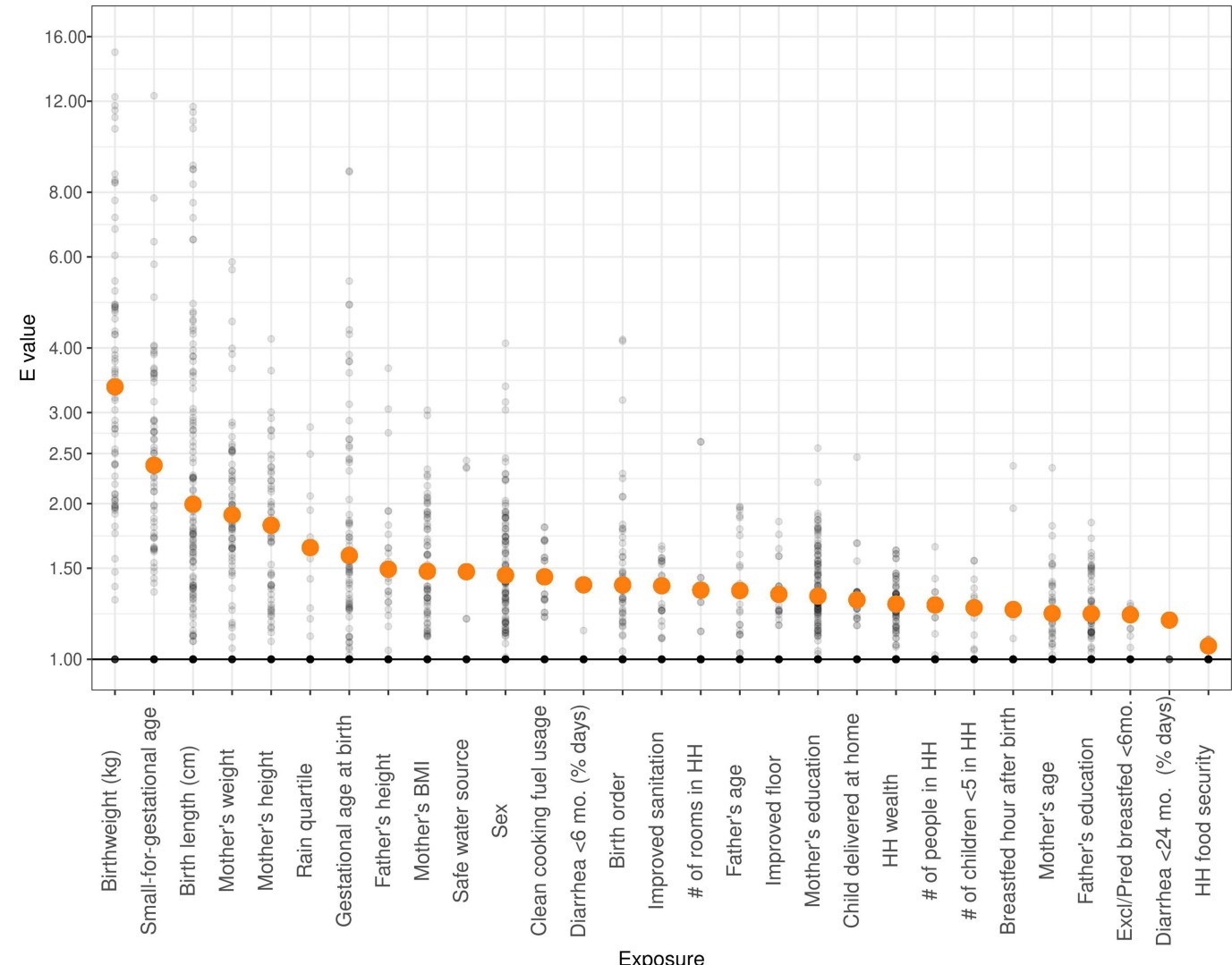

**Extended Data Fig. 13 | Assessing sensitivity of estimates to unmeasured confounding using E-values.** An E-value is the minimum strength of association in terms of relative risk that an unmeasured confounder would need to have with both the exposure and the outcome to explain away an estimated exposure–outcome association[71]. Orange points mark the E-values for the pooled estimates of relative risk for each exposure. Grey points are cohort-specific E-values for each exposure-outcome relationship. Non-significant pooled estimates have points plotted at 1.0. Orange points are median E-values among statistically significant estimates for each exposure. As an example, an unmeasured confounder would on average need to almost double the risk of both the exposure and the outcome to explain away observed significant associations for the birth length exposure.

# Extended Data Table 1 | Summary of *ki* cohorts

| Region, Study ID | Country | Study Years | Design | Children Enrolled* | Anthropometry measurement ages (months) | Total measurements* | Primary References |
|---|---|---|---|---|---|---|---|
| **South Asia** | | | | | | | |
| Biomarkers for EE | Pakistan | 2013-2015 | Prospective cohort | 380 | Birth, 1, 2, ..., 18 | 8918 | Iqbal et al 2018 Nature Scientific Reports[78] |
| Resp. Pathogens | Pakistan | 2011 - 2014 | Prospective cohort | 284 | Birth, 1, 2, ..., 17 | 3177 | Ali et al 2016 Journal of Medical Virology[79] |
| Growth Monitoring Study | Nepal | 2012 - Ongoing | Prospective cohort | 698 | Birth, 1, 2, ..., 24 | 13487 | Not yet published |
| MAL-ED | Nepal | 2010 - 2014 | Prospective cohort | 240 | Birth, 1, 2, ..., 24 | 5936 | Shrestha et al 2014 Clin Infect Dis[80] |
| CMC Birth Cohort, Vellore | India | 2002 - 2006 | Prospective cohort | 373 | Birth, 0.5, 1, 1.5, ..., 24 | 9131 | Gladstone et al. 2011 NEJM[81] |
| MAL-ED | India | 2010 - 2012 | Prospective cohort | 251 | Birth, 1, 2, ..., 24 | 5947 | John et al 2014 Clin Infect Dis[82] |
| Vellore Crypto Study | India | 2008 - 2011 | Prospective cohort | 410 | Birth, 1, 2, ..., 24 | 9825 | Kattula et al. 2014 BMJ Open[83] |
| CMIN | Bangladesh | 1993 - 1996 | Prospective Cohort | 280 | Birth, 3, 6, ..., 24 | 5399 | Pathela et al 2007 Acta Paediatrica[84] |
| TDC | India | 2008-2011 | Quasi-experimental | 160 | Birth, 1, 2, ..., 24 | 3723 | Sarkar et al. 2013 BMC Public Health[85] |
| MAL-ED | Bangladesh | 2010 - 2014 | Prospective cohort | 265 | Birth, 1, 2, ..., 24 | 5816 | Ahmed et al 2014 Clin Infect Dis[86] |
| PROVIDE RCT | Bangladesh | 2011 - 2014 | Individual RCT | 700 | Birth, 6, 10, 12, 14. 17, 18, 24, 39, 40, 52, 53 (weeks) | 12165 | Colgate et al 2016 Clin Infect Dis[87] |
| Food Suppl RCT | India | 1995 - 1996 | Individual RCT | 418 | Baseline, 6, 9, 12 | 2242 | Bhandari et al 2001 J Nutri[88] |
| Optimal Infant Feeding | India | 1999 - 2001 | Cluster RCT | 1535 | Birth, 3, 6, ..., 18 | 9539 | Bhandari et al 2004 J Nutri[89] |
| NIH Birth Cohort | Bangladesh | 2008 - 2009 | Prospective Cohort | 629 | Birth, 3, 6, ..., 12 | 6216 | Korpe et al. 2016 PLOS NTD[90] |
| JiVitA-4 Trial | Bangladesh | 2012 - 2014 | Cluster RCT | 5444 | 6, 9, 12, 14, 18 | 36167 | Christian et al 2015 IJE[91] |
| JiVitA-3 Trial | Bangladesh | 2008 - 2012 | Cluster RCT | 27342 | Birth, 1, 3, 6, 12, 24 | 109535 | West et al JAMA 2014[92] |
| NIH Cryptosporidium Study | Bangladesh | 2014 - 2017 | Prospective cohort | 758 | Birth, 3, 6, ..., 24 | 9774 | Steiner et al 2018 Clin Infect Dis[93] |
| **Africa** | | | | | | | |
| MAL-ED | Tanzania | 2009 - 2014 | Prospective cohort | 262 | Birth, 1, 2, ..., 24 | 5857 | Mduma et al 2014 Clin Infect Dis[94] |
| Tanzania Child 2 | Tanzania | 2007 - 2011 | Individual RCT | 2400 | 1, 2, ..., 20 | 32198 | Locks et al Am J Clin Nutr 2016[95] |
| MAL-ED | South Africa | 2009 - 2014 | Prospective cohort | 314 | Birth, 1, 2, ..., 24 | 6478 | Bessong et al 2014 Clin Infect Dis[96] |
| MRC Keneba | Gambia | 1987 - 1997 | Cohort | 2931 | Birth, 1, 2, ..., 24 | 40952 | Schoenbuchner et al. 2019, AJCN[97] |
| ZVITAMBO Trial | Zimbabwe | 1997 - 2001 | Individual RCT | 14104 | Birth, 6 wks, 3, 6, 9, 12 | 73651 | Malaba et al 2005 Am J Clin Nutr[98] |
| Lungwena Child Nutrition RCT | Malawi | 2011 - 2014 | Individual RCT | 840 | Birth, 1-6 wk, 6, 12 18 | 4346 | Mangani et al. 2015, Mat Child Nutr[99] |
| iLiNS-Zinc Study | Burkina Faso | 2010 - 2012 | Cluster RCT | 3266 | 9, 12, 15, 18 | 10552 | Hess et al 2015 Plos One[100] |
| CMIN GB94 | Guinea Bissau | 1994 - 1997 | Prospective Cohort | 870 | Enrollment and every 3 months after | 6459 | Valentiner-Branth 2001 Am J Clin Nutr[101] |
| **Latin America** | | | | | | | |
| MAL-ED | Peru | 2009 - 2014 | Prospective cohort | 303 | Birth, 1, 2, ..., 24 | 6442 | Yori et al 2014 Clin Infect Dis[102] |
| CONTENT | Peru | 2007 - 2011 | Prospective cohort | 215 | Birth, 1, 2, ..., 24 | 8339 | Jaganath et al 2014 Helicobacter[103] |
| Bovine Serum RCT | Guatemala | 1997 - 1998 | Individual RCT | 315 | Baseline, 1, 2, ...,8 | 2551 | Begin et al. 2008, EJCN[104] |
| MAL-ED | Brazil | 2010 - 2014 | Prospective cohort | 233 | Birth, 1, 2, ..., 24 | 5092 | Lima et al 2014 Clin Infect Dis[105] |
| CMIN Brazil89 | Brazil | 1989-2000 | Prospective Cohort | 119 | Birth, 1, 2, ..., 24 | 889 | Moore et al 2001 Int J Epidemiol.[106] |
| CMIN Peru95 | Peru | 1995 - 1998 | Prospective Cohort | 224 | Birth, 1, 2, ..., 24 | 3979 | Checkley et al. 2003 Am J Epidemiol.[107] |
| CMIN Peru89 | Peru | 1989 - 1991 | Prospective Cohort | 210 | Birth, 1, 2, ..., 24 | 2742 | Checkley et al. 1998 Am J Epidemiol.[108] |
| **Europe** | | | | | | | |
| PROBIT Study | Belarus | 1996 - 1997 | Cluster RCT | 16898 | 1, 2, 3, 6, 9, 12 | 124509 | Kramer et al 2001 JAMA[109] |
| **Mortality analysis only** | | | | | | | |
| Burkina Faso Zinc trial | Burkina Faso | 2010-2011 | Cluster RCT | 7167 | 6, 10, 14, 17, 22 | 15155 | Becquey et al 2016 J Nutr[110] |
| Vitamin A Trial | India | 1995-1996 | Cluster RCT | 3983 | 1, 3, 6, 9, 12 | 32570 | WHO CHD Vitamin A Group 1998 Lancet[111] |
| iLiNS-DOSE | Malawi | 2009-2011 | Individual RCT | 1932 | 6, 9, 12, 18 | 13801 | Maleta et al. 2015 J Nutr[112] |
| iLiNS-DYAD-M | Malawi | 2011-2015 | Individual RCT | 1235 | 1, 6, 12, 18 | 9207 | Ashorn et al 2015 J. Nutr[113] |

*Children enrolled is for children with measurements under 2 years of age. Total measurements are number of measurements of anthropometry on children under 2 years of age.

Data are from refs. 78–113.

## Extended Data Table 2 | Exposure variable summaries and prior published evidence – part 1

| Exposure variable | N children <24 months with measured exposure + length | Exposure levels [N (%)] | Categorization rules | Previous published evidence | Comparison to results in this analysis |
|---|---|---|---|---|---|
| Sex | 78751 | Female: 38444 (48.8%)<br>Male: 40307 (51.2%) | | In a meta-analysis of cohorts and surveys, boys had higher odds of being wasted and stunted than girls (pooled wasting OR 1.2, 95% CI 1.13 to 1.40, pooled stunting OR 1.29 95% CI 1.22 to 1.37). There was some evidence that the sex difference is smaller in South Asia.[114] | Supports our finding of increased risk of stunting (prevalence ratio (PR) of 1.15 (95% CI: 1.06, 1.26) at 24 months for wasting, 1.26 (95% CI: 1.13, 1.39) at 24 months for stunting), and slightly smaller prevalence ratios in South Asia (stunting PR: 1.02, 1.09, wasting PR: 1.22 [95% CI: 1.10, 1.35]). [Different P, CA]* |
| Birth weight (kg) | 65041 | Normal or high birth weight: 50940 (78.3%)<br>Low birth weight: 14101 (21.7%) | | A meta-analysis of 19 birth cohorts found a stunting PR of 2.92 (95% CI: 2.56, 3.33) associated with low birth weight (LBW) in children 1-5 years old, and a wasting PR of 2.68 (95% CI: 2.23, 3.21).[41] A meta-analysis of sub-Saharan African DHS datasets found LBW was strongly associated with stunting (adjusted OR: 1.68 95% CI: 1.58–1.78]) and wasting (aOR: 1.35 [95% CI: 1.20–1.38]) in children under 5. A systematic review of growth failure in sub-Saharan Africa consistently found LBW as a top risk factor for later wasting and stunting.[115] | Birthweight was also one of the strongest risk factors (PR of stunting at 24 months: 1.49 [95% CI: 1.37, 1.62], PR of wasting at 24 months: 1.87 [95% CI: 1.70, 2.06]), though with lower magnitude point estimates compared to the cohort meta-analysis and more aligned with the DHS analysis of older children. [Different P, CA, AV, MOA, SD]* |
| Birth length (cm) | 61703 | >=50 cm: 23313 (37.5%)<br>[48-50) cm: 14136 (39.6%)<br><48 cm: 24426 (22.9%) | | Birth length was the strongest predictor of stunting at in 2-year old children in the four country-specific cohorts included in the Women First trial (adjusted PR of 1.62 [95% CI: 1.39, 1.88] comparing children stunted at birth to children with a LAZ > -1 at birth).[116] | There was a very similar risk of low birth length. Children born with a length <48 cm (close to the stunting cutoff at birth) had 1.52 times the risk of stunting compared to children born with a length >50 cm (95% CI: 1.66, 2.58). Wasting risk was also increased (PR: 1.52 [95% CI: 1.21, 1.92]). [Different P, CA, AV]* |
| Gestational age at birth | 45269 | Full or late term: 23313 (51.5%)<br>Preterm: 6328 (14%)<br>Early term: 15628 (34.5%) | <260 days is preterm, [260-274) days is early term, >= 274 is full term | In a meta-analysis of 19 birth cohorts, infants born preterm had 1.69 times the odds (95% CI: 1.48, 1.93) of stunting and 1.55 times the odds (95% CI: 1.21, 1.97) of wasting from 1 to 5 years of age.[41] | The estimates are higher than in our study (PR of stunting at 24 months: 1.21 [95% CI: 1.13, 1.29], PR of wasting at 24 months: 1.13 [95% CI: 1.01, 1.26]), but support our finding of a significant increase in growth failure risk with preterm birth. [Different P, CA, AV, MOA]* |
| Small for gestational age | 39934 | Not small for gestational age 27161 (68%)<br>Small for gestational age 12773 (32%) | Children were classified as small-for gestational age if they had birthweights below the 10th percentile based on INTERGROWTH gestational age adjusted weight-for-age Z-scores (< -1.282 WAZ).[117] | In a meta-analysis of 19 birth cohorts, infants born small for gestational age (SGA) had 2.32 times the odds (95% CI: 2.12, 2.54) of stunting from 1 to 5 years of age compared to children not SGA, and they had 2.36 times the odds (95% CI: 2.14, 2.60) of wasting. [41] | The estimates are higher than in our study (PR of stunting at 24 months: 1.33 [95% CI: 1.22, 1.46], PR of wasting at 24 months: 1.83 [95% CI: 1.51, 2.21]), but support our finding of a significant increase in growth failure risk with SGA [Different P, CA, AV, MOA]* |
| Birth order | 46099 | 1: 17294 (37.5%)<br>2: 14107 (30.6%)<br>3+: 14698 (31.9%) | | A systematic review found that later birth order was consistently associated with a higher risk of stunting and wasting in the 16% of studies that identified birth order as an important risk factor for malnutrition.[118] In an analysis of 53 country-specific DHS analyses, birth order had an inconsistent relationship with stunting and wasting, with a decreased risk in second and third-born children compared to firstborn, but an increased risk in fourth-born or later.[119] In the four country-specific cohorts included in the Women First trial, second-born or later children had an increased Z-score trajectory from birth to 24 months, but a lower LAZ and higher risk of stunting at 24 months (PR: 1.12 [95% CI: 1.02, 1.24]).[116] | Our results were somewhat incongruous with the previous research. Birth order had a complex association with child growth failure, with a decreased risk of wasting and stunting in thirdborn or later children before 6 months of age (compared to firstborn children), and an increased risk after 6 months (Figure 4c). Stunting risk was similarly increased at 24 months (PR: 1.11 [95% CI: 1.01, 1.22]), but Z-score trajectories were also lower, in contrast to the Women First trial. [Different P, AV]* |
| Delivery location | 8487 | 0: 2793 (32.9%)<br>1: 5694 (67.1%) | | In an urban matched case-control study of infants 0-3 months old in Nigeria, the adjusted odds ratio associated with home delivery was 2.33 for severe stunting (95% CI: 1.50–3.60) and 2.90 for severe wasting (95% CI: 1.32–6.37) compared to delivery in public hospitals.[120] In a cohort in Malawi, home delivery was associated with 1.7 times the odds of severe stunting at 1 year of age after confounder adjustment (95% CI: 1.1 to 2.7).[121] | Home delivery had a significant but smaller association with any wasting (PR: 1.34 95% CI: 1.03, 1.74]) or stunting (PR: 1.14 [95% CI: 1.06, 1.23]) at 6 months, but a null association with severe stunting. Home delivery was still associated with stunting (PR: 1.14 [95% CI: 1.04, 1.24]) but not wasting or severe stunting at 24 months. Severe wasting was too rare among the cohorts that measured home delivery to estimate the association at either age. [Different P, CA, AV, MOA, SD, EC]* |
| Maternal height | 60742 | >=150 cm: 44831 (73.8%)<br><150 cm: 15911 (26.2%) | Cutoff chosen because a 150cm tall, 19-year-old woman has a HAZ of -2 | An analysis of 109 DHS surveys found a 1-cm increase in maternal height was associated with a decreased risk of child stunting (OR, 0.968; 95% CI, 0.967-0.968), and wasting (OR, 0.994; 95% CI, 0.993-0.995).[122] An analysis of 35 DHS surveys also found consistent, significant, exposure-response curve between categories of maternal height and risk of stunting and wasting.[119] | Maternal height was also consistently and strongly associated with all measures of child growth failure at the different examined ages. For example, the risk of stunting at 24 months was 1.63 times higher (95% CI: 1.46, 1.82) for children of stunted mothers compared to non-stunted mothers, and the risk of wasting was 1.18 time higher (95% CI: 1.09, 1.29). [Different P, CA, AV, MOA, SD, EC]* |
| Maternal body mass index (BMI) | 57627 | >=20 BMI: 34952 (60.7%)<br>< 20 BMI: 22675 (39.3%) | Calculated from maternal height and weight. Excludes mothers whose only weight measurement was taken during pregnancy. A 45 kg, 150 cm woman (the cutoffs for height and weight) has a BMI of 20. | A pooled analysis of 35 DHS cohorts found a significant increase in child stunting (OR: 1.64, p-value: <0.001) and wasting (OR: 1.64, p-value: <0.001) when mothers had BMI < 18.5 during pregnancy compared to mothers with a BMI > 25.[119] | Maternal BMI was also consistently and strongly associated with all measures of child growth failure at the different examined ages. The risk of stunting at 24 months was 1.21 times higher (95% CI: 1.06, 1.38) for children of lower weight mothers and the risk of wasting was 1.81 time higher (95% CI: 1.33, 2.47). [Different P, CA, AV, MOA, SD]* |
| Maternal weight | 59256 | >=45 kg: 40338 (68.1%)<br><45 kg: 18918 (31.9%) | Cutoff chosen because a 45kg heavy, 19-year-old woman has a WAZ of -2 | No studied examining maternal weight in kg were found; the studies identified all used BMI to examine associations between maternal weight and child growth failure. | |
| Mother's age | 70548 | [20-30): 41707 (59.1%)<br><20: 17826 (25.3%)<br>>=30: 11015 (15.6%) | | A systematic review found that children born to women under the age of 20 had a consistently greater risk of stunted children compared to women aged ≥ 20 years (OR from 1.37 to 7.56).[123] | We observed a similar increased risk of stunted children (at 24 months) born to teenage mothers (PR: 1.07 [95% CI: 1.02, 1.12] but a less consistent association with wasting (PR: 1.07 [95% CI: 0.83, 1.37]. However, the pooled risk in this study was much smaller, possibly because the children were younger than average, or a more complete control of confounding by SES and maternal size. [Different P, CA, AV, MOA, SD, EC]* |
| Maternal education | 69971 | High: 23013 (32.9%)<br>Low: 23702 (33.9%)<br>Medium: 23256 (33.2%) | Classified by splitting distribution of numbers of years of educations into thirds within each cohort, or grouping ordered categories of educational attainment into three levels. | Multiple systematic reviews have found maternal education to be the most frequently reported significant factor associated with child malnutrition (reported in >50% of studies).[118,124] A meta-analysis of 182 DHS datasets found a strong association between maternal education and wasting and stunting.[125] At a country level, a systematic review found improvements in maternal educational attainment predicted 17% of the total HAZ change in Pakistan (49), between 11% and 14% in Nepal (33, 49–51), 10% in Guinea (29) and India (49), and 7% in Cambodia (55).[126] However, a SRMA found that, while several included studies found inconsistent associations between maternal education and wasting and stunting, the pooled estimates were insignificant for both.[127] | After tertiling years of education within studies, low and medium maternal education was significantly associated with prevalence of stunting at 24 months compared to children of high-education mothers (low education PR: 1.21 [95% CI: 1.13, 1.30], medium education PR: 1.10 [95% CI: 1.04, 1.17]). Education was not associated with wasting at 24 months, but it was associated with the cumulative incidence of wasting between birth and 2 years (low education CIR: 1.09 [95% CI: 1.05, 1.13], medium education CIR: 1.08 [95% CI: 1.04, 1.11]) as well as with the cumulative incidence of stunting. The inconsistent associations in this and prior studies may be due to inconsistent coding of maternal education and differences in educational systems. [Different P, CA, AV, MOA, SD, EC]* |
| Paternal height | 15772 | >=162 cm: 15079 (95.6%)<br><162 cm: 693 (4.4%) | Cutoff chosen because a 162cm tall, 19 year old man has a HAZ of -2 | A meta-analysis of 14 DHS studies found a significant increase in the risk of child stunting (in children under 5 years) comparing the shortest to tallest quintiles of fathers (adjusted RR = 1.56 [95% CI: 1.47, 1.65]), though the association was stronger when using mother's heights. In a sensitivity analysis as part of a meta-analysis of 35 DHS studies, low paternal height (<155cm) was significantly associated with stunting (1.9 [95% CI: 1.7, 2.2]) and wasting (1.7 [95% CI: 1.4, 2.0]), but also less strongly than maternal height.[119] | We utilized a different cutoff than either study, comparing stunted fathers to non-stunted fathers, and found a significant, but smaller, association with the cumulative incidence of stunting in younger children than the DHS analysis (CIR: 1.12 [95% CI: 1.02, 1.23]). This was also smaller than the association between maternal stunting and child stunting, but the association with wasting was significant and of similar magnitude (CIR: 1.16 [95% CI: 1.04, 1.30]). [Different P, CA, AV, MOA, SD, EC]* |
| Paternal age | 18976 | >=35: 2289 (12.1%)<br><30: 13002 (68.5%)<br>[30-35): 3685 (19.4%) | | No meta-analyses examining this risk factor were found, and in general there were limited studies analyzing the association between father's age and child growth failure. A repeated cross-sectional survey in Indonesia found no association with father's age and stunting, wasting, or underweight.[128] | Children of fathers older than 35 had higher WLZ's at 24 months than children of fathers younger than 30 after adjusting for potential confounders, but there were no other growth associations. [Different P, CA, AV, MOA, SD, EC]* |
| Paternal education | 65728 | High: 12684 (19.3%)<br>Low: 23089 (35.1%)<br>Medium: 29955 (45.6%) | Classified by splitting distribution of numbers of years of educations into thirds within each cohort, or grouping ordered categories of educational attainment into three levels. | A meta-analysis of 182 DHS datasets found a similarly strong association between paternal and maternal education and child growth failure after confounder adjustment.[125] | We also found a similar association between paternal and maternal education and growth failure, with a null association with wasting and 24 months and significantly higher risk of stunting in children of low and medium education mothers compared to high education mothers. (low education PR: 1.39 [95% CI: 1.08, 1.57], medium education PR: 1.26 [95% CI: 1.07, 1.47]). [Different P, CA, AV, MOA, SD, EC]* |
| Caregiver partner status | 38222 | 0: 36393 (95.2%)<br>1: 1829 (4.8%) | Caregivers were classified as single if they were unmarried, widowed, or with a long-term long-distance partner. | A meta-analysis found that single mothers had a higher risk of infant low birth weight (OR 1.54 [95%CI 1.39–1.72]).[129] | Caregiver status was not associated with child stunting at 24 months (wasting was too rare to examine) or wasting at 6 months, but children of unpartnered mothers were significantly more likely to be stunted at 6 months (PR: 1.25 [95% CI: 1.08, 1.44]) and the cumulative incidence of wasting before 2 years of age (CIR: 1.12 [95% CI: 1.02, 1.24]). [Different P, CA, AV, MOA, SD, EC]* |
| Asset based household wealth index | 36754 | WealthQ4: 9618 (26.2%)<br>WealthQ3: 9165 (24.9%)<br>WealthQ2: 9012 (24.5%)<br>WealthQ1: 8959 (24.4%) | First principal component of a principal components analysis of all recorded assets owned by the household (examples: cell phone, bicycle, car). | A meta-analysis of 35 DHS surveys from sub-Saharan Africa examined the associations between household wealth indices computed by principal components analyses and stunting in children under 5 years. It found an OR of 1.34 (95% CI: 1.27, 1.42) when comparing the lowest versus highest wealth quintile.[130] Related, a meta-analysis of cash transfer programs found significant effects on height-for-age z-scores (of 0.03 5% CI 0.00 to 0.06) and a 2.1% decrease in stunting (95% CI -3.5% to -0.7%). [131] | Asset based household wealth was significantly associated with stunting at 24 months (PR: 1.26 [95% CI: 1.17, 1.36]) but not wasting (PR: 1.12 [95% CI: 0.98, 1.27]). [Different P, CA, AV, MOA, SD, EC]* |

All exposures included in the analysis, as well as the categories the exposures were classified into across all cohorts, categorization rules, the total number of children, the percentage of children in each category, select evidence from prior literature, and comparisons to our results. We selected the exposures of interest based on variables present in multiple cohorts that met our inclusion criteria, were found to be important determinants of stunting and wasting in prior literature, and could be harmonized across cohorts for pooled analyses. Where possible, we cite findings from recent randomized controlled trials and systematic reviews. All results from this manuscript referenced in this table are available in Supplementary Note 7. *Bracketed codes at the end of each cell in the "Comparison to results in this analysis" indicate limitations to comparisons with previous evidence due to differences in: P=population, CA=child age, AV=adjustment variables used in the analysis, MOA=measure of association, SD=study design, EC=exposure classification. Data are from refs. 40,114–131.

# Extended Data Table 3 | Exposure variable summaries and prior published evidence – part 2

| Exposure variable | N children <24 months with measured exposure + length | .Exposure levels [N (%)] | Categorization rules | Previous published evidence | .Comparison to results in this analysis |
|---|---|---|---|---|---|
| Household food security | 24461 | Food Secure: 12534 (51.2%)<br>Mildly Food Insecure: 7921 (32.4%)<br>Food Insecure: 4006 (16.4%) | Combination of three food security scales:<br>1. The Household Hunger Scale (HHS)[132]<br>2. Food Access Survey Tool (FAST)[133]<br>3. USAID Household Food Insecurity Access Scale (HFIAS), with middle 2 categories classified as mildly food insecure.[134]<br>And one survey question from the NIH Bangladesh birth cohort and NIH Bangladesh Cryptosporidium cohort:<br>"In terms of household food availability, how do you classify your household?"<br>1. Deficit in whole year<br>2. Sometimes deficit<br>3. Neither deficit nor surplus<br>4. Surplus<br>Where the middle two categories were classified as mildly food insecure. | A systematic review and meta-analysis of 21 studies found that food insecurity increased the risk of stunting (odds ratio [OR] = 1.17; 95% CI: 1.09–1.25) but not of wasting (OR = 1.04; 95% CI: 0.96–1.12). The associations were stronger in children older than 5 years old than those younger than 5 years, and in LMIC's.[135] Related to household food security, a recent meta-analysis of randomized controlled trials of small-quantity lipid-based nutrient supplements (SQ-LNSs) found a significant reduction in both stunting and wasting.[11] | Higher food insecurity was consistently but not significantly associated with wasting and stunting (PR of stunting at 24 months between food insecure and secure: 1.17 [95% CI: 0.96, 1.44] and PR of wasting 1.07 [95% CI: 0.95, 1.20], with similar associations with prevalence at 6 months and the cumulative incidence). [Different P, CA, AV, MOA, SD, EC]* |
| Improved flooring | 35354 | 1: 4693 (13.3%)<br>0: 30661 (86.7%) | | No meta-analyses examining this risk factor were found. Overall, there are limited studies specifically intervening to improve flooring or on the associations between improved flooring and growth, but an Ethiopian DHS analysis found an increased risk of stunting among children in households with natural/earth/sand floors versus cement/wood floors (OR: 1.33 [95% CI:1.08, 1.64]).[136] Additional research in two cohorts found improved flooring reduces soil-transmitted helminth and Giardia infections, which are associated with reduced growth.[137] | We also found an increased risk of stunting in younger children than the DHS analysis (PR at 24 months: 1.17 [95% CI: 1.08, 1.28]), and a reduction in LAZ and WLZ, but no effect on wasting. [Different P, CA, AV, MOA, SD, EC]* |
| Improved sanitation | 35086 | 1: 24119 (68.7%)<br>0: 10967 (31.3%) | WHO Joint Monitoring program definition | Two large trials of individual and combined WASH interventions (WASH Benefits Kenya and Bangladesh) found no effect of improved sanitation interventions on HAZ, WHZ, stunting, or wasting.[14,15] In contrast, a pooled meta-analysis of 35 DHS studies found an effect on stunting (PR: 1.10, [95% CI: 1.06, 1.13]) and wasting PR: 1.07, [95% CI: 1.02, 1.12]), potentially indicating residual confounding in observational analyses of WASH condition.[119] | Similar to the DHS analysis but in contrast to evidence from recent randomized trials, unimproved sanitation was associated with increased prevalence of stunting (PR: 1.09, [95% CI: 1.04, 1.14]) and wasting (PR: 1.22, [95% CI: 1.08, 1.38]) at 24 months. This potentially indicated either residual confounding from wealth and health seeking behavior of those with improved sanitation, or increased density of improved sanitation around household in observational studies compared to intervention studies. [Different P, CA, AV, MOA, SD, EC]* |
| Improved water source | 35284 | 1: 33777 (95.7%)<br>0: 1507 (4.3%) | WHO Joint Monitoring program definition | Two large trials of individual and combined WASH interventions (WASH Benefits Kenya and Bangladesh) found no effect of improved water interventions on HAZ, WHZ, stunting, or wasting.[14,15] A pooled meta-analysis of 35 DHS studies also found no effect on stunting, but did find an association with wasting (PR: 1.07, [95% CI: 1.02, 1.12]).[119] | Improved water source was not associated with wasting or stunting, aligned with the randomized trial findings. [Different P, CA, AV, MOA, SD, EC]* |
| Clean cooking fuel usage | 1401 | 1: 407 (29.1%)<br>0: 994 (70.9%) | | A meta-analysis of clean cookstove interventions found a reduction in stunting in children ages 0-59 months (Odds Ratio: 0.79 [95% CI: 0.70–0.89].[138] A different meta-analysis found a similar reduction in low birthweight (Odds Ratio: 0.73 [95% CI: 0.61–0.87], but did not examine growth in older children.[139] | Like the meta-analysis, clean cooking fuel use also associated with reduced stunting (PR at 24 months: 0.81, [95% CI: 0.68, 0.97]), and was also associated with reduced wasting at 24 months (PR: 0.59, [95% CI: 0.43, 0.83]). Clean cooking fuel use was also associated with the cumulative incidence of stunting in the first 6 months, but the studies with cooking fuel data didn't measure children at birth. [Different P, CA, AV, MOA, SD, EC]* |
| Number of children <5 in the household | 31610 | 1: 18963 (60%)<br>2+: 12647 (40%) | | No meta-analyses or systematic reviews examining this risk factor were found. A case-control study from Malaysia found an increased risk of any form of growth failure in children under 5 years old with more children in the household (PR comparing households 4 or more children to three or less: 5.86 [95% CI: 1.96-17.55]).[140] | Other children in the household was associated with increased stunting (PR: 1.19 [9%% CI: 1.04, 1.35]) but not wasting at 24 months. [Different P, CA, AV, SD]* |
| Number of individuals in the household | 1805 | 3 or less: 363 (20.1%)<br>4-5: 745 (41.3%)<br>6-7: 452 (25%)<br>8+: 245 (13.6%) | | No meta-analyses or systematic reviews examining this risk factor were found. A cross-sectional study from Madagascar found an increased risk of stunting and wasting in children 5-14 years with more people in the household (stunting PR comparing households with 5 or more people children to four or less: 1.17 [95% CI: 1.03-1.33], wasting PR: 1.24 [95% CI: 1.04–1.48].[141] | There was a small but non-significant increase in risk of stunting and wasting with more individuals in the household. [Different P, CA, AV, SD]* |
| Number of rooms in household | 35929 | 4+: 2492 (6.9%)<br>1: 20210 (56.2%)<br>2: 9484 (26.4%)<br>3: 3743 (10.4%) | | No meta-analyses examining this risk factor were found, and DHS surveys generally measure the number rooms used for sleeping, not the total number of rooms. | The number of rooms in the household was not associated with increased risk of stunting or wasting. |
| Rain season | 9769 | Opposite max rain: 2469 (25.3%)<br>Pre max rain: 2248 (23.0%)<br>Max rain: 2718 (27.8%)<br>Post max rain: 2334 (23.9%) | Rainfall data was extracted from Terraclimate, a dataset that combines readings from WorldClim data, CRU Ts4.0, and the Japanese 55-year Reanalysis Project.[142] For each study region, we averaged all readings within a 50 km radius from the study coordinates. If GPS locations were not in the data for a cohort, we used the approximate location of the cohort based on the published descriptions of the cohort. The three-month period opposite the three months of maximum rainfall was used as the reference level (e.g., if June-August was the period of maximum rainfall, the reference level is child mean WLZ during January-March). Due to the time-varying nature of this exposure, N's are reported for children with length measures at 24 months and measures of rain season. | In a SRMA, drought conditions were significantly associated with wasting (OR: 1.46 [95% CI: 1.05, 2.04]).[143] A meta-analysis of 55 DHS datasets found that both abnormally high and low rainfall was associated with reduced HAZ and WHZ.[144] A systematic review found extreme rainfall was associated with an increased risk of wasting, but found crop growing season rainfall was protective for wasting.[145] A different systematic review found consistent associations between rainfall and HAZ, but the magnitude and direction of effect varied by study and the timing of the rainfall that the study examined.[146] | WHZ was significantly lower and wasting significantly higher during the three months of highest rainfall and the three months after the highest rainfall period, but there was no significant association between rain and stunting of HAZ. [Different P, CA, AV, MOA, SD, EC]* |
| Breastfed in the hour after birth | 49168 | 1: 11609 (23.6%)<br>0: 37559 (76.4%) | | Early initiation of breastfeeding was significantly associated with stunting in most cross-sectional studies evaluated in a systematic review, but most of these estimates were not adjusted for confounding.[147] A cohort study found no association with stunting and wasting,[148] while a pooled analysis of 35 DHS surveys found an increase in stunting odds (OR: 1.037, P-value <0.001) but a decreased risk of wasting (OR: 0.937, P-value <0.001).[119] | Early breastfeeding was not significantly associated with reduced stunting or wasting, in contrast to prior cross-sectional studies but aligned with prior evidence from analyses of cohorts. [Different P, CA, AV, MOA, SD]* |
| Exclusive or predominant breastfeeding in the first 6 months of life | 26173 | 1: 18285 (69.9%)<br>0: 7888 (30.1%) | Exclusive breastfeeding: mother reported only feeding child breastmilk on all dietary surveys<br>Predominant breastfeeding: mother reported only feeding child breastmilk, other liquids, or medicines on all dietary surveys | A SRMA of studies of exclusive breastfeeding found a reduction in stunting (OR = 0.73 [95% CI = 0.55, 0.95]), but a SRMA of breastfeeding promotion interventions found no impact on LAZ and an unexpected reduction in WLZ.[26] | Non-exclusive breastfeeding was associated with a smaller but still significant increase in the prevalence of stunting at 6 months (PR: 1.11 [95% CI: 1.03, 1.21]) and 24 months (PR: 1.05 [95% CI: 1.00, 1.10]), but there was no association with wasting. [Different P, CA, AV, MOA, SD, EC]* |
| Cumulative percent of days with diarrhea under 6 months | 3735 | [0%, 2%]: 2245 (60.1%)<br>>2%: 1490 (39.9%) | Percent days defined as proportion of disease surveillance days a child had diarrhea during the time interval. Diarrhea defined by 3 or more loose stools, or bloody stool, in a 24 hour period. Only included studies with at least 100 disease surveillance measurements during age range. | A pooled analysis of nine cohorts and trials found that the adjusted odds of stunting at 24 months increased by 1.16 for every 5% absolute increase in longitudinal prevalence of diarrheal disease prior to 24 months (95% CI 1.07–1.25).[43] A separate analysis of 7 cohorts found WHZ, but not LAZ was reduced 30 days after diarrheal disease, while a higher cumulative burden of diarrhea reduced linear growth at 24 months (-0.1 LAZ per 10 days of diarrhea.[150] | We found a similar magnitude in the reduction of LAZ at 24 months (-0.14 z [95% CI: -0.21 -0.06]) associated with increased diarrhea, but no association with WLZ, stunting, or wasting. [Different P, AV, EC]* |
| Cumulative percent of days with diarrhea under 24 months | 12639 | [0%, 2%]: 6133 (48.5%)<br>>2%: 6506 (51.5%) | Same as above. | In the second study detailed above, it was estimated that a child with the average diarrhea burden during the first 6 months of life who then went on to have no diarrhea did not have a significantly lower LAZ at 24 months than a child with no diarrhea.[150] Because there was an overall effect of diarrhea on LAZ at 24 months, this indicated the potential for catch-up growth, or a lower impact of infant diarrhea on growth. | We also found no association between diarrhea before 6 months and growth at 24 months, and there was no association with growth outcomes at 6 months. [Different P, AV, EC]* |

All exposures included in the analysis, as well as the categories the exposures were classified into across all cohorts, categorization rules, the total number of children, the percentage of children in each category, select evidence from prior literature, and comparisons to our results. We selected the exposures of interest based on variables present in multiple cohorts that met our inclusion criteria, were found to be important determinants of stunting and wasting in prior literature, and could be harmonized across cohorts for pooled analyses. Where possible, we cite findings from recent randomized controlled trials and systematic reviews. All results from this manuscript referenced in this table are available in Supplimentary Note 7. *Bracketed codes at the end of each cell in the "Comparison to results in this analysis" indicate limitations to comparisons with previous evidence due to differences in: P=population, CA=child age, AV=adjustment variables used in the analysis, MOA=measure of association, SD=study design, EC=exposure classification. Data are from refs. 11,14,15,101,132–150.

**Extended Data Table 4 | *ki* cohort and country-level mortality rates**

| Study | Country | Number of deaths under 2 | Under 2 mortality rate in cohort (%) | Infant (Under 1) mortality rate in cohort (%) | Infant (Under 1) mortality country rate (%, UNICEF) |
|---|---|---|---|---|---|
| Burkina Faso Zn | Burkina Faso | 39 | 0.54 | 0.42 | 5.4 |
| iLiNS-DOSE | Malawi | 53 | 2.74 | 1.92 | 3.1 |
| iLiNS-DYAD-M | Malawi | 54 | 4.37 | 3.48 | 3.1 |
| JiVitA-3 | Bangladesh | 934 | 3.41 | 2.85 | 2.6 |
| JiVitA-4 | Bangladesh | 49 | 0.9 | 0.39 | 2.6 |
| Keneba | The Gambia | 65 | 2.22 | 1.52 | 3.6 |
| VITAMIN-A | India | 108 | 2.70 | 2.7 | 2.8 |
| ZVITAMBO | Zimbabwe | 1113 | 7.89 | 6.57 | 3.8 |

Under 1-year country-specific mortality rate is from UNICEF (https://data.unicef.org/country), and is higher than the cohort-specific under 2-year mortality rate for all cohorts used in the mortality analysis.

# Reporting Summary

## Statistics

For all statistical analyses, confirm that the following items are present in the figure legend, table legend, main text, or Methods section.

| n/a | Confirmed | |
|---|---|---|
| ☐ | ☒ | The exact sample size (*n*) for each experimental group/condition, given as a discrete number and unit of measurement |
| ☐ | ☒ | A statement on whether measurements were taken from distinct samples or whether the same sample was measured repeatedly |
| ☒ | ☐ | The statistical test(s) used AND whether they are one- or two-sided<br>*Only common tests should be described solely by name; describe more complex techniques in the Methods section.* |
| ☐ | ☒ | A description of all covariates tested |
| ☐ | ☒ | A description of any assumptions or corrections, such as tests of normality and adjustment for multiple comparisons |
| ☐ | ☒ | A full description of the statistical parameters including central tendency (e.g. means) or other basic estimates (e.g. regression coefficient) AND variation (e.g. standard deviation) or associated estimates of uncertainty (e.g. confidence intervals) |
| ☐ | ☒ | For null hypothesis testing, the test statistic (e.g. *F*, *t*, *r*) with confidence intervals, effect sizes, degrees of freedom and *P* value noted<br>*Give P values as exact values whenever suitable.* |
| ☒ | ☐ | For Bayesian analysis, information on the choice of priors and Markov chain Monte Carlo settings |
| ☐ | ☒ | For hierarchical and complex designs, identification of the appropriate level for tests and full reporting of outcomes |
| ☐ | ☒ | Estimates of effect sizes (e.g. Cohen's *d*, Pearson's *r*), indicating how they were calculated |

*Our web collection on statistics for biologists contains articles on many of the points above.*

## Software and code

Policy information about availability of computer code

| Data collection | This manuscript is a secondary data analysis of existing study data from 33 cohorts and trials, so we were not involved in original data collection. |
|---|---|
| Data analysis | All analyses were conducted using R statistical software, and scripts that reproduce all analyses are available on Github here: https://github.com/child-growth/ki-longitudinal-growth |

For manuscripts utilizing custom algorithms or software that are central to the research but not yet described in published literature, software must be made available to editors and reviewers. We strongly encourage code deposition in a community repository (e.g. GitHub). See the Nature Portfolio guidelines for submitting code & software for further information.

## Data

Policy information about availability of data

All manuscripts must include a data availability statement. This statement should provide the following information, where applicable:
- Accession codes, unique identifiers, or web links for publicly available datasets
- A description of any restrictions on data availability
- For clinical datasets or third party data, please ensure that the statement adheres to our policy

Data are available upon agreement from the data contributors of individual studies, whose contact information is available from the Bill and Melinda Gates Foundation Knowledge Integration project (email [TO BE ADDED] upon reasonable request.

# Research involving human participants, their data, or biological material

Policy information about studies with [human participants or human data](). See also policy information about [sex, gender (identity/presentation), and sexual orientation]() and [race, ethnicity and racism]().

| | |
|---|---|
| Reporting on sex and gender | We use the term sex throughout; data on gender was not collected in the original studies used. We include sex as a risk factor and examine sex as an effect modifier of mortality risk (supplementary material) |
| Reporting on race, ethnicity, or other socially relevant groupings | We did not have information on the race or ethnic groups of study participants. |
| Population characteristics | See below. |
| Recruitment | N/A: Secondary analysis of 33 completed studies. |
| Ethics oversight | N/A |

Note that full information on the approval of the study protocol must also be provided in the manuscript.

# Field-specific reporting

Please select the one below that is the best fit for your research. If you are not sure, read the appropriate sections before making your selection.

☐ Life sciences　　☒ Behavioural & social sciences　　☐ Ecological, evolutionary & environmental sciences

For a reference copy of the document with all sections, see [nature.com/documents/nr-reporting-summary-flat.pdf]()

# Behavioural & social sciences study design

All studies must disclose on these points even when the disclosure is negative.

| | |
|---|---|
| Study description | This study performed a quantitative analysis of de-identified secondary, longitudinal data on child growth. |
| Research sample | The data analyzed in this study were amassed as part of the Knowledge Integration (ki) initiative of the Bill & Melinda Gates Foundation, which aggregated observations on millions of participants from a global collection of studies on child birth, growth and development. We selected longitudinal cohorts from the database that met five inclusion criteria: 1) conducted in LMICs; 2) enrolled children between birth and age 24 months and measured their length and weight repeatedly over time; 3) did not restrict enrollment to acutely ill children; 4) enrolled at least 200 children; and 5) collected anthropometry measurements at least every 3 months. Thirty-three cohorts from 15 countries met inclusion criteria, and 83,671 children and 592,030 total measurements were included in this analysis. Four additional cohorts were included in analyses of mortality, including 14,317 more children and 70,733 additional measurements. |
| Sampling strategy | Not applicable. |
| Data collection | Not applicable. |
| Timing | Included datasets were collected between 1990 and 2014. |
| Data exclusions | We dropped 1,190 (0.2%) unrealistic measurements of LAZ (>+6 or <−6 Z), 1,330 (0.2%) measurements of WAZ (> 5 or < −6 Z), and 1,670 (0.3%) measurements of WLZ (>+5 or −5 Z), consistent with WHO recommendations. |
| Non-participation | Not applicable. |
| Randomization | Participants were not randomly assigned. |

# Reporting for specific materials, systems and methods

We require information from authors about some types of materials, experimental systems and methods used in many studies. Here, indicate whether each material, system or method listed is relevant to your study. If you are not sure if a list item applies to your research, read the appropriate section before selecting a response.

## Materials & experimental systems

| n/a | Involved in the study |
|---|---|
| ☒ | ☐ Antibodies |
| ☒ | ☐ Eukaryotic cell lines |
| ☒ | ☐ Palaeontology and archaeology |
| ☒ | ☐ Animals and other organisms |
| ☒ | ☐ Clinical data |
| ☒ | ☐ Dual use research of concern |
| ☒ | ☐ Plants |

## Methods

| n/a | Involved in the study |
|---|---|
| ☒ | ☐ ChIP-seq |
| ☒ | ☐ Flow cytometry |
| ☒ | ☐ MRI-based neuroimaging |

