## [Peer Review File · Nature]

Manuscript Title: Causes and consequences of child growth failure in low- and middle-income countries

Reviewer Comments & Author Rebuttals

Reviewer Reports on the Initial Version:

Referees' comments:

Referee #1 (Remarks to the Author):

This paper addresses the risk factors and longer term health effects of wasting and stunting. Unlike the two descriptive papers, one can argue that the inferred associations represent etiological relationships and therefore are not affected by the issue of when/where the cohorts were enrolled. That being said, given the nature of risk factors considered and some of the results, there may be true (vs. merely stochastic) differences across cohorts which should be empirically assessed.

A feature of the analysis – the inclusion of a large number of risk factors and application of flexible ML methods - which I assume was meant to be a strength of the paper may well have acted to weaken because of the number and nature of variables (risk factors) included.

The majority of risk factors analyzed here have been evaluated in retrospective or prospective cohorts and in some cases trials, in some cases with very detailed analysis to reveal temporal patterns (e.g. episodes of diarrhea by some consortium members). So unless the combination of data and flexible ML methods reveal some major unknown relationship the new knowledge generated is more limited than the two descriptive papers. In view of this reviewer, this is especially the case given findings from trials and quasi-randomized studies on both what works (e.g. Mexico's PROGRESA experiment) and what doesn't at least in real-world conditions (e.g. some WASH trials and perhaps community-based zinc supplementation). The main result, on the role of pre-conception/pregnancy (via birth size), was shown a few years ago via the work of CHERG including by Dr Christian who is one of the authors, as has been the role of macro development conditions (arguably with more granularity where people have looked at within country variations during rapid growth or structural adjustment).

Methodologically, the large number of variables included has made the authors make choices about what to adjust for to avoid adjusting for potential mediators. If mediators were only affected by the variables in the model, then this would serve to estimate their total effects. However, if there is an unmeasured cause for the mediator, then excluding it from the model would lead to unmeasured confounding. I would recommend a full analysis with all variables, and then doing specific mediation analysis where relevant as some sort of additional materials.

Substantively, the large number of variables included, possibly together with data issues or specific temporal relationships, leads to results that in some instances seem spurious. To be clear, this is an empirical analysis and the authors should be commended for reporting what they find. But previous in-depth work by some consortium members and others has shown the role of diarrhea, breastfeeding, food insecurity and persistent wasting which prove to have no or close to no effect here. Other variables, like wealth or birth order, have effects that are a bit "all over the place". Why is this emerging? Is it a matter of heterogeneity that when pooled leads to such results? Is it about temporal relationships? Is it some form of model mis-specification? Etc?

As a specific question, given the mother's height and BMI are both in the model, why was weight included?

The mortality analysis largely confirms the results of Reference 1 and its companion paper that looked at severity of stunting and all-cause and cause-specific mortality. As a specific questions: I would have thought that the analysis would be based on time-to-event design and using a Cox model, perhaps with exposure status updated at each follow up (as done in the previous pooled analysis of cohorts for all-cause and cause-specific mortality) since there is nothing particular about dying before two years of age versus later in childhood. Is there a particular data reason to deviate from this?

Referee #2 (Remarks to the Author):

This is a very impressive and detailed paper on child growth failure in LMICs, based on a large database of 35 cohorts. The results on age-varying effects on growth failure are fascinating. I only have two major comments and a few minor comments.

Major comments

-Have all children been consistently followed from birth to their second birthday in the 8 cohorts analyzed for child mortality? Which are these cohorts? I was surprised by the small share of neonatal deaths in these cohorts (about 20% of neonatal deaths among deaths that occurred in the first 2 years). According to UN IGME estimates (childmortality.org), neonatal deaths accounted for 46% of all under-five deaths in LMICs and 62% of all infant deaths in 2019. I was expecting a much higher percentage of neonatal deaths among those occurring in the first 2 years (somewhere in between 46% and 62%). Is this a sign of underreporting of neonatal deaths (and corresponding births) in the cohorts? Or could this be due to the fact that early neonatal deaths were not in the analysis as the children did not have anthropometry measurements? This could be added in a note for Extended Data Figure 10.

- I find it strange that you report a relative risk of death in the first 24 months associated with persistent wasting under 6 months, as children would need to survive for a sufficiently long time to be measured at least 4 times. Would it be preferable to report on all-cause mortality rates between month 6 and month 24, based on anthropometric measures taken in the first 6 months?

Minor comments

Summary

- Why do the authors state that the higher risk of illness and mortality (or the association with child growth failure) contributed to the SDG 2.2 goal in the summary? This is a bit unclear.
- Could you specify in the abstract that you are analyzing trajectories until age 2?
- The title refers to low- and middle-income countries, but the summary only mentions low-income settings.

Introduction

- "WASH interventions, which aim to reduce childhood infections... in non-emergency settings". Why indicate in non-emergency-settings?

Pooled longitudinal analyses

- Persistent wasting is defined as >50% measurements of WLZ <-2 during an age period with at least 4 measurements. But unless I missed it, the paper does not provide the mean of the interval between measurements. I was wondering if these 4 measurements were typically spread over a long or short period.

Age-varying effects on growth failure

- Can you provide some explanation as to why children with birth order 2+ have a lower risk of

stunting and wasting at less than 6 months of age? What are the possible explanations? Could it be that first-born children are born to younger mothers?

Consequences of early growth failure

- "we estimated the relative risk of mortality across measures of growth failure in the first 6 months within eight cohorts that reported mortality endpoints, including 2,510 child deaths by age 24 months". This sentence could be reformulated. At first, I thought it was referring to mortality in the first 6 months, I had to read twice to understand that mortality was counted up to age 24 months, while the 6 months refer to growth failure.

- Extended Data Figure 10 is difficult to interpret and not very informative as we can't see the anthropometry measurements for those who survived.

Referee #3 (Remarks to the Author):

Summary of the key results

Originality and significance: The breadth of work undertaken in this last paper of this series is considerable and reveals important findings that are well situated in the previous literature. The population attributable difference was a useful measure of association to use for this type of an analyses because there are important policy relevant implications in how this can be communicated to policy makers and program implementers. The population level predictors of LAZ and WLZ at 24 months identified reflect that both nutrition-sensitive and specific interventions are warranted to shift those population level attributes. Also, noting that the PAF was relatively modest for the factors identified as significant possibly signals to the need for cost effectiveness of certain interventions as it relates to improving nutritional outcomes. Not that cost-effectiveness should drive choice of an intervention for improved nutrition alone but that it is definitely a factor to consider. Other important findings include the relationship between persistent wasting and subsequent linear growth failure and mortality and the timing of these relationships.

Data & methodology:

The methods portion of this manuscript has been undertaken with much care and detail. One missing element was the DAGs eluded to in linked 796-798 to assess adjustments to the models. Can these please be included in the supplementary online material?

Lines 316-318: Did the authors find this to be the case for growth velocity as well?

Appropriate use of statistics and treatment of uncertainties

Lines 324-325: How do authors explain the finding of birth order and its associations with growth failure before and after 6 mos?

Conclusions: Given the modest effects of some of these factors on reducing the incident of growth failure, especially maternal nutritional status, wouldn't language around the preconception and pregnancy interventions to promote optimal growth need to be tempered? What are the tradeoffs here from a more practical standpoint of application by policy makers and practitioners?

Clarity and context:

Lines 39-41: High attributable risk from prenatal causes, and severe consequences for children who experienced early growth failure, support a focus on pre-conception and pregnancy as key opportunities for new preventive interventions. - This sentence needs to be rephrased. High attributable risk of what?

Lines 185-186: Many of the statements included in the first paragraph of the introduction included in this sentence reflects the evidence referenced as causal when this is not necessarily the case.

Would encourage more careful epidemiologic language to be used here giving careful consideration to the measures of association used in the references included when stating existing relationships between stunting/wasting and long-term consequences.

Given the richness of the data presented and analyses conducted, the abstract is rather limited in showcasing the depth of the work. While there is a word limitation here, would reconsider reframing to make the abstract more compelling.

Referee #4 (Remarks to the Author):

1. Based on the content of the study focusing on maternal anthropometrics and sociodemographic details, the title of "causes... of child growth failure" is a bit misleading. By my read, with the possible exceptions of breastfeeding differences, maternal anthropometrics, and rainy season, the authors have not explored biologic pathways of causality. Only the first is potentially modifiable (in any short period of time). While this doesn't negate their findings, it does call for a tempering of language and perhaps reconsideration of the title.
2. See comments below under "conclusions". My overall takeaway is that careful analysis of these data is extraordinarily valuable, and the authors have made a good first pass. However, there is such a mixture of concepts, approaches, principles, and methods used in this paper that it is really hard to compare the insights either to each other or to previous "knowledge."
3. This is an important paper and an important topic. With some focused work in clarification of language and harmonization of conceptual approach across all the presented analyses, this could be an important contribution to child health globally.
4. An interim point at 1000 days is reasonable, but why stop at 24 months? Growth failure continues to accumulate/ accelerate in older children (as suggested by figure 3) so this choice seems odd.
5. Was the outcome only AT 24 months? Or anytime IN the first 24 months of life? Please clarify throughout.
6. Line 221 + 229 – this goes from "millions" of children to 94k. Why the huge drop off?
7. Line 238 – why asymmetry in "unrealistic" measurement truncation? +/-6 vs. +6/-5 vs. +/-5. WHO recommendations on this do not have empirical basis. How many children were eliminated by this? It is a small % overall, but could be a large proportion of the outcomes since these are theoretically the most at-risk subgroups.
8. Line 250 – I conceptually disagree with this assertion. Maternal height SHOULD be adjusted for BW. The authors statement is correct, but this makes BW a mediator of stunting. If, as they claim later, that length trajectories AFTER birth are no longer related to maternal height, this further supports separating out into direct causation vs. mediation pathway.
9. Line 260 – what covariates? What was the source of them?
10. Line 274 – breastmilk *quality*? How was this measured? And standardized?
11. Figure 2 – multiple comments/ questions. Overall I find this figure quite difficult to decipher.
 - ♣ Very hard to follow because the ordering of the factors in each panel are in different order. Some visualization that relates the relative importance of different factors for LAZ vs. WLZ would be very helpful here. It would probably be better to have the same order of factors in each panel and not order each by effect size. Then the width of each panel can be larger and the effect sizes labeled.
 - ♣ The difference between "fixed reference" and "optimal intervention" is inadequately described.
 - ♣ Labels are too long and the relationship between factors and "optimal low-risk levels" is hard to follow. Suggest adding a table to describe and categorize all the factors and their corresponding "optimal low-risk levels" including which of the latter were empirically derived (and how) vs. selected by the authors.
 - ♣ I presume from the text that all the factors in figure 2 are univariate analyses? If so, how did the authors control for the correlation / collinearity between different factors in the list? If multivariate, what was the selection process?

♣ The legend states that the largest outliers are marked, but if so this labelling is unclear. Maybe different colors instead of different symbols would be helpful. What is special about them? How were they selected? Looking at the scatter, it appears there are many factors quite far from the line of equivalence.

12. In the univariate analyses, the authors do not appear to have controlled for comorbidity/mediation of growth failure. (i.e. how much of the stunting is actually underweight, etc). If the answer is that the only two variables considered were LAZ and WLZ, then that is inadequately explained as well. Was the excess risk to WAZ functionally assumed to be zero? How is that operationalized? See comment below on lines 362-4.

13. Line 319 – in my opinion, the entire structure of this analysis should be constructed in the way that this paragraph is. A) what are the predictors of LBW and/or stunting in <3 months, B) controlling for those, what are the additional predictors from 3-6 months, C) 6 months +, and D) predictors of recovery at various points. The initial analysis is sufficiently muddled in this way that it is hard to make sense of the findings in terms of policy or intervention significance.

14. Line 339 – again, doesn't this accelerate even more after 6 months and also after 2 years? What is the first derivative of the age pattern of incidence? If there is so much data on LAZ and WLZ, why reinforce the notion of categorical stunting, wasting, and stunting+wasting? Can't these be parsed? And turned into continuous functions? If no, why? Also, why a different end point (in time) for mortality, persistent wasting, and wasting+stunting?

15. Line 362-4 – two questions:

♣ underweight appears again for the first time in the results. Same question applies – how was the mediation effect calculated? Is underweight RR controlling for stunting and wasting? Was concurrent stunting and wasting regardless of age? What about persistent wasting after 6 months?

♣ What were the RRs for cause-specific mortality? Calculating all-cause mortality RRs are generally unreliable when comparing across populations with different cause structures. How did the authors control for cause composition of mortality? If they did not, what analysis supports the exclusion of this step?

16. There is a very long online github repo of secondary analyses, but few of any of them are described in terms of purpose of the sensitivity analysis and what the results of them were. Suggest adding text describing each, including purpose, interpretation, and implications of each.

17. See comments above. I found many of the conclusions to be imprecise because of multiple overlapping definitions of growth failure and lack of clear distinction between associations, sociodemographic/ social determinants, biologically-causative agents (the first time these are mentioned is in the discussion), mediators (not mentioned at all), and confounders (mentioned in discussion). In its current presentation, therefore, I feel confident that the authors identified important factors (and likely the correct ones), but do not have faith in the numerical quantities they present.

18. Lines 378-381: The text of this sentence is confused. Are the authors suggesting that CGF is a risk factor for death? Or an underlying cause of death? Why would lower mortality rates have affected RRs for CGF as a risk factor?

19. See #13 above. Need to redo the univariate analyses to control for other factors, including mediators in time and causality. Mother's height and LBW/preterm are the most obvious issues for after 6 months.

20. See #15 above: Need to clarify the analysis as being cause-specific RRs and/or redo the all-cause mortality RR analysis in a way that controls for cause composition.

21. See #4 above. Suggest continuing the analysis past age 2 as would presumably be supported by the data.

Author Rebuttals to Initial Comments:

Referees' comments:

Referee #1 (Remarks to the Author):

1) This paper addresses the risk factors and longer term health effects of wasting and stunting. Unlike the two descriptive papers, one can argue that the inferred associations represent etiological relationships and therefore are not affected by the issue of when/where the cohorts were enrolled. That being said, given the nature of risk factors considered and some of the results, there may be true (vs. merely stochastic) differences across cohorts which should be empirically assessed.

Response: For consistency with the other manuscripts in this series, we have updated the cohort inclusion criteria. In the revision, we excluded studies with a median birth year before 1990 and included smaller studies with fewer than 200 children, so we have limited the issue of when cohorts were enrolled. Because of the number of risk factors considered and the space limitations, we have not included cohort-specific estimates for many of the results and figures presented within the main manuscript, but they are available in the online supplement (<https://child-growth.github.io/causes/RR-forest.html>) to examine cohort-specific differences in estimates. We also added cohort-specific estimates to figures where possible (Figures 3c, 3d, 4b-d). Additionally, regionally pooled estimates show relatively similar results across the exposures with the strongest associations with child growth faltering (Extended data figures 7-8). The random-effects method of pooling parameters also leads to large confidence intervals when results are heterogenous across cohorts. While there is likely some true differences across cohorts, we believe that our primary conclusions, based on large effect sizes and narrow confidence intervals, accurately reflects the relationships between risk factors and child growth faltering within cohorts. An example is shown in extended data figure 1 for the relationship between child sex and stunting incidence.

2) A feature of the analysis – the inclusion of a large number of risk factors and application of flexible ML methods - which I assume was meant to be a strength of the paper may well have acted to weaken because of the number and nature of variables (risk factors) included.

The majority of risk factors analyzed here have been evaluated in retrospective or prospective cohorts and in some cases trials, in some cases with very detailed analysis to reveal temporal patterns (e.g. episodes of diarrhea by some consortium members). So unless the combination of data and flexible ML methods reveal some major unknown relationship the new knowledge generated is more limited than the two descriptive papers. In view of this reviewer, this is especially the case given findings from trials and quasi-randomized studies on both what works (e.g. Mexico's PROGRESA experiment) and what doesn't at least

in real-world conditions (e.g. some WASH trials and perhaps community-based zinc supplementation). The main result, on the role of pre-conception/pregnancy (via birth size), was shown a few years ago via the work of CHERG including by Dr Christian who is one of the authors, as has been the role of macro development conditions (arguably with more granularity where people have looked at within country variations during rapid growth or structural adjustment).

Response: The present analyses were designed to complement small, high resolution cohorts, which contribute valuable understanding but often are quite small (hundreds of children), thus potentially limiting their generalizability, and rigorous RCTs, which answer narrow questions well but cannot study all exposures of interest.

In our view, the present study makes three major, novel contributions:

(1) The analysis presents, to our knowledge, the first comprehensive synthesis to compare the relative influence of diverse child-, parent-, and household-specific exposures. As the referee notes, members of our team and the CHERG research consortium previously identified pre-conception and pregnancy-related risk factors to be large contributors to wasting and stunting by ages 12-60 months (Christian et al. 2013 <https://pubmed.ncbi.nlm.nih.gov/23920141/>). The importance of prenatal exposures herein is consistent with the CHERG findings, but the present analysis also identified many other important exposures, such as maternal anthropometry, birth order, and child sex not considered in previous pooled analyses. Providing a rank-ordered comparison of key exposures on the same scale, in the same synthesis, will be of great value to the global health community to help prioritize targets for intervention. The population intervention effect analyses presented in Figure 2 have focused on continuous LAZ and WLZ (rather than binary stunting or wasting), which is novel and reflects our findings in the companion papers that show child growth faltering is a continuous process and that a majority of children in the study populations fell below international growth standards, even if not classified as “stunted” or “wasted.” Past syntheses, including those from CHERG, focused on binary outcomes — our emphasis on CGF as a continuous process reflects an evolution in thinking in the field (e.g., <https://pubmed.ncbi.nlm.nih.gov/29546307/>). Further, our analyses also tries to understand early vs. later growth faltering and it’s underlying risk factors to better inform future research on interventions. For completeness and continuity with past research, we separately estimated the analogous effects on binary stunting and wasting outcomes in Extended Data Figures 6a and 6b. We also conduct novel analyses estimating the effects of exposures on stunting reversal and wasting recovery.

(2) Inclusion of a large set of longitudinal cohorts (97,988 children, 662,763 measurements) has enabled us to study effects of exposures on rare, but important outcomes including: persistent wasting, concurrent stunting and wasting, and mortality — which would be difficult or impossible to study in any single cohort or RCT. A past study used a similar approach to examine various measures of child growth faltering as risk factors for mortality (McDonald et al. 2013

<https://pubmed.ncbi.nlm.nih.gov/23426036/>). Our results with respect to mortality are largely confirmatory vis-a-vis McDonald et al. (2013), which we view as a strength since the present analyses included more contemporary cohorts — showing that the relationships still hold in current settings. Unlike McDonald et al. (2013), we present effects on mortality alongside several other novel findings, including effects on persistent wasting, concurrent wasting and stunting, and the longitudinal relationship between a child’s attained WLZ and subsequent linear growth velocity (all in revised Figure 4). To our knowledge, no other synthesis has examined the near-term consequences of CGF with this level of breadth or rigor.

(3) Combining information across diverse cohorts required innovation in our analysis approach. We would not expect the use of flexible machine learning (ML) in the analysis to reveal new, unknown relationships. However, the ML piece of the analysis enabled us to use a single analysis approach that could handle diverse data across cohorts. Some cohorts measured numerous covariates, while others were more limited (data availability summary in Figure 1). The statistical approach we used for variable selection, confounding adjustment, and pooling enabled us to maximally control for confounding within the limits of the data in a way that was flexible and agnostic, once we specified the causal structure of the exposure-outcome relationship. We expect this could serve as a template for other pooled analyses of diverse data sources, which are growing in application across fields far beyond nutrition.

We have noted these novel contributions in the first paragraph of the discussion on lines 346-356:

“This synthesis of LMIC cohorts during the first 1000 days of life has provided new insights into the principal drivers and near-term consequences of growth faltering. Our use of a novel, semi-parametric method to adjust for potential confounding provided a harmonized approach to estimate population intervention effects that spanned child-, parent-, and household-level exposures with unprecedented breadth (30 exposures) and scale (662,763 anthropometric measurements from 33 cohorts). Our focus on effects of shifting population-level exposures on continuous measures of growth faltering reflect a growing appreciation that growth faltering is a continuous process. Our results show children in LMICs stand to benefit from interventions to support optimal growth in the first 1000 days. Combining information from high-resolution, longitudinal cohorts enabled us to study critically important outcomes not possible in smaller studies or in cross-sectional data, such as persistent wasting and mortality, as well as examine risk-factors by age.”

In summary, we agree that the synthesis has not identified a large set of previously unconsidered risk factors for child growth faltering. But, the study’s novelty lies in its breadth and depth of investigation, the head-to-head comparison of very diverse exposures using the same population intervention effects scale, the pooled analyses across cohorts to study rarer outcomes that are of public health interest, and the methods that we developed to standardize and harmonize the statistical approach to do

the best job we could within the limits of the data. For these reasons, we feel that present study represents a significant advance for the field.

3) Methodologically, the large number of variables included has made the authors make choices about what to adjust for to avoid adjusting for potential mediators. If mediators were only affected by the variables in the model, then this would serve to estimate their total effects. However, if there is an unmeasured cause for the mediator, then excluding it from the model would lead to unmeasured confounding. I would recommend a full analysis with all variables, and then doing specific mediation analysis where relevant as some sort of additional materials.

Response:

We focused our primary analyses on unmediated effects because of our interest in total effects and because, conceptually, for population intervention effects we are attempting to capture all downstream effects — direct effects and mediated (indirect) effects — resulting from a population shift in an exposure.

We agree with the referee that confounding of mediators is a theoretical concern. However, controlling for additional covariates that only confound the mediator-outcome relationship can sometimes lead to unintended effects — namely collider bias, as we explain in more detail below.

For example, birth order could both affect fetal growth, leading to an increased risk of later growth faltering (CGF), or directly leading to a risk of growth faltering if being of

later birth order decreases food security or increases infectious disease risk from sibling. Household air pollution also affects birth weight and child growth faltering, but not birth order. We choose not to adjust the estimates of the associations between birth-order and child growth faltering for birth weight, because if household air pollution is a strong confounder of the birth weight and CGF relationship, and any unmeasured causes of low birth weight are weak confounders, then adjusting for birth weight will bias the birth order child growth faltering relationship by inducing an association between household air pollution and birth order.

See “To Adjust or Not to Adjust? When a “Confounder” Is Only Measured After Exposure”, (Groenwold et al. 2021, <https://www.ncbi.nlm.nih.gov/pmc/articles/PMC7850592/#!po=13.6364>) for a good discussion of the tradeoffs in deciding to adjust for mediators. Also see ‘Quantification of collider-stratification bias and the birthweight paradox” Whitcomb et al. 2009 <https://www.ncbi.nlm.nih.gov/pmc/articles/PMC2743120/> for a good example of the effects of collider bias. We have added this rationale for not controlling for mediators to the Methods section on lines 1003-1009:

“For each exposure, we used the directed acyclic graph (DAG) framework to identify potential confounders from the broader set of exposures used in the analysis.⁹ We did not adjust for characteristics that were assumed to be intermediate on the causal path between any exposure and the outcome, because while controlling for mediators may help adjust for unmeasured confounders in some conditions, it can also lead to collider bias. Detailed lists of adjustment covariates used for each analysis are available online (<https://child-growth.github.io/causes/dags.html>). Confounders were not measured in every cohort, so there could be residual confounding in cohort-specific estimates.”

The manuscript includes a mediation analysis for the effects of maternal anthropometry as it might be mediated through child’s birth size (see Extended Data Figure 5). This analysis was conducted to better understand if maternal anthropometry influences child growth faltering risk through at-birth biological versus SES factors or postnatal breastfeeding. Mediation estimates were slightly attenuated toward the null, and only in the case of maternal height and child LAZ were they statistically different from the primary analysis. These results imply that the causal pathway between parental anthropometry and child growth faltering operates through its effect on birth size, but most of the effect is through other pathways.

4) Substantively, the large number of variables included, possibly together with data issues or specific temporal relationships, leads to results that in some instances seem spurious. To be clear, this is an empirical analysis and the authors should be commended for reporting what they find. But previous in-depth work by some consortium members and others has shown the role of diarrhea, breastfeeding, food insecurity and persistent wasting which prove to have no or close to no effect here. Other variables, like wealth or birth order, have effects that are a bit “all over the place”. Why is this emerging? Is it a matter of

heterogeneity that when pooled leads to such results? Is it about temporal relationships? Is it some form of model mis-specification? Etc?

Response: Our estimation of attributable difference is the main underlying reason for some variation across related exposures and for potential differences with some past studies, as we describe in more detail below. Harmonization of exposure measures could also potentially contribute to attenuation of relationships, in cases where they required coarsening (averaging higher resolution data). Between-study heterogeneity and model mis-specification are unlikely culprits for reasons described in detail in Referee #2, comment #8 and Referee #4, comment #12.

An important distinction between the results in this paper and most past analyses is that we have estimated population intervention effects (sometimes called “population attributable differences”) and not the overall relative risk associated with specific exposures. As Referee 3 notes in his or her summary (below), population intervention effects are a policy relevant-parameter because they provide estimates of improvements in outcome that could be achievable through intervention (see this article for additional motivation: <https://pubmed.ncbi.nlm.nih.gov/28282339/>). Population intervention effects are a function of the difference between the unexposed and the exposed in a child’s anthropometry, as well as the prevalence of the exposure. Overall relative risks estimated in prior studies compare the risk among the exposed and unexposed. However, from a policy standpoint, the relevant counterfactual is not a population in which all children are unexposed, but rather, the current distribution of a risk factor in the population. Thus, the population intervention effect provides information about how much risk would change if an exposure was removed (or an intervention delivered) in a population compared to its current risk level. In general, population intervention effects will be attenuated compared to the overall relative risk in direct proportion to the percent of children who would have changed their exposure (see the Supplement of this recent paper from members of our team for a direct mathematical relationship between them, in section 2: <https://pubmed.ncbi.nlm.nih.gov/34151953/>). We have updated the manuscript from the previous version to refer to “population attributable differences” as “population intervention effects” throughout, and added the motivation to lines 222-227:

“In a series of analyses, we estimated population intervention effects, the estimated change in population mean Z-score if all individuals in the population had their exposure shifted from observed levels to the lowest-risk reference level. The PIE is a policy-relevant parameter; it estimates the improvement in outcome that could be achievable through intervention for modifiable exposures, as it is a function of the degree of difference between the unexposed and the exposed in a children’s anthropometry Z-scores, as well as the observed distribution of exposure in the population.”

For rare exposures, even if they are strong risk factors for disease, the population intervention effects can be close to zero. Patterns of estimates that vary across groups of related exposures (such as different dimensions of household wellbeing) arise, in large part, from differences in the proportion exposed. For example, 91% of children

would have their exposure changed to “number of rooms in the household shifted to 4+” whereas 31% of children would have their exposure changed to “improved sanitation”.

One limitation of the analysis, inherent with any synthesis across diverse studies, was the need to harmonize exposure definitions in a way that maximized their resolution within the limitations of the data across studies. For example, different questionnaire data on food availability were coded into low/medium/high food security, and instead of analyzing specific diarrhea episodes, we analyzed percent of measurements with diarrhea under 6 or 24 months of age because of differential measurement frequency across studies. A more in-depth analysis of timing of specific diarrheal episodes may find a stronger association with subsequent WAZ in some studies. But our findings are also consistent with cohort-specific findings (where available). For example, the MAL-ED consortium found no association between diarrhea and growth (<https://gh.bmj.com/content/2/4/e000370>), and interventions promoting exclusive breastfeeding in the first 6 months of life have not found to impact linear growth ([https://www.thelancet.com/journals/lancet/article/PIIS0140-6736\(15\)01024-7/fulltext](https://www.thelancet.com/journals/lancet/article/PIIS0140-6736(15)01024-7/fulltext)) Our results suggest that shifting the percentage of days <24 months to between 0-2% with diarrhea would modestly improve LAZ (PIE = +0.05z, 95% CI: 0.02 to 0.09), which we view as consistent (we acknowledge that and 7 of the 23 cohorts with diarrhea measured in this analysis are MAL-ED cohorts, so some consistency would be expected in this case).

In the revision, we added a more explicit caveat for this limitation in the Discussion:

Lines 369 - 391

“Previous studies have implicated prenatal exposures as key determinants of child growth faltering, and our finding of a limited impact of exclusive or predominant breastfeeding through 6 month (+0.01 LAZ) is congruent with a meta-analysis of breastfeeding promotion, but our findings of limited impact of reducing diarrhea through 24 months (+0.05 LAZ) contrast with some observational studies. We found that growth faltering before age 6 months puts children at far higher risk of persistent wasting and concurrent wasting and stunting at older ages (Fig 4c), which predispose children to longer-term morbidity and mortality. Our results agree with the limited success of numerous postnatal preventive interventions in recent decades, as well as evidence that improvements in maternal education, nutrition, parity, and maternal and newborn health care are primary contributors in countries that have had the most success in reducing stunting, reinforcing the importance of interventions during the window from conception to one year, when fetal and infant growth velocity is high. A recent study examining metabolism across the life span identified infancy as one of the highest periods of energy needs related to growth or development with energy expenditure by 1 year being about 50% above adult values.

The analyses had caveats. In some cases, detailed exposure measurements like longitudinal breastfeeding or diarrhea history were coarsened to simpler measures to harmonize definitions across cohorts, potentially attenuating their association with growth faltering. Other key exposures such as dietary diversity, nutrient consumption, micronutrient status, maternal and child morbidity indicators, pathogen-specific

infections, and sub-clinical inflammation and intestinal dysfunction were measured in only a few cohorts, so were not included. The absence of these exposures in the analysis, some of which have been found to be important within individual contributed cohorts, means that our results emphasize exposures that were more commonly collected, but likely exclude some additional causes of growth faltering..”

5) As a specific question, given the mother’s height and BMI are both in the model, why was weight included?

Response: In our analyses of mother’s height, weight, and BMI as exposures, we did not adjust for the other 2 anthropometry measures in each analysis because of their interdependence (e.g., BMI adjusted for weight would effectively measure the inverse association between height and the outcome). In analyses of exposures beyond mother’s height, weight and BMI, where these anthropometric measures were adjusted covariates (not exposures themselves), we allowed for all three covariates to be considered. The use of an ensemble approach allowed us to be more agnostic about variable inclusion and selection, which we believe is a strength for analytic methods that are applied across a very large number of cohorts. It enabled us to specify a single causal model for each exposure, and then run the same analysis method across all cohorts, rather than needing to develop a tailored model for each individual cohort, which might differ depending on covariates measured and empirical associations in each dataset. Models included in the cross-validated ensemble used variable selection steps that would drop a highly collinear covariate, in particular LASSO penalized regression and gradient boosting. For members of the ensemble library without such variable selection, such as linear regression, potential collinearity would affect coefficients on variables that were collinear, but not the key exposure of interest. Additionally, in instances of a rare outcome (like severe wasting in a smaller cohort), we included only 1 covariate per 10 cases of the outcome, chosen by rank ordering the deviance ratios. Including all three variables ensured we selected the one most strongly associated with the outcome to maximize our control of confounding within constraints.

6) The mortality analysis largely confirms the results of Reference 1 and its companion paper that looked at severity of stunting and all-cause and cause-specific mortality. As a specific questions: I would have thought that the analysis would be based on time-to-event design and using a Cox model, perhaps with exposure status updated at each follow up (as done in the previous pooled analysis of cohorts for all-cause and cause-specific mortality) since there is nothing particular about dying before two years of age versus later in childhood. Is there a particular data reason to deviate from this?

Response: These are all good points. In the revision, we enhanced the mortality analysis to use a time-to-event analysis to ensure that the analyses presented in this synthesis used an analysis approach that was consistent with the McDonald et al. (2013) analysis (reference 1). We used Cox proportional hazard models, with exposure status updated at each follow-up, using all measures of growth from birth to 2 years. In

addition to the switch from targeted maximum likelihood estimation to proportional hazard models for estimation, this led to several other changes in the analytic approach. Instead of using the cumulative incidence of types of child growth faltering before 6 months of age, the exposure was the last anthropometry measurement before death or end of follow-up, and exposures were classified into mutually-exclusive measures of child growth faltering (e.g. only wasted, wasted and stunted, wasted and underweight, etc.). Results were largely similar, though stunting and severe stunting alone were not associated with mortality. Previously, the cumulative incidence of stunting was associated with mortality, but many of these children were likely also wasted or underweight. Combined measures of child growth faltering were all significantly associated with increased mortality, though being underweight alone was a stronger risk factor than being underweight and wasted or stunted.

We view it as a strength of the present analyses that the mortality results largely confirm those from McDonald et al. (2013), since we have relied on a completely separate set of nine more contemporary cohorts in this study. The mortality analyses are one piece of the present study that seeks to summarize the consequences of early life child growth faltering. Additional novel results show that among children who survive, growth faltering before age 6 months puts them at far higher risk of persistent wasting and concurrent wasting and stunting at older ages — all severe forms of child growth faltering that predispose children to longer-term morbidity and, for some, mortality.

In the updated time-to-event analyses, we revised the Results and Methods:

Results, Lines 334-343:

“Finally, we estimated hazard ratios (HR) of all-cause mortality by 2 years of age associated with measures of growth faltering within eight cohorts that reported ages of death, which included 1,689 child deaths by age 24 months (2.4% of children in the eight cohorts). Included cohorts were highly monitored, and mortality rates were lower than in the general population in most cohorts (Extended Data Table 3). Additionally, data included only deaths that occurred after anthropometry measurements, so many neonatal deaths may have been excluded, and without data on cause-specific mortality, some deaths may have occurred from causes unrelated to growth faltering. Despite these caveats, growth faltering increased the hazard of death before 24 months for all measures except stunting alone, with strongest associations observed for severe wasting, stunting, and underweight (HR=8.7, 95% CI: 4.7, 16.4) and severe underweight alone (HR=4.2, 95% CI: 2.0, 8.6) (Fig 4d).”

Methods, Lines 1047-1061

“Mortality analyses estimated hazard ratios using Cox proportional hazards models with a child’s age in days as the timescale, adjusting for potential confounders, with the CGF exposure status updated at each follow-up that preceded death or censoring by age 24 months. CGF exposures included moderate (between $-2 z$ and $-3 z$) wasting, stunting, and underweight, severe (below $-3 z$) wasting, stunting, and underweight, and

combinations of concurrent wasting, stunting, and underweight. CGF failure categories were mutually exclusive within moderate or severe classifications, so children were classified as only wasted, only stunted, or only underweight, or some combination of these categories. We estimated the hazard ratio associated with different anthropometric measures of CGF in separate analyses, considering each as an exposure in turn with the reference group defined as children without the deficit. For children who did not die, we defined their censoring date as the administrative end of follow-up in their cohort, or age 24 months (730 days), whichever occurred first. Because mortality was a rare outcome, estimates are only adjusted for child sex and trial treatment arm. To avoid reverse causality, we did not include child growth measures occurring within 7 days of death. Extended Data Table 3 lists the cohorts used in the mortality analysis, the number of deaths in each cohort, and a comparison to country-level infant mortality rates.”

Referee #2 (Remarks to the Author):

This is a very impressive and detailed paper on child growth failure in LMICs, based on a large database of 35 cohorts. The results on age-varying effects on growth failure are fascinating. I only have two major comments and a few minor comments.

Major comments

1) Have all children been consistently followed from birth to their second birthday in the 8 cohorts analyzed for child mortality? Which are these cohorts? I was surprised by the small share of neonatal deaths in these cohorts (about 20% of neonatal deaths among deaths that occurred in the first 2 years). According to UN IGME estimates (childmortality.org), neonatal deaths accounted for 46% of all under-five deaths in LMICs and 62% of all infant deaths in 2019. I was expecting a much higher percentage of neonatal deaths among those occurring in the first 2 years (somewhere in between 46% and 62%). Is this a sign of underreporting of neonatal deaths (and corresponding births) in the cohorts? Or could this be due to the fact that early neonatal deaths were not in the analysis as the children did not have anthropometry measurements? This could be added in a note for Extended Data Figure 10.

Response: The lower percentage of neonatal mortality is due to many neonatal deaths not being included in the analysis because they did not have anthropometry measurements. The cohorts with mortality information and sufficient deaths measured children quarterly, and not monthly. Some of the included cohorts did not measure neonatal infants, and other studies enrolled pregnant women, but may not have measured growth prior to neonatal death. There may be under-reporting of neonatal deaths in some of the included cohorts if enrolled children who died during the neonatal period were not included in the shared datasets, as among neonatal deaths a large

proportion occur in the first 24 hours of life and 2/3 in the first week, and so many neonatal deaths were likely to occur before the first anthropometry measurement and therefore not be included in the shared datasets. Additionally, the overall mortality in the cohorts was lower than the national mortality rates, probably because of the highly-monitored nature of the cohorts (see table below). We have dropped Extended Data Figure 10 based on Peer review comment #14, but we have noted the lack of anthropometry measures prior to neonatal mortality.

Lines 334-343:

“Finally, we estimated hazard ratios (HR) of all-cause mortality by 2 years of age associated with measures of growth faltering within eight cohorts that reported ages of death, which included 1,689 child deaths by age 24 months (2.4% of children in the eight cohorts). Included cohorts were highly monitored, and mortality rates were lower than in the general population in most cohorts (Extended Data Table 3). Additionally, data included only deaths that occurred after anthropometry measurements, so many neonatal deaths may have been excluded, and without data on cause-specific mortality, some deaths may have occurred from causes unrelated to growth faltering. Despite these caveats, growth faltering increased the hazard of death before 24 months for all measures except stunting alone, with strongest associations observed for severe wasting, stunting, and underweight (HR=8.7, 95% CI: 4.7, 16.4) and severe underweight alone (HR=4.2, 95% CI: 2.0, 8.6) (Fig 4d).”

We also added a new Extended data table 3 to list out the cohorts used in the mortality analysis, the number of deaths recorded in each cohort, and comparisons to national infant mortality rates.

2) I find it strange that you report a relative risk of death in the first 24 months associated with persistent wasting under 6 months, as children would need to survive for a sufficiently long time to be measured at least 4 times. Would it be preferable to report on all-cause mortality rates between month 6 and month 24, based on anthropometric measures taken in the first 6 months?

Response: We agree that the estimates of associations between persistent wasting and mortality from the previous mortality analysis may have been influenced by survivor bias due to children needing to survive long enough to be measured four times. In response to this comment and to Referee #1, comment #6 (above), we have updated the mortality analysis to a time-to-event analysis using Cox-PH models, and so have dropped persistent wasting as a child growth faltering exposure. See our response to Referee #1, comment #6 for a description of how the Methods and Results changed.

As a supplemental analysis, we estimated associations between the cumulative incidence of measures of child growth faltering under 6 months (including persistent

wasting) and all-cause mortality after 6 months and included a figure of the results in the online supplementary material (<https://child-growth.github.io/causes/mortality.html>).

Minor comments

Summary

3) Why do the authors state that the higher risk of illness and mortality (or the association with child growth failure) contributed to the SDG 2.2 goal in the summary? This is a bit unclear.

Response: We agree that the phrasing was unclear and we removed it in the revised summary.

4) Could you specify in the abstract that you are analyzing trajectories until age 2?

Response: We have added the age range of children we analyzed to the summary.

Lines 157 -160

“Here, we show using a population intervention effects analysis of 33 longitudinal cohorts (83,671 children) and 30 separate exposures that improving maternal anthropometry and child condition at birth, in particular child length-at-birth, could lead to population increases by age 24 months in length-for-age Z of 0.04 to 0.40 and weight-for-length Z by 0.02 to 0.15.”

5) The title refers to low- and middle-income countries, but the summary only mentions low-income settings.

Response: We have included middle income countries in the summary.

Lines 151 - 155

“Growth faltering (low length-for-age or weight-for length) in the first 1000 days — from conception to two years of age — influences both short and long-term health and survival. Evidence for interventions to prevent growth faltering such as nutritional supplementation during pregnancy and the postnatal period has increasingly accumulated, but programmatic action has been insufficient to eliminate the high burden of stunting and wasting in low- and middle-income countries.”

Introduction

6) "WASH interventions, which aim to reduce childhood infections... in non-emergency settings". Why indicate in non-emergency-settings?

Response: We agree that emergency vs. non-emergency settings is not a key distinction in the context of overall causes of child growth faltering. We have removed it in the revision.

7) Pooled longitudinal analyses. Persistent wasting is defined as >50% measurements of WLZ <-2 during an age period with at least 4 measurements. But unless I missed it, the paper does not provide the mean of the interval between measurements. I was wondering if these 4 measurements were typically spread over a long or short period.

Response: The median interval between measurements was 80 days (IQR: 62-93, added on line 906). The measurement interval varied based both on the measurement designs of different cohorts, and on variation in the actual visit ages of different children.

8) Age-varying effects on growth failure. Can you provide some explanation as to why children with birth order 2+ have a lower risk of stunting and wasting at less than 6 months of age? What are the possible explanations? Could it be that first-born children are born to younger mothers?

Response: We have added one possible explanation to the Results, but elucidating the true mechanism for this effect modification isn't possible within the limits of the data. Firstborn children were more likely to have younger mothers (61% of firstborn children had mothers <20 years old compared with 3% of children with birth order of 2+). For children with birth order 3+, 40% were born to mothers 30 years or older. However, the analysis adjusted for maternal age when estimating the association between birth order and growth faltering, so to the degree that statistical adjustment could control for the effect of maternal age the current analyses have done so.

In response to this query, we searched the literature on this specific topic. Our results are broadly consistent with previous assessments, though a caveat is that we judge the past literature on this topic to be more limited than the present analysis, both in terms of scope and risk of bias. An unadjusted meta-analysis of parity found that firstborn children were at higher risk of low birth weight and being small-for-gestational age, which is consistent with our finding of higher wasting/stunting in the first 6 months (<https://pubmed.ncbi.nlm.nih.gov/20583931/>). Additionally, cross-sectional analyses of household surveys found higher rates of child growth faltering in older children with later birth order in Bangladesh, India, Nepal (<https://www.ncbi.nlm.nih.gov/pmc/articles/PMC4254892/>; https://papers.ssrn.com/sol3/papers.cfm?abstract_id=3362642), and Ethiopia (<http://citeseerx.ist.psu.edu/viewdoc/download?doi=10.1.1.627.9668&rep=rep1&type=pdf>). Lastly, it may be that mothers who have multiple children could be a healthier subgroup of mothers.

We have proposed one potential mechanism in the revised text:

Lines 299 - 303

“The specific mechanism for effect modification of birth order on growth faltering by age is unknown, but primiparous mothers may be younger, have lower pre-pregnancy weight, have lower weight gain during pregnancy, or have less experience breastfeeding — a key source of nutrition during the first 6 months — while children with older siblings could have lower quality and quantity of complementary foods compared with firstborn children in food insecure households.”

9) Consequences of early growth failure "we estimated the relative risk of mortality across measures of growth failure in the first 6 months within eight cohorts that reported mortality endpoints, including 2,510 child deaths by age 24 months". This sentence could be reformulated. At first, I thought it was referring to mortality in the first 6 months, I had to read twice to understand that mortality was counted up to age 24 months, while the 6 months refer to growth failure.

Response: We have updated this sentence as we have changed the approach to a time-to-event analysis. The revised results are now:

Lines 334-343:

“Finally, we estimated hazard ratios (HR) of all-cause mortality by 2 years of age associated with measures of growth faltering within eight cohorts that reported ages of death, which included 1,689 child deaths by age 24 months (2.4% of children in the eight cohorts). Included cohorts were highly monitored, and mortality rates were lower than in the general population in most cohorts (Extended Data Table 3). Additionally, data included only deaths that occurred after anthropometry measurements, so many neonatal deaths may have been excluded, and without data on cause-specific mortality, some deaths may have occurred from causes unrelated to growth faltering. Despite these caveats, growth faltering increased the hazard of death before 24 months for all measures except stunting alone, with strongest associations observed for severe wasting, stunting, and underweight (HR=8.7, 95% CI: 4.7, 16.4) and severe underweight alone (HR=4.2, 95% CI: 2.0, 8.6) (Fig 4d).”

10) Extended Data Figure 10 is difficult to interpret and not very informative as we can't see the anthropometry measurements for those who survived.

Response: We removed Extended Data Figure 10 in the revision.

Referee #3 (Remarks to the Author):

Summary of the key results

Originality and significance: The breadth of work undertaken in this last paper of this series is considerable and reveals important findings that are well situated in

the previous literature. The population attributable difference was a useful measure of association to use for this type of an analyses because there are important policy relevant implications in how this can be communicated to policy makers and program implementers. The population level predictors of LAZ and WLZ at 24 months identified reflect that both nutrition-sensitive and specific interventions are warranted to shift those population level attributes. Also, noting that the PAF was relatively modest for the factors identified as significant possibly signals to the need for cost effectiveness of certain interventions as it relates to improving nutritional outcomes. Not that cost-effectiveness should drive choice of an intervention for improved nutrition alone but that it is definitely a factor to consider. Other important findings include the relationship between persistent wasting and subsequent linear growth failure and mortality and the timing of these relationships.

Response: Thank you for this accurate summary and comments regarding the study's implications.

Data & methodology:

1) The methods portion of this manuscript has been undertaken with much care and detail. One missing element was the DAGs eluded to in linked 796-798 to assess adjustments to the models. Can these please be included in the supplementary online material?

Response: We have added example DAGs to the online supplement. The original DAGs were hand-drawn due to complexity across the measured exposures, and the large number of exposures considered. We have added example DAGs and full lists of the adjustment covariates used for each exposure here: (<https://child-growth.github.io/causes/dags.html>)

2) Lines 316-318: Did the authors find this to be the case for growth velocity as well?

Response: We have provided additional results with respect to growth velocity in the paper's online supplement (<https://child-growth.github.io/causes/velocity.html>). We examined height, weight, LAZ, and WAZ velocity in 3-month intervals and found that maternal anthropometry is generally not associated with growth velocity, consistent with the mostly parallel growth trajectories shown in Figure 3, except for associations between maternal height and early growth.

Mother's height was strongly associated with all 4 measures of growth velocity in the first 3 months of life. Taller mothers gave birth to infants who grew faster (PIE: 0.007 [95% CI: 0.002, 0.013] for LAZ velocity. This is also noticeable in Figure 3a, where growth faltering as measured by change in LAZ is lower in the first 3 months for children of taller mothers than for children of shorter mothers. After 3 months, maternal height had non-significant or very small effects on LAZ, WAZ, height, and weight velocities.

Mother's weight had inconsistent associations with LAZ and length velocity, was not associated with WAZ velocity, but was associated with significantly increased weight velocities before 9 months and marginally significantly increased weight velocities at older ages. Higher maternal BMI was associated with an increased weight velocity before 6 months, but not with other ages or measures of growth velocity.

3) Lines 324-325: How do authors explain the finding of birth order and its associations with growth failure before and after 6 mos?

Response: Please see our response to earlier Referee #2, comment #8, which addressed this same question.

4) Conclusions: Given the modest effects of some of these factors on reducing the incident of growth failure, especially maternal nutritional status, wouldn't language around the preconception and pregnancy interventions to promote optimal growth need to be tempered? What are the tradeoffs here from a more practical standpoint of application by policy makers and practitioners?

Response: Our interpretation and framing of the results was that overall, the effects were modest but nevertheless provide crucial guidance toward future targets for intervention. Our estimates that shifting the most influential population exposures, such as maternal nutritional status, would prevent 20-30% of incident stunting and wasting and improve Z-scores by 0.04-0.4 Z in the study populations — at the population level— could influence many millions of children in LMICs. In the Discussion, we have tried to appropriately frame the magnitude of effects whilst not over-promoting specific interventions:

First two paragraphs of the Discussion, Lines 346 - 368

"This synthesis of LMIC cohorts during the first 1000 days of life has provided new insights into the principal drivers and near-term consequences of growth faltering. Our use of a novel, semi-parametric method to adjust for potential confounding provided a harmonized approach to estimate population intervention effects that spanned child-, parent-, and household-level exposures with unprecedented breadth (30 exposures) and scale (662,763 anthropometric measurements from 33 cohorts). Our focus on effects of shifting population-level exposures on continuous measures of growth faltering reflect a growing appreciation that growth faltering is a continuous process. Our results show children in LMICs stand to benefit from interventions to support optimal growth in the first 1000 days. Combining information from high-resolution, longitudinal cohorts enabled us to study critically important outcomes not possible in smaller studies or in cross-sectional data, such as persistent wasting and mortality, as well as examine risk-factors by age. We found that maternal, prenatal, and at-birth characteristics were the strongest predictors of growth faltering across regions in LMICs. Many of these predictors, like child sex or season, are not modifiable but could guide interventions that

mitigate their effects, such as seasonally targeted supplementation or enhanced monitoring among boys. Shifting several key population exposures (maternal height or BMI, education, birth length) to their observed low-risk level would improve LAZ and WLZ in target populations and could be expected to improve Z-scores by 0.06 to 0.4 Z in the study populations and prevent 8% to 32% of incident stunting and wasting (Fig 2, Extended Data Fig 6). Maternal anthropometric status strongly influenced birth size, but the parallel drop in postnatal Z-scores among children born to different maternal phenotypes was much larger than differences at birth, indicating that growth trajectories were not fully “programmed” at birth (Fig 3a-b).

Previous studies have implicated prenatal exposures as key determinants of child growth faltering, and our finding of a limited impact of exclusive or predominant breastfeeding through 6 months (+0.01 LAZ) is congruent with a meta-analysis of breastfeeding promotion, but our findings of limited impact of reducing diarrhea through 24 months (+0.05 LAZ) contrast with some observational studies. We found that growth faltering before age 6 months puts children at far higher risk of persistent wasting and concurrent wasting and stunting at older ages (Fig 4c), which predispose children to longer-term morbidity and mortality. Our results agree with the limited success of numerous postnatal preventive interventions in recent decades, as well as evidence that improvements in maternal education, nutrition, parity, and maternal and newborn health care are primary contributors in countries that have had the most success in reducing stunting, reinforcing the importance of interventions during the window from conception to one year, when fetal and infant growth velocity is high. A recent study examining metabolism across the life span identified infancy as one of the highest periods of energy needs related to growth or development with energy expenditure by 1 year being about 50% above adult values.”

Clarity and context:

5) Lines 39-41: High attributable risk from prenatal causes, and severe consequences for children who experienced early growth failure, support a focus on pre-conception and pregnancy as key opportunities for new preventive interventions. - This sentence needs to be rephrased. High attributable risk of what?

Response: We have clarified:

Lines 164 - 166

“The importance of prenatal causes, and severe consequences for children who experienced early growth faltering, support a focus on pre-conception and pregnancy as key opportunities for new preventive interventions.”

6) Lines 185-186: Many of the statements included in the first paragraph of the introduction included in this sentence reflects the evidence referenced as causal when this is not necessarily the case. Would encourage more careful epidemiologic language to be used here giving careful consideration to the

measures of association used in the references included when stating existing relationships between stunting/wasting and long-term consequences.

Response: We have revised the language in the first paragraph of the Introduction. The revised paragraph is now:

Lines 169-175

“Child growth failure (CGF) in the form of stunting, a marker of chronic malnutrition, and wasting, a marker of acute malnutrition, is common among young children in low-resource settings, and may contribute to child mortality and adult morbidity.^{1,2} Worldwide, 22% of children under age 5 years are stunted and 7% are wasted, with most of the burden occurring in low- and middle- income countries (LMIC).³ Current estimates attribute > 250,000 deaths annually to stunting and > 1 million deaths annually to wasting.² Stunted or wasted children also experience worse cognitive development⁴⁻⁹ and adult economic outcomes.¹⁰”

7) Given the richness of the data presented and analyses conducted, the abstract is rather limited in showcasing the depth of the work. While there is a word limitation here, would reconsider reframing to make the abstract more compelling.

Response: We have revised the Abstract, attempting to make it more compelling given the word limit:

“Growth faltering (low length-for-age or weight-for length) in the first 1000 days — from conception to two years of age — influences both short and long-term health and survival. Evidence for interventions to prevent growth faltering such as nutritional supplementation during pregnancy and the postnatal period has increasingly accumulated, but programmatic action has been insufficient to eliminate the high burden of stunting and wasting in low- and middle-income countries. In addition, there is need to better understand age-windows and population subgroups in which to focus future preventive efforts. Here, we show using a population intervention effects analysis of 33 longitudinal cohorts (83,671 children) and 30 separate exposures that improving maternal anthropometry and child condition at birth, in particular child length-at-birth, could lead to population increases by age 24 months in length-for-age Z of 0.04 to 0.40 and weight-for-length Z by 0.02 to 0.15. Boys had consistently higher risk of all forms of growth faltering than girls, and early growth faltering predisposed children to subsequent and persistent growth faltering. Children with multiple growth deficits had higher mortality rates from birth to two years than those without deficits (hazard ratios 1.9 to 8.7). The importance of prenatal causes, and severe consequences for children who experienced early growth faltering, support a focus on pre-conception and pregnancy as key opportunities for new preventive interventions.”

Referee #4 (Remarks to the Author):

1) Based on the content of the study focusing on maternal anthropometrics and sociodemographic details, the title of “causes... of child growth failure” is a bit misleading. By my read, with the possible exceptions of breastfeeding differences, maternal anthropometrics, and rainy season, the authors have not explored biologic pathways of causality. Only the first is potentially modifiable (in any short period of time). While this doesn’t negate their findings, it does call for a tempering of language and perhaps reconsideration of the title.

Response: We agree that charting a complete biological mechanism between causes and effects is ideal, but in epidemiology we must often settle for a gradual accumulation of evidence toward that ideal. Often, evidence is sufficient with respect to exposures and outcomes long before the mechanisms are understood. For example, the mechanism for how mass azithromycin distribution reduces all cause mortality in high mortality settings is not completely understood, but most scientists accept that the relationship is causal, based on evidence from randomized controlled trials (e.g., <https://pubmed.ncbi.nlm.nih.gov/29694816/>); Additional evidence following the original trial has begun to elucidate the biological mechanism (e.g., <https://pubmed.ncbi.nlm.nih.gov/31981558/>), but the mechanism remains poorly understood. Yet, even without a complete understanding of mechanism, WHO has presented guidelines for countries to consider the intervention as a temporary measure to promote child survival (<https://www.who.int/publications/i/item/9789240009585>). A similar example from the nutrition field is maternal calcium supplementation to prevent pre-eclampsia and preterm birth in LMICs (<https://pubmed.ncbi.nlm.nih.gov/30277579/>).

A cause that is difficult or impossible to modify, such as season or child sex, does not negate the causal relationship. Instead, if a causal relationship exists for unmodifiable exposures it forces us to consider interventions that might mitigate their effects, such as seasonally targeted supplementation, or enhanced monitoring among boys. We note this on lines 357-360 of the discussion:

“We found that maternal, prenatal, and at-birth characteristics were the strongest predictors of CGF across regions in LMICs. Many predictors, like child sex or season, are not modifiable but they could guide interventions that might mitigate their effects, such as seasonally targeted supplementation or enhanced monitoring among boys.”

Nevertheless, we acknowledge the results presented herein are observational analyses — inference is not based on randomized, controlled trials. In response to this suggestion we have checked carefully the language throughout and have been careful to temper the use of causal language and limit its use only where it is most appropriate. We have also updated the title to: “Risk factors and impacts of child growth faltering in low- and middle-income countries” to temper the causal language in the title.

2) See comments below under “conclusions”. My overall takeaway is that careful

analysis of these data is extraordinarily valuable, and the authors have made a good first pass. However, there is such a mixture of concepts, approaches, principles, and methods used in this paper that it is really hard to compare the insights either to each other or to previous “knowledge.”

Response: See our response below to Referee #4 comments #8, 12, 13, and 14.

3) This is an important paper and an important topic. With some focused work in clarification of language and harmonization of conceptual approach across all the presented analyses, this could be an important contribution to child health globally.

Response: Thank you, we have updated the language throughout the manuscript to better clarify our conceptual and statistical approaches.

4)) An interim point at 1000 days is reasonable, but why stop at 24 months? Growth failure continues to accumulate/ accelerate in older children (as suggested by figure 3) so this choice seems odd.

Response: We agree that growth faltering continues to accumulate beyond age 24 months. The BMGF ki data repository focused its efforts on the period of the first 1000 days, and so nearly all cohorts that met our inclusion criteria stopped measurement at age 24 months — the few studies that measured children beyond age 24 months measured a small subset of children. Because of the smaller and different population of children older than 24 months, we restricted our analyses to children ≤ 24 months. Below are the histograms of child ages from each of the included cohorts. The vertical line in each panel marks age 24 months.

In these histograms, the y-axis shows the number of measurements by child age in days among the 27 studies (33 cohorts when examining country-specific MAL-ED sites) with monthly or quarterly measurements included in the analysis. The vertical line

marks 24 months (730 days). All observations to the left of the vertical line were included in our analysis.

5) Was the outcome only AT 24 months? Or anytime IN the first 24 months of life? Please clarify throughout.

Response: Timing of outcomes varied across analyses. We have clarified the timing of outcome measurement in every figure and throughout the manuscript. All figures clearly label the age of outcome measurement, or in the case of cumulative incidence measures, the age period over which we summarized incidence.

For example, population intervention effects on LAZ and WLZ (Figures 2a, 2b) used z-scores measured at 24 months. The revised title is: *“Population intervention effects of child, parental, and household exposures on length-for-age z-scores and weight-for-length z-scores at age 24 months.”*

6) Line 221 + 229 – this goes from “millions” of children to 94k. Why the huge drop off?

Response:

The original sentence referred to millions of children included in cross-sectional datasets and data from high-income and historic cohorts. We have now revised this sentence:

Lines 204-206

“Cohorts were assembled as part of the Bill & Melinda Gates Foundation's Knowledge Integration (ki) initiative, which included studies of childbirth, growth and development in the first 1000 days, beginning at conception.”

7) Line 238 – why asymmetry in “unrealistic” measurement truncation? +/-6 vs. +6/-5 vs. +/-5. WHO recommendations on this do not have empirical basis. How many children were eliminated by this? It is a small % overall, but could be a large proportion of the outcomes since these are theoretically the most at-risk subgroups.

Response: The WHO recommendations vary by Z-score anthropometry measurement, based on thresholds beyond which anthropometry is not compatible with human life, though of course there are true outliers beyond those cutoffs. We agree with the comment that the WHO cutoffs for outlier exclusion do not have a strong empirical basis, but we used the standard WHO cutoffs for consistency with other studies and previously reported results from the included cohorts. Using the WHO cutoffs, we dropped 1,190 (0.2%) unrealistic LAZ measurements ($>+6$ or <-6 Z), 1,330 (0.2%) WAZ measurements ($>+5$ or <-6 Z), and 1,670 (0.3%) WLZ measurements ($>+5$ or <-5 Z).

The small percentage of measurements excluded would not influence the conclusions. By using these cutoffs, we believe that we excluded more erroneous anthropometry measurements than true extreme measurements.

In the revision, we have reported the number and proportion of children excluded in the Methods section:

Lines 891 - 901

“We calculated length-for-age Z-scores (LAZ), weight-for-age Z-scores (WAZ), and weight-for-length Z-scores (WLZ) using WHO 2006 growth standards.¹ We used the medians of triplicate measurements of heights and weights of children from pre-2006 cohorts to re-calculate Z-scores to the 2006 standard. We dropped 1,190 (0.2%) unrealistic measurements of LAZ (> 6 or < -6 Z), 1,330 (0.2%) measurements of WAZ (> 5 or < -6 Z), and 1,670 (0.3%) measurements of WLZ (> 5 or < -5 Z), consistent with WHO recommendations.² See Benjamin-Chung (2020) for more details on cohort inclusion and assessment of anthropometry measurement quality.³ We also calculated the difference in linear and ponderal growth velocities over three-month periods. We calculated the change in LAZ, WAZ, length in centimeters, and weight in kilograms within 3-month age intervals, including measurements within a two-week window around each age in months to account for variation in the age of each length measurement.”

8) Line 250 – I conceptually disagree with this assertion. Maternal height SHOULD be adjusted for BW. The authors statement is correct, but this makes BW a mediator of stunting. If, as they claim later, that length trajectories AFTER birth are no longer related to maternal height, this further supports separating out into direct causation vs. mediation pathway.

Response: In the revision we have included a mediation analysis that adjusts for birth anthropometry (Extended Data Figure 5) in the parental anthropometry analyses. Mediation estimates were attenuated toward the null by a median of 31.5%. These results imply that the causal pathway between parental anthropometry and child growth faltering operates through its effect on birth size, but most of the effect is through other pathways. See also our response to Referee #1, comment #3, above.

9) Line 260 – what covariates? What was the source of them?

Response: We have updated this sentence to clarify that the low-risk level is estimated from the same set of covariates as the primary analyses, selected via directed acyclic diagrams to avoid adjusting for mediators:

Lines 242 - 244

“We also estimated the effects of optimal dynamic interventions, where each child’s individual low-risk level of exposure was estimated from potential confounders (details in Methods).”

See our response below to Referee #4 Comment 11 for additional description of the optimal intervention parameter and for the updated Methods text.

10) Line 274 – breastmilk *quality*? How was this measured? And standardized?

Response: Studies did not include measurements of breastmilk quality. We offered breastmilk quality as one possible factor that might differ between mothers with different anthropometry (in that anthropometry likely integrates many different factors that influence child growth). In the revision, we shortened this section and have no longer included the sentence to which the reviewer refers.

11) Figure 2 – multiple comments/ questions. Overall I find this figure quite difficult to decipher.

Very hard to follow because the ordering of the factors in each panel are in different order. Some visualization that relates the relative importance of different factors for LAZ vs. WLZ would be very helpful here. It would probably be better to have the same order of factors in each panel and not order each by effect size. Then the width of each panel can be larger and the effect sizes labeled.

Response: We have updated Figure 2 based on these suggestions. We have ordered both panels by groups of characteristics and then shown the LAZ effect size, so that the order of factors is the same. We have also moved the comparison of fixed vs. optimal intervention to Extended Data Figure 9, with more clearly-labelled points and additional explanation in the figure legend.

The difference between “fixed reference” and “optimal intervention” is inadequately described.

Response: We have moved the comparison of fixed vs. optimal intervention to Extended Data Figure 9, with more clearly-labelled points and additional explanation in the figure legend (also see our response above to Referee #4 comment #9). All optimal risk levels were empirically determined on a child level via the approach detailed on lines 985-994 of the Methods section. See our response below to Referee #4 comment #12 and the following Methods text for an explanation of the estimation approach:

“Optimal individualized intervention impact We employed a variable importance measure (VIM) methodology to estimate the impact of an optimal individualized intervention on an exposure.⁷ The optimal intervention on an exposure was determined through estimating individualized treatment regimes, which give an individual-specific

rule for the lowest-risk level of exposure based on individuals' measured covariates. The covariates used to estimate the low-risk level are the same as the adjustment documented in section 6 below. The impact of the optimal individualized intervention is derived from the VIM, which is the predicted change in the population-mean outcome from the observed if every child's exposure was shifted to their optimal level. This differs from the PIE and PAF parameters in that we did not specify the reference level, and the reference level could vary across participants. PIE and PAF parameters assume a causal relationship between exposure and outcome. For some exposures we considered attributable effects have a pragmatic interpretation — they represent a summary estimate of relative importance that combines the exposure's strength of association and its in the population.⁸ Comparisons between optimal intervention estimates and PIE estimates are shown in Extended Data Fig 9."

□ Labels are too long and the relationship between factors and “optimal low-risk levels” is hard to follow. Suggest adding a table to describe and categorize all the factors and their corresponding “optimal low-risk levels” including which of the latter were empirically derived (and how) vs. selected by the authors.

Response: In the revised figure, we enhanced the exposure labels and reference levels. The row labels serve as a table that describes the reference level chosen for each exposure, with additional columns for effect size (95% CI), sample size and the percentage of children whose exposure was shifted to estimate population intervention effects. We moved the “optimal intervention” estimates previously presented in insets to extended data (see our response to the following comments).

□ I presume from the text that all the factors in figure 2 are univariate analyses? If so, how did the authors control for the correlation / collinearity between different factors in the list? If multivariate, what was the selection process?

Response: All of the estimates in Figure 2 are fully adjusted for all measured confounders, per the Methods section. We have clarified this in the main text and in the caption of Figure 2.

Main Text, Lines 222 - 232

“In a series of analyses, we estimated population intervention effects, the estimated change in population mean Z-score if all individuals in the population had their exposure shifted from observed levels to the lowest-risk reference level. The PIE is a policy-relevant parameter; it estimates the improvement in outcome that could be achievable through intervention for modifiable exposures, as it is a function of the degree of difference between the unexposed and the exposed in a children's anthropometry Z-scores, as well as the observed distribution of exposure in the population. We selected exposures that were measured in multiple cohorts, could be harmonized across cohorts for pooled analyses, and had been identified as important predictors of stunting or

wasting in prior literature (Fig 1, Extended Data Table 2). Different cohorts measured different sets of exposures, but all estimates were adjusted for all other measured exposures that we assumed were not on the causal pathway between the exposure of interest and the outcome (Fig 1).”

Title and caption:

Figure 2 | Population intervention effects of child, parental, and household exposures on length-for-age z-scores and weight-for-length z-scores at age 24 months.

(a) Population intervention effects on child length-for-age z-scores (LAZ) at age 24 months.

(b) Population intervention effects on child weight-for-length z-scores (WLZ) at age 24 months.

Exposures were rank ordered in both panels by effects on LAZ. Each exposure label includes the reference level used to estimate population intervention effects, shifting exposures for all children from their observed exposure to that level. Cohort-specific estimates were adjusted for all measured confounders using ensemble machine learning and TMLE, and then pooled using random effects (Methods). Columns for each exposure summarize the number of children that contributed to each analysis and the percentage of children for whom exposure was shifted to the reference level, and the estimated population intervention effect (PIE) and 95% confidence interval.

□ **The legend states that the largest outliers are marked, but if so this labelling is unclear. Maybe different colors instead of different symbols would be helpful. What is special about them? How were they selected? Looking at the scatter, it appears there are many factors quite far from the line of equivalence.**

Response: Revised Figure 2 no longer includes the scatter plots. In Extended Data Figure 9, we provide an enhanced version of the scatter plot that compares our primary analysis with fixed reference levels and the analysis that used a data-adaptive “optimal” intervention rule for each child. In the revision, we do not mark outliers.

12) In the univariate analyses, the authors do not appear to have controlled for comorbidity/ mediation of growth failure. (i.e. how much of the stunting is actually underweight, etc). If the answer is that the only two variables considered were LAZ and WLZ, then that is inadequately explained as well. Was the excess risk to WAZ functionally assumed to be zero? How is that operationalized? See comment below on lines 362-4.

Response: We regret if the language in the initial submission caused confusion regarding statistical adjustment in the analyses. All estimates presented in the paper were adjusted for measured confounders; none were univariate analyses. We have clarified this in the revised manuscript, both in the main text and in the Methods.

We did not adjust for child underweight or WAZ in analyses of stunting or LAZ, because we would expect that a child's weight could partially mediate relationships between exposures and linear growth. When selecting confounders for each exposure, we intentionally excluded mediators because we were interested in the total effect between exposures and measures of child growth faltering, not the direct effects after adjusting for all variables downstream on the causal pathway. Our analyses in Figure 2 have focused on attributable effects estimated to occur were we to change the population's exposure to some counterfactual level. In other words, the effects include all downstream effects of the exposure, both direct and indirect (i.e., mediated). See our responses above to Referee #1 comment #3 and Referee #4 comment #8 for additional details regarding mediation analyses.

The population intervention effects presented in main text figures focus on LAZ and WLZ (Figure 2). We also applied the approach to the binary outcomes of stunting and wasting and presented the population attributable fraction estimates for each exposure in Extended Data Figures 6a and 6b. We included continuous outcomes in the main text figures because our results in the companion articles demonstrate the continuous process of child growth faltering. Even among those not stunted or wasted, the vast majority of children in the study populations had z-scores <0 and would thus potentially benefit from reductions in harmful exposures (or conversely benefit from increasing beneficial exposures).

We did not functionally assume that excess risk due to WAZ was zero, but we did not separately present population intervention effects on WAZ for parsimony due to limited space in the manuscript and because together, LAZ and WLZ capture the two main dimensions of child growth faltering (LAZ: linear growth faltering from more chronic malnutrition; WLZ: wasting from more acute malnutrition) which are of key interest as outcomes for population monitoring. Weight is the sum of lean tissue, fat tissue, water and length of skeleton. A child's weight can be low for their age because their length is low even if they have a healthy weight for their size, or weight can be low because of low fat or muscle mass, or from a combination of low height and muscle/fat mass. WAZ, therefore, includes information from (and therefore substantially overlaps with) LAZ and WLZ, and WAZ is a combination of LAZ and WLZ. In responses to Referee comments on the companion paper focused on wasting and concurrent stunting (Referee 3, comment 16b of that paper), we describe how just 2% of children experience underweight alone in the absence of stunting or wasting. We have included results for WAZ in supplemental materials for completeness (<https://child-growth.github.io/causes/>).

13) Line 319 – in my opinion, the entire structure of this analysis should be constructed in the way that this paragraph is. A) what are the predictors of LBW and/or stunting in <3 months, B) controlling for those, what are the additional predictors from 3-6 months, C) 6 months +, and D) predictors of recovery at

various points. The initial analysis is sufficiently muddled in this way that it is hard to make sense of the findings in terms of policy or intervention significance.

Response: Tradeoffs are inevitable in analyses that pool information across diverse sources. In our study, we present a series of analyses that approach the question of key causes of child growth faltering from different angles, each with advantages and disadvantages. The approach that the referee has suggested is well-conceived but presumes that most predictors were measured in most cohorts. In reality, our analysis synthesizes information over a patchwork of exposure measures (Figure 1) and is tailored to get the most policy-relevant estimates within this constraint.

Our analyses of population intervention effects synthesize the relative contribution of different exposures to attained growth at age 24 months, controlling for all pre-exposure confounders measured in each cohort **before pooling** cohort-specific estimates. The rationale for conducting the analysis in this way, rather than a series of conditional analyses as the referee has suggested, was that it enabled us to include the most information from the most cohorts. If we were to condition analyses in each age period based on predictors of growth faltering in earlier age periods, the varying exposure measurements across cohorts (Figure 1) would mean that analyses of later periods would be limited to cohorts that jointly measured those predictors. For example, if we were to study predictors from ages 3-6 months, conditional on birth order and mother's height (both important predictors of early growth faltering), we would be limited to just 6 cohorts that measured both of those covariates — excluding data from the 21 other cohorts.

Beyond the initial, overall summary, we did feel it would be informative to drill down and study age-specific effects — as highlighted by the referee in this comment — which we completed where possible and have summarized in Figure 3c.

14) Line 339 – again, doesn't this accelerate even more after 6 months and also after 2 years? What is the first derivative of the age pattern of incidence? If there is so much data on LAZ and WLZ, why reinforce the notion of categorical stunting, wasting, and stunting+wasting? Can't these be parsed? And turned into continuous functions? If no, why? Also, why a different end point (in time) for mortality, persistent wasting, and wasting+stunting?

Response: The Referee's comments refers to the sentence that, in turn, points to results from the two companion articles: "*We documented high incidence rates of wasting and stunting from birth to age 6 months.*" In the revised linear growth paper (Benjamin-Chung et al.) we enhanced the summary of incident stunting and reversal by age, demonstrating that stunting incidence was substantially higher before 3 months

and was relatively flat thereafter; in other words, its first derivative approached zero by age 6 months (Figure 4b in the linear growth faltering revision, reproduced here):

Figure 4b from Benjamin-Chung et al. (in review). Incidence proportion of new stunting, stunting relapse, and stunting reversal by age. The black line presents estimates pooled using random effects with restricted maximum likelihood estimation. Colored lines indicate cohort-specific estimates

As to why we have included categorical results for stunting and wasting in the analysis, the rationale was to maintain continuity with previous studies and policy guidelines like the Sustainable Development Goals which focus on these indicators. Reducing the continuous Z-scores to categories focuses the analysis on the lower end of the Z-score distributions, which is important, but at the cost of losing information (in the continuous outcome) and potentially ignoring the growth faltering that is present among most children in the analysis who are not technically stunted or wasted but are below international growth standards. However, we also estimated risk of mortality and measures of serious child growth faltering by combinations of categorical wasting/stunting/underweight. The results have important programmatic implications, since these cutoffs are used to identify high-risk children and enroll them into treatment or monitoring programs.

We did not feel that any further transformation of the data would yield new insights beyond the use of continuous Z-scores, as we have presented in the main analyses. Introducing new transformations could confuse readers and policy makers, who are generally familiar with Z-scores and the categorical outcomes (wasting, stunting).

We used different endpoints (in time) for mortality, persistent wasting, and wasting + stunting because of different definitions of each outcome and data availability. Persistent wasting was defined longitudinally (here, between 6 and 24 months of age to avoid overlap with the exposures of early child growth faltering before 6 months). We chose to summarize persistent wasting from 6 to 24 months because it enabled us to identify children who had consistently low WLZ across repeated measures, and it enabled us to study the effect of very early child growth faltering, before 6 months, on this subsequent outcome. Concurrent wasting and stunting was defined cross-sectionally at 18 months because most concurrent stunting and wasting was manifest by that age (per results in the companion paper “Wasting and concurrent stunting...”)

and because many cohorts had measurements at that age, so we could include the most data possible in the analysis. The updated mortality analysis in the revision uses Cox proportional hazards models (see response to Referee #1, comment #6) with proximate anthropometry measurements before death or censoring through 24 months to maximize the available data, as mortality is a rare outcome.

15) Line 362-4 – two questions:

15.1) underweight appears again for the first time in the results. Same question applies – how was the mediation effect calculated? Is underweight RR controlling for stunting and wasting? Was concurrent stunting and wasting regardless of age? What about persistent wasting after 6 months?

Response: Our response to Referee #1, comment #3, explains why we focused on total effects, which do not condition on causal intermediates (mediating variables). RR estimates for each anthropometric measure do not control for other anthropometric measures. Extremely limited word space meant that at times, such as here, we omitted some details in the main text for, say, age ranges of the outcome, but those details are presented in the figures, to which we refer in each case. For example, concurrent stunting and wasting were assessed at age 18 months — an age chosen because most concurrent stunting and wasting was manifest by that age (per results in the companion paper “Wasting and concurrent stunting...”) and because many cohorts had measurements at that age, so we could include the most data possible in the analysis. Details about the age are in the X-axis label of Figure 4c, and the detailed methods and rationale for the choice are in the Methods.

Lines 911-918

“For child morbidity outcomes (Figure 4c), concurrent wasting and stunting prevalence at age 18 months was estimated using the anthropometry measurement taken closest to age 18 months, and within 17-19 months of age, while persistent wasting was estimated from child measurements between 6 and 24 months of age. We chose 18 months to calculate concurrent wasting and stunting because it maximized the number of child observations at later ages when concurrent wasting and stunting was most prevalent, and used ages 6-24 months to define persistent wasting to maximize the number of anthropometry measurements taken after the early CGF exposure measurements.”

We did not control underweight for stunting and wasting as WAZ is a function of both WLZ and LAZ, so collinearity is a problem.

15.2) What were the RRs for cause-specific mortality? Calculating all-cause mortality RRs are generally unreliable when comparing across populations with different cause structures. How did the authors control for cause composition of mortality? If they did not, what analysis supports the exclusion of this step?

Response: The mortality analysis (now updated to a time-to-event analysis to estimate hazard ratios, see our response to Referee #1, comment #6) used all-cause mortality. Cause-specific mortality was unavailable in many cohorts (9 of 12 studies), and our reliance on all-cause mortality could have included some deaths unrelated to CGF, such as accidents. Mortality is a very rare outcome and one which, we feel, exemplifies the value of pooling information across individual cohorts which, on their own, would not typically be sufficiently powered to study the relationship on their own. Although cause structures could differ across the cohorts included in the analysis, we accounted for that variation when we estimated pooled hazard ratios by including random effects for each cohort. In the revised Figure 4d, we included cohort-specific estimates to show the heterogeneity (with the caveat that, since mortality is a rare outcome, each cohort's estimate will show much uncertainty). We included as a limitation of the study the absence of cause-specific mortality information:

Lines 336-340:

“Included cohorts were highly monitored, and mortality rates were lower than in the general population in most cohorts (Extended Data Table 3). Additionally, data included only deaths that occurred after anthropometry measurements, so many neonatal deaths may have been excluded, and without data on cause-specific mortality, some deaths may have occurred from causes unrelated to growth faltering.”

16) There is a very long online github repo of secondary analyses, but few of any of them are described in terms of purpose of the sensitivity analysis and what the results of them were. Suggest adding text describing each, including purpose, interpretation, and implications of each.

Response: We have added text at the top of each page of the online repository describing the purpose of each additional set of analyses or figures, including purpose, interpretation, and implications of each (<https://child-growth.github.io/causes/>)

17) See comments above. I found many of the conclusions to be imprecise because of multiple overlapping definitions of growth failure and lack of clear distinction between associations, sociodemographic/ social determinants, biologically-causative agents (the first time these are mentioned is in the discussion), mediators (not mentioned at all), and confounders (mentioned in discussion). In its current presentation, therefore, I feel confident that the authors identified important factors (and likely the correct ones), but do not have faith in the numerical quantities they present.

Response: See our response above to Reviewer # 1, comment #2.

18) Lines 378-381: The text of this sentence is confused. Are the authors

suggesting that CGF is a risk factor for death? Or an underlying cause of death? Why would lower mortality rates have affected RRs for CGF as a risk factor?

Response: We revised this sentence to more clearly explain the potential limitations of the mortality analyses and their likely influence on estimates. We avoided the topic of whether child growth faltering was a direct cause vs. underlying cause of death because the present analyses would not provide any empirical evidence one way or another regarding that distinction):

Lines 336 - 340

“Included cohorts were highly monitored, and mortality rates were lower than in the general population in most cohorts (Extended Data Table 3). Additionally, data included only deaths that occurred after anthropometry measurements, so many neonatal deaths may have been excluded, and without data on cause-specific mortality, some deaths may have occurred from causes unrelated to growth faltering.”

19) See #13 above. Need to redo the univariate analyses to control for other factors, including mediators in time and causality. Mother’s height and LBW/preterm are the most obvious issues for after 6 months.

Response: Analyses were adjusted for measured confounders in each cohort, and then adjusted estimates were pooled across cohorts. Please see our responses above to Referee #1 comment #3 and Referee #4 comment #8.

20) See #15 above: Need to clarify the analysis as being cause-specific RRs and/or redo the all-cause mortality RR analysis in a way that controls for cause composition.

Response: The analysis of child mortality uses all-cause mortality, because only 3 of 12 studies with mortality information have cause of death information. See our response above to Referee #4, comment #15.2 for additional details.

21) See #4 above. Suggest continuing the analysis past age 2 as would presumably be supported by the data.

Response: Please see our response above to Referee #4, comment #4, which provides a detailed response to this query.

Reviewer Reports on the First Revision:

Referees' comments:

Referee #1 (Remarks to the Author):

Like the stunting and wasting papers, the paper has been systematically revised. My comments below focus on presentation, interpretation and contextualization in the light of metrics presented and especially the fact that the data used for causal inference (which is different from the stunting and wasting papers' descriptive scope) are observational.

First, I think the paper should move away from presentation of population intervention effect (which has now been made more explicit) and only present measures of risk that are independent of exposure. As the authors know, population intervention effect depends on both the level of exposure in the cohort and in the size of the causal association regardless of how simple or complex the function describing the latter is. The former is only relevant to the cohort studied and has no generalizability; the latter, presumably reflects some mechanism of risk that can be generalized. population intervention effect combines these two and hence can be interpreted as: in a group of children, at a specific place and time in the past, with some unspecified level of various exposures, population intervention effect was X%. For readers familiar with the metric, this is unhelpful; for those unfamiliar, it can be misleading as they may not realize the role of the exposure component. If they author want to show some sort of etiological fractions, they can do so at hypothetical exposures (e.g., in a population where X% of people have this level of exposure, like those in Country/Region X in 2020, etc).

As a secondary issue, I suggest replace population intervention effect with population attributable fraction because the former, even though understood by causal inference epidemiologists to imply specific inference procedures, could easily be interpreted by the great majority of readers as the effect a real (vs. analytically constructed) intervention.

Second, while the authors are using causal inference methodology, such methods depend entirely on what confounders were assumed to be relevant and which ones had data available – leading to the usual concerns about residual confounding. Therefore the authors should systematically and comprehensively put their results in the context of current knowledge (a model that medical journals have developed over the years so that epidemiological studies do not overlook all the other evidence). Currently this is done briefly, in my view far too briefly, in the last paragraph of P. 9. The right way to do this is to have a table that has a row for each risk factor and presents the association estimated in this study together with a summary for the following types of evidence (separately from one another): trials, other prospective studies (other data/other methods compared to what is used here) and retrospective cohort studies when exposure is unlikely to have changed in relation to outcome (e.g., in DHS, it is impossible or highly unlikely for parity, type of toilet or parental education to change in response to a child's growth). I realize this will involve additional work but for the sake of good science and policy, this sort of contextualization is necessary and can overcome the concerns that we all have about extent of residual confounding and model dependence in observational causal inference. I also realize how studies report things (measures of risk, age groups, etc) are not always comparable but it should be the authors' responsibility to synthesize across these, as well do when we report etiological effects.

Related to the above, and for transparent reporting, the papers needs an appendix table that shows for each exposure: the variables adjusted for and those that were not adjusted for on the premise that they were mediators. This would be a summary of the DAG for each risk factor (I am assuming from the text that there was a separate DAG for each risk factor; if this was not the case, the text needs to be made for explicit). I leave the review of multivariate application of targeted maximum likelihood estimation to reviewers with expertise in causal inference methods.

Finally, the statement on pre-natal vs. post-natal opportunities for growth (P. 9) is far stronger than supported by this and other studies, and should be toned down. If this were true, at the extreme there is little to be done for growth catch up once a child reaches post-natal period. But programs like the Dutch school milk programs have shown that catch up is possible throughout the entire childhood and adolescence. The findings, and their implications, should be systematically discussed in context of our broader knowledge (among others, see Prentice et al Am J Clin Nutr 2013, and Georgiadis and Penny Lancet Public Health 2017).

Analytically, the way to make the evidence on how pre-natal and post-natal periods matter more robust and relevant, and to link this paper with the two descriptive papers, is to do separate/stratified analysis of risk factors by birth size (clearly identifying those risk factors that are expected to act pre- and post-birth): what are the risk factors for growth catch up/non-catch up in children with small birth size? And what are the risk factors for growth failure/non-failure for children with healthier birth size? (this is different from Figure 4 which treats early infancy failure as a risk factor; rather it would involve analyzing the effects of other risk factors, conditional on status in early infancy).

As a specific methodological issue, why impute using median/mode, for example compared to a multiple imputation approach? I suggest that the statistical reviewer is consulted on the implications of this. Regardless of the appropriateness of the method, for transparency there should be a table that shows extent of missingness by cohort and variable. Also for transparency, a complete case analysis should be presented as a sensitivity analysis.

Referee #2 (Remarks to the Author):

The detail and depth of the analysis are exemplary. Thank you for addressing my questions. I don't have additional comments, except that I spotted some typos in the revision:

Line 361: highlights should be plural.

Line 897-899: The sentence "We also calculated the difference in linear and ponderal growth velocities over three-month periods." appears twice.

Line 1022: " is show bin"

Referee #3 (Remarks to the Author):

Overall, the authors have adequately addressed issues I, as well as other reviewers have raised. The only pending points from my read are:

Lines 294-295: Authors do not provide a clear direction/ magnitude in this hypothesis.

Lines 399-405: There is a heavy focus on the conclusion the conclusions of the other papers.

Sex, specifically boys being consistently associated with being stunted, wasted or both, both as pooled estimate and across most study and regional contexts, was a finding of consequence and one that finds itself in the summary but is not flushed out as a finding at all throughout the paper. The authors may chalk this up to the fact that it is not modifiable but discussing this finding meaningful seems essential given the findings prominence.

Referee #4 No remarks to the Author

Referee #5 (Remarks to the Author):

I would like to congratulate the authors on a tour de force of data harmonization, data analysis, and interpretations. What makes this work novel is that the underlying methodological framework allows for (almost) direct conclusions and comparisons across diverse cohorts. This would not have been possible without a statistical analytical framework, which separates sampling and data modelling, from interpretation. The former will be cohort depended while the latter can be universal. To the best of my knowledge this is the first time – within the subfield – such cross-cutting data analysis has been completed.

A single methodological concern is if the cohorts have been “collected for a reason” and therefore perhaps are less representative for the country/population. This could be further addressed. In the methods section is it also somewhat unclear – at least for this reader – which values covariates are set to when computing population effects. Are they from the literature or derived from the cohorts? It seemed to be a mix.

In short: the basic scientific merit and methodology is top-notch.

The presentation is fairly verbal. It is of course a matter of taste if this is preferable. I would have preferred a few of the key-findings had been highlighted through tables. Instead the table material is mixes both summaries and very detailed. Given the amount of sensitivity analyses etc (which is of course good) it will be very difficult for the average reader to really take in the quantitative results. This is regrettable as one of the strengths of the paper exactly is the quantifications. In short: I suggest to make it more clear what the key tables are and where is more for the dedicated reader.

The paper finds that the most important aspects causing missed growth are biological – and very hard to change. This has the obvious effect that whilst important biologically, it does really help for prevention. This is only partly acknowledged. It is said that the insight can be used to find out targets for interventions, but since no intervention can ever deliver the kind of changes hypothesized in the paper, it could be that the importance of certain aspects are over-sold. In short: The population effect reported in the paper should in some sense be multiplied by a factor (smaller than 1) corresponding to “ease of intervention”.

Author Rebuttals to First Revision:

Referees' comments:

Referee #1 (Remarks to the Author):

1. Like the stunting and wasting papers, the paper has been systematically revised. My comments below focus on presentation, interpretation and contextualization in the light of metrics presented and especially the fact that the data used for causal inference (which is different from the stunting and wasting papers' descriptive scope) are observational.

Response: Thank you for your continued, constructive input to improve the paper. Your comments have led to additional refinements (detailed below), which we feel have substantially improved the presentation and interpretation of the results.

2. First, I think the paper should move away from the presentation of population intervention effect (which has now been made more explicit) and only present measures of risk that are independent of exposure. As the authors know, population intervention effect depends on both the level of exposure in the cohort and in the size of the causal association regardless of how simple or complex the function describing the latter is. The former is only relevant to the cohort studied and has no generalizability; the latter, presumably reflects some mechanism of risk that can be generalized. population intervention effect combines these two and hence can be interpreted as: in a group of children, at a specific place and time in the past, with some unspecified level of various exposures, population intervention effect was X%. For readers familiar with the metric, this is unhelpful; for those unfamiliar, it can be misleading as they may not realize the role of the exposure component. If they author want to show some sort of etiological fractions, they can do so at hypothetical exposures (e.g., in a population where X% of people have this level of exposure, like those in Country/Region X in 2020, etc).

Response: Thank you for sharing this perspective. We agree that presenting effects that are independent of the population's exposure level would provide valuable, additional results. We revised the central results presented in Figure 2 (now Fig 2 for LAZ, Fig 3 for WLZ) to display the average treatment effects for each exposure studied, along with the population intervention effect (PIE) estimated at a counterfactual level of exposure as in the previous version. We believe this higher resolution of information provides readers with a more complete view into the analysis and allows for a direct view into each exposure's independent association with child LAZ and WLZ. In discussion of the

revised figure, the research consortium elected to remove mode of delivery (C-section, vaginal birth) as an exposure because the policy and public health implications of any result would be unclear. Instead, we replaced it with small for gestational age (SGA), which integrates gestational age and birth weight — whilst providing a small bit of redundancy with gestational age and birth weight, SGA presents a key, biologically meaningful exposure that has clear public health interpretation and implications.

In the development of the analysis plan for this project, we discussed at length the relative merits of presenting different parameters of interest. We have presented Population Intervention Effects (PIE) because they are widely considered to be a more accurate depiction of the population-level importance of a risk factor. There are examples where observational analyses that estimate PIE more closely align with results from randomized trials because the effects are closer to what a real-world intervention could expect when delivered to the general population (one recent example from members of our team related' to sanitation and child growth: Rogawski-McQuade et al. 2022 <https://pubmed.ncbi.nlm.nih.gov/34151953/>). Additionally, the PIE parameters are analogous to other large-scale syntheses, such as the Global Burden of Disease project at the Institute for Health Metrics and Evaluation, which uses population attributable fractions for binary outcomes (also see response to next comment), and are generally regarded as most informative in the translational step of moving from estimated effects (or associations) to expected impacts of intervening on exposures in actual populations (Westreich et al. 2016 <https://www.ncbi.nlm.nih.gov/pmc/articles/PMC4880276/> , Westreich 2017 <https://pubmed.ncbi.nlm.nih.gov/28282339/>).

As the referee notes, the PIE is a function of the prevalence of exposure in the population. An idea we discussed at the planning stage of the study was to use exposure distributions from outside data sources, such as the Demographic and Health Surveys (DHS) — however, such surveys do not include most of the exposures that were of interest in this study, so standardizing exposure distributions using DHS was impossible. Since this analysis draws from a large number of cohorts across diverse populations we felt that PIE estimates would be meaningful even if based on the empirical distribution of the exposures in the cohorts.

We added this caveat to the Discussion:

Lines 363-367

“Population intervention effects were based on exposure distributions in the 33 cohorts, which were not necessarily representative of the general population in each setting. Use of external exposure distributions from population-based surveys would be difficult because many key exposures we considered, such as at-birth characteristics or longitudinal diarrhea prevalence, are not measured in such surveys.”

- 3. As a secondary issue, I suggest replace population intervention effect with population attributable fraction because the former, even though understood by causal inference epidemiologists to imply specific inference procedures, could**

easily be interpreted by the great majority of readers as the effect a real (vs. analytically constructed) intervention.

Response: Thank you for this suggestion. We have presented population attributable fractions for binary outcomes. The population intervention effect (PIE) is a generalization of the population attributable fraction and allows for any type of outcome. Since our primary inference focused on continuous measures of child growth (LAZ, WLZ) we have used “population intervention effect” to be consistent with the terminology widely used in epidemiology (Westreich 2017 <https://pubmed.ncbi.nlm.nih.gov/28282339/>). When estimating comparable effects for binary outcomes (stunting, wasting), the effects are labeled “population attributable fractions” (Extended Data Figs 6, 7).

- 4. Second, while the authors are using causal inference methodology, such methods depend entirely on what confounders were assumed to be relevant and which ones had data available – leading to the usual concerns about residual confounding. Therefore the authors should systematically and comprehensively put their results in the context of current knowledge (a model that medical journals have developed over the years so that epidemiological studies do not overlook all the other evidence). Currently this is done briefly, in my view far too briefly, in the last paragraph of P. 9. The right way to do this is to have a table that has a row for each risk factor and presents the association estimated in this study together with a summary for the following types of evidence (separately from one another): trials, other prospective studies (other data/other methods compared to what is used here) and retrospective cohort studies when exposure is unlikely to have changed in relation to outcome (e.g., in DHS, it is impossible or highly unlikely for parity, type of toilet or parental education to change in response to a child’s growth). I realize this will involve additional work but for the sake of good science and policy, this sort of contextualization is necessary and can overcome the concerns that we all have about extent of residual confounding and model dependence in observational causal inference. I also realize how studies report things (measures of risk, age groups, etc) are not always comparable but it should be the authors’ responsibility to synthesize across these, as well do when we report etiological effects.**

Response: We agree that contextualization is important. A comprehensive, systematic review of the literature is valuable for any scientific report. That level of review is feasible when a study focuses on a single exposure or intervention — the typical scenario for many medical journal articles, in our experience. However, this study has assessed 30 different exposures with respect to linear growth faltering and weight-for-length. Conducting a systematic review of the literature, synthesizing it, and reporting results for each of 30 exposures and multiple outcomes is, in our view, beyond the scope of this paper. In response to this suggestion, we have updated Extended Data Table 2 to summarize key evidence for the exposures studied in this analysis. The evidence we summarized is not the result of a systematic review on each question (though we have

listed recent reviews where present). We conducted a literature search for each risk factor and, based on the available evidence, preferentially report findings from meta-analyses of trials, individual trial results, meta-analyses of cohorts, or pooled demographic and health survey analyses. After reviewing the summary, we made comparisons of effect estimates from the present study to previous studies where possible. Most of the previous reports on this topic were not from randomized studies, included older children on average than our study, and were from cross-sectional surveys or smaller cohorts, so we believe prior findings cannot be used to assess the extent of residual confounding or model dependence in this study. Additionally, there was substantial variation in study design, populations, exposure definitions, analytic methods, and specific comparisons, so we note specific differences between prior research and our study that limit comparability for each risk factor. This heterogeneity in the scientific record was a key motivation for this study's novel synthesis of 33 cohorts.

- 5. Related to the above, and for transparent reporting, the papers needs an appendix table that shows for each exposure: the variables adjusted for and those that were not adjusted for on the premise that they were mediators. This would be a summary of the DAG for each risk factor (I am assuming from the text that there was a separate DAG for each risk factor; if this was not the case, the text needs to be made for explicit). I leave the review of multivariate application of targeted maximum likelihood estimation to reviewers with expertise in causal inference methods.**

Response: Thank you for this suggestion. This paper includes a lot of online supplemental information and might have not been clearly linked in the text. The table the referee has requested is available in the online supplement, and we have now more clearly linked to this in the Methods: <https://child-growth.github.io/causes/dags.html>

Regarding a separate DAG for each risk factor, the team of faculty and postdocs that underwent the exercise at Berkeley did so on paper and white board through in-person sessions, and the main record from the consensus discussion was the covariate lists summarized in the table referenced above. We included an example DAG to illustrate the process in the online supplement along with the table (link above), but recreating the other 29 DAGs electronically would delay the revision substantially and we felt would not meaningfully improve validity assessment or reproducibility of the analysis.

- 6. Finally, the statement on pre-natal vs. post-natal opportunities for growth (P. 9) is far stronger than supported by this and other studies, and should be toned down. If this were true, at the extreme there is little to be done for growth catch up once a child reaches post-natal period. But programs like the Dutch school milk programs have shown that catch up is possible throughout the entire childhood and adolescence. The findings, and their implications, should be systematically discussed in context of our broader knowledge (among others, see Prentice et al Am J Clin Nutr 2013, and Georgiadis and Penny Lancet Public Health 2017).**

Response: Thank you for this suggestion. We agree and have revised the discussion in two ways. First, we clarified that our focus has been on the first 24 months but that does not preclude the potential for consequences (and interventions) at older ages:

Lines 375-380

“A final caveat is that we studied consequences through age 24 months — the primary age range of contributed ki cohort studies — and thus have not considered effects on longer-term outcomes. Several studies have suggested that puberty could be another potential window for intervention to enhance catch-up growth. Improving girls’ stature at any point through puberty could help blunt the intergenerational transfer of growth faltering by improving maternal height, which in turn could improve outcomes among their children (Figs 2, 3, 4a, 4b).”

Second, we have distinct language in the revised discussion that speaks to this point:

Lines 358-362

“Maternal anthropometric status strongly influenced birth size, but the parallel drop in postnatal Z-scores among children born to different maternal phenotypes was much larger than differences at birth, indicating that growth trajectories were not fully “programmed” at birth (Fig 4a-b). This accords with the transition from a placental to oral nutrient supply at birth.”

Lines 385-388

“A stronger focus on prenatal interventions should not distract from renewed efforts for postnatal prevention. The observed pre- and postnatal growth faltering we observed reinforce the need for sustained support of mothers and children throughout the first 1000 days.”

- 7. Analytically, the way to make the evidence on how pre-natal and post-natal periods matter more robust and relevant, and to link this paper with the two descriptive papers, is to do separate/stratified analysis of risk factors by birth size (clearly identifying those risk factors that are expected to act pre- and post-birth): what are the risk factors for growth catch up/non-catch up in children with small birth size? And what are the risk factors for growth failure/non-failure for children with healthier birth size? (this is different from Figure 4 which treats early infancy failure as a risk factor; rather it would involve analyzing the effects of other risk factors, conditional on status in early infancy).**

Response: In response to the referee’s suggestion, we have provided estimates of the relationship between each exposure and child LAZ or WLZ at 24 months, stratified by birthweight. In doing so, we reduced exposure levels so that they were binary to make the comparison clearer and to help ensure there would be sufficient observations in each

strata given the additional level of stratification by birthweight. For example, rather than consider maternal education categories of high, medium, and low, we compared (medium or high) vs low. A caveat to the analysis is that only 7 cohorts measured birthweight, and so estimates below are based on measurements from between 3 and 7 cohorts, depending on which measured each exposure of interest. Not all exposures were measured in this subset of cohorts.

Our broad conclusion from this analysis is that there is no substantial difference in measures of association between each exposure and child z-scores at 24 months among children with different birth status. Note that unlike stratifying by a confounder, the pooled estimates do not necessarily fall between the two stratified estimates for prenatal exposures. This is because birthweight is a mediator for them, and the complex causal structure can create unintuitive — and potentially less reliable — results, which we will discuss in more detail below.

Figure. Adjusted difference in length-for-age z-score (LAZ) and weight-for-length z-scores (WLZ) at 24 months, overall (unstratified) and stratified by child birthweight.

When developing the analysis plan for this project, we had several discussions amongst our team about whether to pursue this specific analysis. In the end, we did not for two reasons.

First, the analysis is complex in that birthweight is a likely mediator for many of the prenatal exposures we considered. Stratified estimates above thus represent *direct effects* measured in a model allowing for exposure-mediator interaction. The focus of our overall analysis has been on *total effects*, which include the overall effect through both direct and indirect (mediated) pathways. Capturing the overall effect is, in our view, most relevant when rank-ordering exposures based on population intervention effects because it reflects the expected impact in the study populations from a shift in the exposure.

Second, there is a long history in perinatal epidemiology of difficulties related to birthweight-stratified analyses — so much so, that it is often called the “birth weight paradox,” the structural basis for which was first described in 2006 as a form of collider-stratification bias (<https://pubmed.ncbi.nlm.nih.gov/16931543/>). Collider stratification bias results if there are unmeasured confounders of the mediator (birth weight) and the outcome (child LAZ at 24 months) — often leading to paradoxical results. This bias has been observed so often when conditioning on birthweight in studies of prenatal exposures that a member of this paper’s consortium wrote in 2009 it “...has provided food for thought, and papers for publication, for at least two generations of perinatal epidemiologists” (<https://pubmed.ncbi.nlm.nih.gov/19689490/>). There is reason to suspect this type of bias is present in the above analyses. There are several examples where the stratified (direct) effects, are larger than the pooled (total) effect. Child sex is one example — since the pooled effect is likely unconfounded (no common causes of child sex and LAZ or WLZ), stratifying by birthweight likely induces some bias in the effect away from the null.

For these two conceptual reasons, and the third practical reason that birthweight was not measured in a large number of cohorts (leading to restricted inference), we have not included this stratified analysis in the revision.

- 8. As a specific methodological issue, why impute using median/mode, for example compared to a multiple imputation approach? I suggest that the statistical reviewer is consulted on the implications of this. Regardless of the appropriateness of the method, for transparency there should be a table that shows extent of missingness by cohort and variable. Also for transparency, a complete case analysis should be presented as a sensitivity analysis.**

Response: As the referee has suggested, for transparency we added a new figure showing percent missingness by covariate and study and the complete-case sensitivity analysis (<https://child-growth.github.io/causes/missing.html>)

We would like to clarify that imputation was only used for covariates in adjusted analyses — outcomes and exposures were not imputed and so the study reflects a complete case analysis with respect to outcomes and exposures. We clarified this in the methods:

Lines 1091-1092

“Analyses used a complete case approach that only included children with non-missing exposure and outcome measurements.”

We used imputation for adjustment covariates to maximally include them where possible. To our knowledge, there is no straight-forward way to use a multiple imputation approach for covariates in individual cohort estimates that are then combined into an individual participant data meta-analysis. Instead, for covariates in the adjustment set, we imputed them at the median (for quantitative variables) or mode (for categorical variables) and additionally included an indicator variable to flag imputed values (Stuart 2010). The approach is technically sound and, compared with multiple imputation, easily integrates into the TMLE and meta-analysis analysis framework — we provide additional technical details below.

Technical details. Imputing covariates is relevant to our estimates of the two factors of the likelihood: (1) the outcome regression model of Y versus the explanatory variable of interest, A , and covariates, W (estimates of $E(Y|A, W)$) and (2) the estimate of the so-called treatment mechanism as a regression of categorical exposure A versus W ($P(A=a|W)$). There are many ways missingness can be handled and no theory that can be used to definitely predict which will handle missingness in the most efficient and least biased fashion. The method we used with indicators of missingness for any covariate that has missing information and simple imputations of the covariate values from summary statistics has several virtues. First, it allows us to make an explicit identification assumption, that is an assumption under which our estimators will be unbiased, so that we can be transparent and explicit about such an assumption. Specifically, our transformation of the W leads to a new set of measurable covariates, or, for every variable with missing data, j , we create a new W_j^*, Δ_j , where W_j^* contains either the original value of W_j if observed, or the imputed value, and Δ_j is the indicator of being observed ($=1$ if W_j was observed, 0 if missing). Then, the new W^* are used in the SuperLearner (SL) algorithm, an ensemble of regression models and machine learning algorithms for each of the factors in the likelihood. The SL is fit to minimize the squared residuals, and can use the flexibility to glean as much predictive information that is available in the actual covariate data, which W^* represents. Because the goal is “explain” as much residual variance as possible, the SL uses the actual observed data to control as much confounding as possible (assuming the original W would be preferable for this goal). Thus, the SL will use the predictive information contained in the Δ_j to adjust for confounding as aggressively as possible (that is if the fact that the values are missing is predictive in itself). In this new data structure, we thus can assert no residual unmeasured confounding based upon an explicit assumption, that is the variable of interest is independent of the counterfactuals, Y_a , conditional not on the fully measured W but on our transformation W^* , or $A \perp Y_a | W^*$. Another advantage of this approach is higher efficiency compared to an analysis that would exclude children with missing adjustment covariates. In addition, if many children are missing a single or just a few

weak confounders, then limiting the analysis to children with non-missing values of the weak confounders can result in a significant selection bias or become untenable if a very large proportion of observations have at least one covariate missing.

Stuart EA. Matching methods for causal inference: a review and a look forward. *Stat Science*. 2010;25(1):1–21.

Referee #2 (Remarks to the Author):

1. The detail and depth of the analysis are exemplary. Thank you for addressing my questions. I don't have additional comments, except that I spotted some typos in the revision:

Line 361: highlights should be plural.

Line 897-899: The sentence "We also calculated the difference in linear and ponderal growth velocities over three-month periods." appears twice.

Line 1022: " is show bin"

Response: Thank you for identifying these typos. We have fixed them all in the revision.

Referee #3 (Remarks to the Author):

Overall, the authors have adequately addressed issues I, as well as other reviewers have raised. The only pending points from my read are:

1. Lines 294-295: Authors do not provide a clear direction/ magnitude in this hypothesis.

Response: Thank you for this suggestion. The original wording was intentionally agnostic about the direction and magnitude because it differs by exposure and there are so many different exposures. In the revision, we replaced the topic sentence to provide two examples to help ground the hypothesis in more specific terms.

Lines 285-287:

"We hypothesized that causes of growth faltering could differ by age of growth faltering onset — for example, we expected children born preterm would have higher risk of incident growth faltering immediately after birth, while food insecurity might increase risk at older ages, after weaning."

2. Lines 399-405: There is a heavy focus on the conclusion the conclusions of the other papers.

Response: We agree. We revised the discussion to remove extensive reference to the companion papers.

- 3. Sex, specifically boys being consistently associated with being stunted, wasted or both, both as pooled estimate and across most study and regional contexts, was a finding of consequence and one that finds itself in the summary but is not flushed out as a finding at all throughout the paper. The authors may chalk this up to the fact that it is not modifiable but discussing this finding meaningful seems essential given the findings prominence.**

Response: We included a new sentence in the results that described this in more detail, and cites a recent review that proposed mechanisms. The revised text is:

Lines 251-252

“Girls had consistently better LAZ and WLZ than boys, potentially from sex-specific differences in immunology, nutritional demands, care practices, and intrauterine growth (Thurstans et al. 2022).”

Thurstans, S. et al. Understanding Sex Differences in Childhood Undernutrition: A Narrative Review. *Nutrients* 14, 948 (2022). <https://pubmed.ncbi.nlm.nih.gov/35267923/>

Referee #4 No remarks to the Author

Referee #5 (Remarks to the Author):

- 1. I would like to congratulate the authors on a tour de force of data harmonization, data analysis, and interpretations. What makes this work novel is that the underlying methodological framework allows for (almost) direct conclusions and comparisons across diverse cohorts. This would not have been possible without a statistical analytical framework, which separates sampling and data modelling, from interpretation. The former will be cohort depended while the latter can be universal. To the best of my knowledge this is the first time – within the subfield – such cross-cutting data analysis has been completed.**

Response: Thank you for this summary. We are grateful for the constructive suggestions, below.

- 2. A single methodological concern is if the cohorts have been “collected for a reason” and therefore perhaps are less representative for the country/population. This could be further addressed. In the methods section is it also somewhat**

unclear – at least for this reader – which values covariates are set to when computing population effects. Are they from the literature or derived from the cohorts? It seemed to be a mix. In short: the basic scientific merit and methodology is top-notch.

Response: This is an important detail that we have also discussed in our response to Referee 1 comment #2 (above). We agree that in the last version of the paper the chosen reference level for each exposure was listed but not in the most transparent way. We revised the previous Fig 2 (now Figs 2, 3) to include adjusted mean differences for each exposure level and clearly labeled the value each exposure was set to when estimating the population intervention effects (PIE). We agree with the referee that since the cohorts are not necessarily representative of the general population, that PIE estimates might be different if we had used different counterfactual distributions of the exposure. Many key exposures in this analysis are not measured in representative, population-based surveys, so it is difficult to establish a reference distribution for LMICs or study regions. Since this analysis draws from 33 cohorts across diverse populations we felt that PIE estimates would be meaningful even if based on the empirical distribution of the exposures in the cohorts. By additionally including the adjusted mean differences, which are independent of the exposure distribution, we have provided another view through which readers can assess the relative importance of each exposure.

We added this caveat to the Discussion:

Lines 363-367

“The analyses have caveats. Population intervention effects were based on exposure distributions in the 33 cohorts, which were not necessarily representative of the general population in each setting. Use of external exposure distributions from population-based surveys would be difficult because many key exposures we considered, such as at-birth characteristics or longitudinal diarrhea prevalence, are not measured in such surveys.”

- 3. The presentation is fairly verbal. It is of course a matter of taste if this is preferable. I would have preferred a few of the key-findings had been highlighted through tables. Instead the table material is mixes both summaries and very detailed. Given the amount of sensitivity analyses etc (which is of course good) it will be very difficult for the average reader to really take in the quantitative results. This is regrettable as one of the strengths of the paper exactly is the quantifications. In short: I suggest to make it more clear what the key tables are and where is more for the dedicated reader.**

Response: Our overall strategy was to use figures to synthesize the results, since the data underlying the analyses are very dense, and then use the text to highlight key findings (with reference to relevant main text and extended data figures). The referee's suggestion to summarize key points in a table or box is a very good suggestion. However, given the space limitations in the article format in the revision we have

continued to present the quantitative results in figures and use the text to summarize key messages from each figure to help general readers interpret the results.

- 4. The paper finds that the most important aspects causing missed growth are biological – and very hard to change. This has the obvious effect that whilst important biologically, it does really help for prevention. This is only partly acknowledged. It is said that the insight can be used to find out targets for interventions, but since no intervention can ever deliver the kind of changes hypothesized in the paper, it could be that the importance of certain aspects are over-sold. In short: The population effect reported in the paper should in some sense be multiplied by a factor (smaller than 1) corresponding to “ease of intervention”.**

Response: We acknowledge that many of the key determinants of growth faltering identified in this analysis are very hard to change and realistic interventions could not modify many of the characteristics over a short time frame — such as household wealth or maternal education. For unmodifiable characteristics, knowing they are important determinants can inform interventions because it motivates strategies to mitigate them (e.g., seasonal wasting) and/or targeting to subgroups. Comparison between risk factors on a single scale is also valuable. For example, preventing low birthweight would be expected to have similar impact on child LAZ as moving all households into the richest wealth quartile — this type of comparison puts into perspective how valuable interventions could be on that modifiable characteristic (low birthweight) for which there are known, effective interventions, such as nutritional supplementation for pregnant mothers and IPTp-SP treatment for malaria-endemic populations. Finally, demonstrating that key causes of growth faltering are in many cases intergenerational (from mothers) helps calibrate the global community to a realistic timeframe required to make progress on preventing child growth faltering — since our results show that prevention of child growth faltering will require improvements in maternal status, this implies a slower path to population-level improvements (years, decades) compared with health conditions that can be treated or prevented over the course of weeks or months.

We revised the final paragraph of the Discussion so that it clearly emphasizes modifiable characteristics and a path forward, namely: maternal nutritional status and child status at birth.

Lines 381 – 387

“Countries that have reduced stunting most have undergone improvements in maternal education, nutrition, reductions in number of pregnancies, and maternal and newborn health care, reinforcing the importance of interventions during the window from conception to one year, when fetal and infant growth velocity is high and energy expenditure for growth and development is about 50% above adult values (adjusted for fat-free mass). A stronger focus on prenatal interventions should not distract from renewed efforts for postnatal prevention. The observed pre- and postnatal growth

faltering we observed reinforce the need for sustained support of mothers and children throughout the first 1000 days. Efficacy trials that delivered prenatal nutrition supplements to pregnant mothers, therapeutics to reduce infection and inflammation for pregnant mothers, and nutritional supplements to children 6-24 months have reduced child growth faltering but have fallen short of completely preventing it. Our results suggest that the next generation of preventive interventions should focus on the early period of a child's first 1000 days — from preconception through the first months of life —because maternal status and at-birth characteristics are key determinants of growth faltering through 24 months. Halting the cycle of growth faltering early should reduce the risk of its severe consequences, including mortality, during this formative window of child development. Long-term investments and patience may be required, as it will take decades to eliminate the intergenerational factors limiting mothers' size.”